# FOREST-BASED GRAPH LEARNING FOR SEMI-SUPERVISED NODE CLASSIFICATION

**Jin Li[1], Shenghao Gao[1], Kaichen Zhang[1], Xinlong Chen[2], Ying Sun[1†], Hui Xiong[1,3†]**
[1]Thrust of Artificial Intelligence, The Hong Kong University of Science and Technology (Guangzhou)
[2]College of Computer and Data Science, Fuzhou University
[3]The Hong Kong University of Science and Technology (Hong Kong SAR)
jslijin2015@outlook.com, gaoshenghao512@gmail.com
kzhangbi@connect.ust.hk, fjxinlong@gmail.com
yings@hkust-gz.edu.cn, xionghui@ust.hk *

## ABSTRACT

Existing Graph Neural Networks usually learn long-distance knowledge via stacked layers or global attention, but struggle to balance cost-effectiveness and global receptive field. In this work, we break the dilemma by proposing a novel **f**orest-based **g**raph **l**earning (FGL) paradigm that enables efficient long-range information propagation. Our key insight is to reinterpret message passing on a graph as transportation over spanning trees that naturally facilitates long-range knowledge aggregation, where several trees–a forest–can capture complementary topological pathways. Theoretically, we demonstrate that as edge-homophily estimates improve, the induced distribution biases towards higher-homophily trees, which enables generating a high-quality forest by refining a homophily estimator. Furthermore, we propose a linear-time tree aggregator that realizes quadratic node-pair interactions. Empirically, our framework achieves comparable results against state-of-the-art counterparts on semi-supervised node classification tasks while remaining efficient. Codes are available at https://anonymous.4open.science/r/FGL/.

## 1 INTRODUCTION

Graph Neural Networks (GNNs) (Wu et al., 2020; Chen et al., 2020b; Thomas et al., 2022) attract much attention in recent years due to their expressivity in solving various graph-related tasks (*e.g.*, node and graph classifications (Feng et al., 2020; Xie et al., 2022) or clustering (Bianchi et al., 2020), link prediction (Yun et al., 2021), and anomaly detections (Dong et al., 2025; Gong et al., 2023)), with also many applications in, *e.g.*, texts (Wang et al., 2024b), images (Nazir et al., 2021; Guan et al., 2022), generation (Zhuang et al., 2025; Gong & Sun, 2025), traffic (Jiang & Luo, 2022), and other domains (Gong & Sun, 2024; Cui et al., 2026). Despite their popularity and successes, most GNNs restrict receptive fields to 2-/3-hop local neighborhoods and focus on nearby information aggregation while ignoring distant knowledge, which would limit their real-world application scopes when dealing with challenging tasks where long-range interactions are critical and necessary. For example, as discussed in Sec. A.1, the imbalance of densities or degrees often causes insufficient local knowledge for some nodes, which becomes more severe under graph heterophily and risks further over-fitness from label scarcity. In this paper, we focus on semi-supervised node classifications to underscore labeling challenges.

To facilitate long-distance interactions, existing works have devoted much effort and can be generally categorized into two different architectures: (1) Deep local models (*e.g.*, deep GNNs (Chen et al., 2022c; Li et al., 2019; Chen et al., 2020a)) expand the global receptive fields by stacking multiple local layers, with each considering only first-order information. (2) Shallow global models (*e.g.*, Global Graph Transformers (Ying et al., 2021; Kreuzer et al., 2021)) integrate 1 or 2 non-local aggregating operators (*e.g.*, global attentions), encapsulating all pairwise node interactions in a single layer. Unfortunately, most of them suffer from high time and space complexities (Li et al., 2021; Wu

---

*† Corresponding authors: Ying Sun (yingsun@connect.hkust-gz.edu.cn) and Hui Xiong (huixiong@ust.hk).

et al., 2022), due to excessive unparallelizable layers (former) or quadratic node-pair interactions (latter). Recently, few prior works attempt to mitigate complexities via some sparsity techniques such as Adaptive Selection (Chen et al., 2022b; Wu et al., 2022) and Graph Rewiring (Shirzad et al., 2023). Yet, they sacrifice global coverage and have to make selections, and thus either risk dropping some important node interactions or heavily rely on extra sophisticated selection strategies. Overall, such methods fail to simultaneously address comprehensive long-range knowledge extraction and cost-effectiveness, which is rooted in the inherent limitation of existing learning paradigms.

Such a graph learning dilemma urges us to rethink existing paradigms and explore an alternative that **breaks the unavoidable trade-off between cost-effectiveness and a global receptive field**. *The essential observation is that these paradigms view a graph as a fusion of structures*, whose total costs can be calculated as follows:

$$\text{Total cost} = (\text{cost per structure}) \times (\text{number of structures}). \tag{1}$$

Thus, when modeling with local primitives—first-order neighborhoods (Li et al., 2019) or short random walks (Zhang et al., 2020)—the per-structure cost is low, but numerous such structures are required for covering long distances. In contrast, global operators (Ying et al., 2021) can reduce the number of structures, yet at the expense of prohibitive per-structure cost due to dense pairwise interactions. Based on the above analysis, we naturally raise a question: *Does there exist a structure that simultaneously controls these two factors?* To answer this question, we recognize that a spanning tree is the minimal subgraph connecting all nodes. Therefore, under limited structure counts, such a tree is **the simplest structure that achieves global coverage** (Fig. 1), indicating that it may be more suitable for long-range propagation. Furthermore, we suggest using a forest (tree set), since a single spanning tree may be insufficient to capture all topological knowledge.

In this paper, we propose **f**orest-based **g**raph **l**earning (FGL), a novel paradigm that models information propagation on a graph as transport on a forest of spanning trees, economically achieving global coverage. To obtain a high-quality forest, we expect to sample the trees from a distribution biased towards homophilous trees. Theoretically, we demonstrate that as edge-homophily estimates improve, the induced tree distribution asymptotically approaches the ideal one. Accordingly, we propose a tree sampler, based on a well-trained edge-homophily estimator, to enable generating several spanning trees with higher homophily via the weighted Wilson algorithm (Wilson, 1996). Besides, we design a general tree aggregator [1], by deriving two recursions on trees, which propagates global messages in linear running time. Additionally, a post-hoc mean operator is adopted as our tree fuser to merge knowledge from different trees. These components constitute our full framework, as illustrated in Fig. 2.

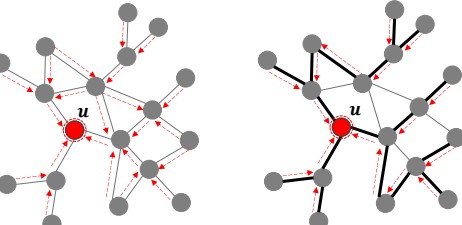

Figure 1: Our paradigm (right) utilizes the most sparse structures of a graph, *i.e.*, spanning trees, to aggregate global messages against the prior paradigms (left).

Our contributions are summarized as follows: 1. **New Paradigm**: We introduce a forest-based graph learning paradigm FGL, which can comprehensively capture long-range knowledge with high efficiency. 2. **Theoretical Insight**: We establish a rigorous asymptotic relationship between the accuracy of the edge-homophily estimator and the quality of the induced tree distribution, which reveals that refining the estimator provably yields a better tree distribution. 3. **Effective Approach**: We propose 1) a homophily estimator-based tree sampler, which generates homophilous trees with higher probability; and 2) a general tree aggregator that conducts quadratic pairwise node interactions with only linear complexities. 4. **Experimental Results**: Our framework achieves competitive results against state-of-the-art counterparts in semi-supervised node classifications with higher efficiency, *e.g.*, 11.90% and 16.14% average relative gains against GCNII and DIFFormer (representative Deep GNN and Graph Transformer), respectively.

---

[1] Here, a tree aggregator is to aggregate intra-tree messages, while a tree fuser merges the inter-tree messages.

## 2 RELATED LITERATURE

**Deep Local Models.** Deep Graph Neural Networks (GNNs) expand their receptive fields by iteratively stacking local aggregators, enabling fine-grained control over neighborhood information at each layer Yang et al. (2020); Fang et al. (2023); Chen et al. (2020a). This depth provides strong expressiveness but comes with notable drawbacks: sequential computation limits parallelism, higher time/space complexities, and the risk of over-smoothing. To mitigate over-smoothing, various strategies have been explored, including normalization layers Zhao & Akoglu (2020); Zhou et al. (2021); Yang et al. (2020), random dropping techniques Rong et al. (2020b); Huang et al. (2020); Fang et al. (2023), and skip connections Li et al. (2019); Chen et al. (2020a); Luan et al. (2019); Xu et al. (2018). Despite these improvements, deep local models inherently rely on step-wise neighborhood aggregation, which prevents efficient global message passing and parallelization.

**Shallow Global Models.** Graph Transformers (GTs) adopt a contrasting perspective: instead of gradual local aggregation, they model direct pairwise interactions among nodes, often in just a few global layers Min et al. (2022); Hussain et al. (2022); Ying et al. (2021). This shallow global paradigm ($G \approx x \rightarrow y_{x,y \in V}$) allows rapid global communication but typically incurs quadratic complexity. To improve scalability, recent works either sparsify interactions via sampling or pruning (*e.g.*, Gophormer Zhao et al. (2021), NodeFormer Wu et al. (2022), Exphormer Shirzad et al. (2023)), or simplify attention mechanisms to reduce computation (*e.g.*, SGFormer Wu et al. (2024), GOAT Kong et al. (2023)). While these strategies address efficiency, they often lose structural bias, motivating the use of positional encodings Ying et al. (2021); Chen et al. (2022a) or walk-based formulations Zhang et al. (2020). However, designing encodings that are both expressive and efficient remains challenging.

**Tradeoff Between Local and Global Models.** Deep Local Models excel at capturing fine-grained neighborhood structures but struggle with scalability and long-range dependencies. In contrast, Shallow Global Models enable efficient global message propagation with fewer layers, but often overlook nuanced local structures or incur high complexity without careful approximation. Hybrid designs attempt to combine both perspectives Wu et al. (2021); Rong et al. (2020a); Kreuzer et al. (2021). In contrast, we analyze the essential limitation of existing learning paradigms and propose a novel forest-based paradigm that enables efficient long-range modeling along with natural structural knowledge preservation, addressing this dilemma from a more fundamental perspective.

## 3 PRELIMINARY

**Notations.** Let $G = (V, E)$ be an unweighted graph with $n$ nodes $V = \{v_i\}_{i=1}^n$ and $m$ edges $E = \{e_{i,j}\}$. The graph is represented by a feature matrix $X \in \mathbb{R}^{n \times d}$ and an adjacency matrix $A \in \{0, 1\}^{n \times n}$, where $A_{ij} = 1$ if and only if $(v_i, v_j) \in E$. We also define the normalized adjacency matrix $\hat{A} = D^{-\frac{1}{2}}(A + I)D^{-\frac{1}{2}}$, where $D$ is the degree matrix of $A + I$.

**Problem Formulation.** In semi-supervised node classification, a subset of nodes $V_L \subset V$ has labels $y_i \in \{0, 1, ..., c-1\}$, while the remaining nodes are unlabeled. The goal is to learn node embeddings $H'' \in \mathbb{R}^{n \times d}$ such that a simple linear predictor can be applied to $H''$ to predict node labels for all $v_i \in V$, leveraging both labeled and unlabeled nodes.

## 4 METHOD

Existing paradigms suffer from the trade-off between cost-effectiveness and a global receptive field. To obtain global coverage, deep local models with small local structures require stacking *a large number of structures*, while shallow global models with large complex structures incur *high per-structure computational costs*. In this work, we introduce an intermediate-level structure—the tree—that offers a principled way to balance this trade-off, exhibiting a new learning paradigm. A tree connects all nodes in a graph in a cost-efficient and non-redundant manner.

We build on this insight to propose the **Forest-based Graph Learning (FGL)** framework illustrated in Fig. 2, which is composed of four key components: (1) **Pre-processing**, which augments the original input graph to facilitate downstream computation; (2) **Tree Sampler**, which derives a target distribution over spanning trees and generates multiple trees accordingly; (3) **Tree Aggregator**,

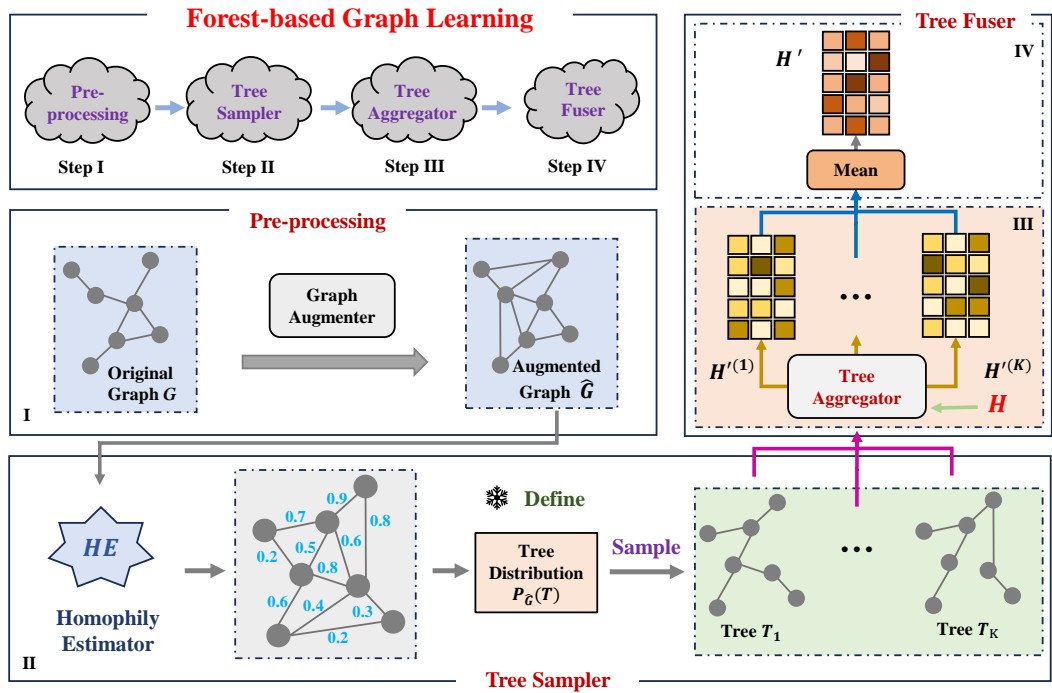

Figure 2: Our framework contains 4 key steps: (I) **Pre-processing** first augments the vanilla graph; (II) **Tree Sampler** then generates multiple spanning trees from a derived distribution; (III) **Tree Aggregator** efficiently propagates messages ($H$ in Eq. 9) over each tree next; and (IV) **Tree Fuser** finally integrates the aggregated messages from all trees into unified embeddings $H'$.

which performs message passing along each individual spanning tree; and (4) **Tree Fuser**, which integrates the aggregated messages from all sampled trees into a unified representation.

## 4.1 PRE-PROCESSING

Real-world graphs are often not connected, which hinders the subsequent spanning tree sampling process. To address this issue, we begin by computing pseudo-labels for each node, denoted as $Y' \in \mathbb{R}^{n \times c}$. For heterophilous graphs, we employ a simple feed-forward layer, $Y' = \sigma(XW)$ whereas for homophilous graphs, we use a GCN layer, $Y' = \sigma(\hat{A}XW)$. The learnable parameters $W \in \mathbb{R}^{d \times c}$ are optimized on the labeled nodes using the standard cross-entropy loss. We then construct an augmented graph $\hat{G}$ by leveraging the pseudo-labels. For each node, we use its pseudo-label representation $y' \in \mathbb{R}^{1 \times c}$ to identify its $k$ nearest neighbors. If an edge does not already exist between the node and one of these neighbors, we introduce a new edge.

This pre-processing step offers two key benefits at the same time. First, it ensures graph connectivity, which is necessary for subsequent spanning tree sampling. Second, it increases the *homophily ratio*—the proportion of edges linking nodes with similar class labels—which has been shown to improve performance in semi-supervised node classification (Chien et al., 2021).

## 4.2 TREE SAMPLER

To generate a high-quality forest composed of several spanning trees, we identify two essential principles: 1) *homophily ratios*: Since we target node classification, it is a critical measure on graphs and thus can be naturally transferred to trees. 2) *diversity*: if these trees tend to overlap, then the forest would be degraded into a single tree, which may be insufficient to cover all the topological knowledge of a graph, therefore necessitating diversity.

Therefore, we expect to sample the trees independently from a distribution $P_{\hat{G}}(T)$ biased towards trees with high homophily ratios. We assume each tree $T$ has a score $s(T)$ that can be calculated as the product of edge scores $s(e)$, thereby defining the tree distribution on a graph as follows:

$$P_{\hat{G}}(T) = \frac{s(T)}{\sum_{T \subseteq \hat{G}} s(T)} = \frac{\prod_{e \in T} s(e)}{\sum_{T \subseteq \hat{G}} \prod_{e \in T} s(e)}. \tag{2}$$

The only remaining step is to determine the edge scores $s(e)$. Our main idea is to assign higher scores to those homophilous edges and lower scores to heterophilous edges, which intuitively improves

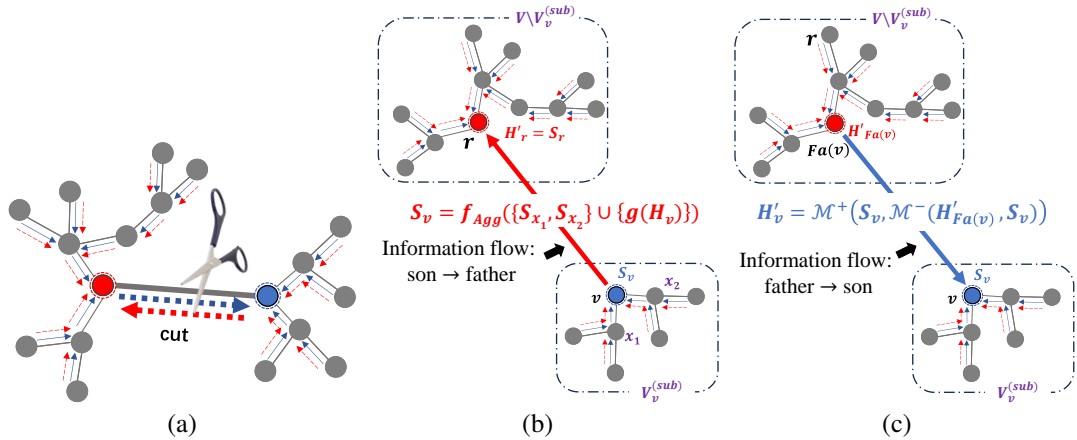

Figure 3: Illustration of the tree aggregator. The red node denotes the root, and the blue node indicates the focal node. (a) Red dashed lines depict the bottom-up computation of $S$, while blue dashed lines represent the computation of $H'_v$. (b)(c) Detailed computations along the focal edge are shown.

the probabilities assigned to homophilous trees. We justify this intuition in Sec. 4.6 by theoretically demonstrating that this scoring strategy can induce a distribution biased towards higher-homophily trees (Theorem 2). Therefore, we introduce a homophily estimator to find those homophilous edges and assign higher scores to them. Here, we implement this homophily estimator via local attention:

$$\alpha_{i \to j} = \frac{\exp\left(Q_i K_j^\top / \sqrt{c}\right)}{\sum_{v \in \mathcal{N}(i)} \exp\left(Q_i K_v^\top / \sqrt{c}\right)}, \quad \forall\, i, j \in V \tag{3}$$

where $Q = XW_Q$, $K = XW_K$, and $V = XW_V$ with learnable $W_Q, W_K, W_V \in \mathbb{R}^{d \times c}$. $\mathcal{N}_i$ denotes the first-order neighborhood of node $i \in V$. We train the local graph attention by minimizing the cross-entropy loss with targets $Y'$. Thus, the edge score $s(e)$ for $e = (i, j)$ is defined by $s(e) = (\alpha_{i \to j} + \alpha_{j \to i})/2$. Finally, our tree sampler generates $N_T$ independent spanning trees from $P_{\widehat{G}}(T)$ via the algorithm of Wilson (1996) in nearly $\mathcal{O}(n)$ time per-tree.

### 4.3 Tree Aggregator

The tree aggregator $f_{\text{Agg}}^{(T)}$ over tree $T$ with root $r$ is defined as $f_{\text{Agg}}^{(T)} : H \in \mathbb{R}^{n \times d} \mapsto H' \in \mathbb{R}^{n \times d}$, which is designed based on a general message aggregator $f_{\text{Agg}}(\cdot)$. The idea is rooted in a key observation: for neighboring nodes $u, v$ on tree $T$, the globally merged messages targeting them differ only at one edge direction (visualized in Fig. 3). Leveraging this observation can facilitate efficient tree propagation by any general $f_{\text{Agg}}(\cdot)$ that satisfies: given two message sets $A, B$ with possible auxiliary information (*e.g.*, weights), if merging $A$ into $B$ getting $S$, then there always exists two operators $\mathcal{M}^{+/-}(\cdot)$ to make the following sufficient properties hold.

$$
\begin{aligned}
f_{\text{Agg}}(S) &= \mathcal{M}^+\left(f_{\text{Agg}}(B),\ f_{\text{Agg}}(A)\right), && \textbf{Property (I): Combine} \\
f_{\text{Agg}}(B) &= \mathcal{M}^-\left(f_{\text{Agg}}(S),\ f_{\text{Agg}}(A)\right), && \textbf{Property (II): Disentangle}
\end{aligned}
\tag{4}
$$

where $\mathcal{M}^{+/-}\left(\vec{a}, \vec{b}\right)$ denote adding vector $\vec{b}$ to $\vec{a}$ or deleting $\vec{b}$ from $\vec{a}$, which are allowed unsymmetrical via auxiliary information. These identified properties do not sacrifice the generality of $f_{\text{Agg}}(\cdot)$. Indeed, many popular auto-regressive sequence models and first-order GNN aggregators can be adopted, *e.g.*, linear attention Zhou et al. (2021); Wu et al. (2024), linear Recurrent Neural Networks (RNNs) Liu et al. (2024), and State Space Models (SSMs) Sarem et al. (2024); Zhang et al. (2025); Xiao et al. (2024) as well as non-linear variants (Sec. A.6), thus highlighting its generality.

Based on these properties, we can theoretically derive a general tree aggregator $f_{\text{Agg}}^{(T)}$ high-levelly via two recursions in Theorem 1. The proof and further explanation can be found in Sec. B.1 of Appn.

**Theorem 1.** *Given a tree $T$ with a root $r \in V$, each node $v \in V$ has a subtree $T_v^{(\text{sub})}$ with nodes $V_v^{(\text{sub})} \subseteq V$. Denote the father node and the children nodes of $v$ on tree $T$ as $\mathrm{Fa}(v)$ and $\mathrm{Child}(v)$. Let $S_v$ represent the aggregated message at node $v$ from all messages from $V_v^{(\text{sub})}$. Then, given any message aggregator $f_{\text{Agg}}(\cdot)$ satisfying Properties (I) and (II) as well as function $g(\cdot)$, our tree aggregator $f_{\text{Agg}}^{(T)} : H \mapsto H' \in \mathbb{R}^{n \times d}$ can be always derived as two recursions via operators $\mathcal{M}^{+/-}$:*

$$\forall\, u \in V, \quad S_u = f_{\text{Agg}}\left(\{S_v\}_{v \in \mathrm{Child}(u)} \cup \{g(H_u)\}\right), \quad \textbf{Recursion (I)} \tag{5}$$

$$\forall\, v \in V, \quad H'_v = \mathcal{M}^+ \left( S_v, \, \mathcal{M}^- \left( H'_{\mathrm{Fa}(v)}, S_v \right) \right), \quad H'_r = S_r, \quad \textbf{\textit{Recursion (II)}} \quad (6)$$

*where $H, H' \in \mathbb{R}^{n \times d}$ denote node embeddings before and after aggregation.*

This theorem provides an efficient way to propagate long-distance information on a tree: (1) First, to calculate $S_u$ for each node $u \in V$, it suffices to collect all distant messages targeting the root once, by recursively calling $f_{\mathrm{Agg}}(\cdot)$; (2) Then, apart from the root $H'_r = S_r$, we can calculate $H'$ for other nodes efficiently via the operator $\mathcal{M}^-$ followed by $\mathcal{M}^+$.

**Implementation** Despite the strong generality, we still prioritize a linear variant for simplicity and ease of implementation. Specifically, adopting $f_{\mathrm{Agg}}$ and $\mathcal{M}^+$ as weighted sums, $\mathcal{M}^-$ as weighted difference, and $g$ as a linear transformation, we implement Eq. 5 and Eq. 6 as follows:

$$\forall\, u \in V, \quad S_u = \sum_{v \in \mathrm{Child}(u)} (\alpha_{v \to u} \cdot W_A) \cdot S_v + W_B \cdot H_u \in \mathbb{R}^d, \quad (7)$$

$$\forall\, v \in V, \quad H'_v = S_v + \alpha_{\mathrm{Fa}(v) \to v} \cdot W_A \cdot \left( H'_{\mathrm{Fa}(v)} - \alpha_{v \to \mathrm{Fa}(v)} \cdot W_A \cdot S_v \right) \in \mathbb{R}^d, \quad (8)$$

where $W_A \in \mathbb{R}^{d \times d}$ and $W_B \in \mathbb{R}^{d \times d}$ are learnable matrices. The local attentions $\{\alpha_{i \to j}\}_{i,j}$ (defined in Eq. 3) are utilized to enhance the impact of homophilous edges and weaken heterophilous edges.

**Acceleration and Extensions** Note that parallelization can be conducted both between trees and between aggregations inside a single tree. For higher parallelization, we can intuitively make a rooted tree shallower yet wider to support many threads working together by selecting its centroid as the root. Furthermore, there exist different greedy strategies for nodes' priority for different recursions (Eq. 5 and Eq. 6) to reduce the waiting time of threads. We discuss their specific implementations in Sec. D of Appn. Due to space limits, we will discuss more on several potential extensions of the above tree aggregators in Sec. C of Appn., which includes how to: (1) efficiently integrate a global linear attention to the framework similar to Wu et al. (2024) and conveniently incorporate the kernel decomposition techniques (*e.g.*, Random Feature Likhosherstov et al. (2022)) to improve the expressivity of attention; (2) conduct fine-grained propagation control, such as discounting or truncating the distance, similar to some deep GNNs Xu et al. (2018); Chen et al. (2020a); (3) generalize forests to eliminate the need for Recursion (II), *i.e.*, Eq. 6.

### 4.4 TREE FUSER

Motivated by prior work Wu et al. (2024); Kreuzer et al. (2021); Wu et al. (2021), we utilize a local module to supplement local knowledge to mitigate the local sparsity of trees. Thus, the tree fuser first computes the local information $H$ from input features $X$, which is formalized as below:

$$H = \left( \beta_1 \cdot \widehat{A}_{\widehat{G}} + \beta_2 \cdot \alpha + (1 - \beta_1 - \beta_2) \cdot \mathbb{I}_{n \times n} \right)^{K_L} X W_H \in \mathbb{R}^{n \times d}, \quad (9)$$

where $\beta_1 + \beta_2 \le 1, K_L \le 2$ are hyper-parameters and $W_H$ are training parameters.

The tree fuser then computes the results of $N_T$ different tree aggregators, $H'^{(k)} = f_{\mathrm{Agg}}^{(T_k)}(H)$, $k \in [1, N_T]$. For each $H'^{(k)}$, the tree fuser normalizes each row to 1 using the $L_2$-norm for numerical stabilization. Afterwards, the tree fuser averages all the tree aggregators as global information:

$$H' = \mathrm{Mean}\left( \left\{ \mathrm{RowNorm}\left( H'^{(k)} \right) \right\}_{k \in [1, N_T]} \right) \in \mathbb{R}^{n \times d}. \quad (10)$$

Subsequently, the tree fuser uses a residual connection controlled by the hyper-parameter $\gamma \in [0, 1]$ to balance local and global information, which can be formulated as follows:

$$H'' = (1 - \gamma) \cdot H' + \gamma \cdot H. \quad (11)$$

The $H''$ are final node embeddings that can be fed into a linear predictor for node classification.

### 4.5 COMPLEXITY ANALYSIS

The comprehensive time and space complexities per epoch are linear against the number of nodes and edges, *i.e.*, $n$ and $m$, as well as hidden dim $d$. Specifically, suppose we sample and utilize $N_T$ trees. Each pre-training epoch costs $\mathcal{O}\left( (n + m)\, d \right)$ time and space. Each training epoch of the student requires only $\mathcal{O}\left( (n + m)\, K d \right)$ time and space, which can be further parallelized.

Table 1: The results of performance comparison (with the best bolded and the runner-ups underlined)

| Method | Category | Cora | Citeseer | Pubmed | Actor | Cornell | Texas | Wisconsin | Arxiv | Flickr | Avg. Rank |
|---|---|---|---|---|---|---|---|---|---|---|---|
| MLP | Classic | 58.30 | 58.68 | 72.94 | 35.62 | 72.70 | 77.84 | 79.61 | 32.84 | 42.01 | 14.11 |
| GCN | GNN | 82.06 | 71.60 | 79.58 | 27.88 | 53.51 | 69.19 | 57.25 | 53.77 | 38.40 | 14.89 |
| GAT | GNN | 82.84 | 72.28 | 78.52 | 28.71 | 55.14 | 68.65 | 58.82 | 55.73 | 40.32 | 12.78 |
| GraphSAGE | GNN | 81.40 | 71.68 | 78.50 | 36.24 | 63.78 | 75.14 | 76.08 | 51.42 | 41.42 | 11.00 |
| SuperGAT$_{SD}$ | GNN | 82.70 | 72.50 | **81.30** | 30.18 | 54.59 | 69.73 | 58.04 | 51.52 | 36.24 | 13.22 |
| APPNP | GNN | 84.10 | 72.14 | 80.02 | 33.47 | 61.08 | 71.35 | 65.10 | 55.60 | 43.07 | 9.22 |
| ClusterGCN | GNN | 82.04 | 70.08 | 77.26 | 29.66 | 49.73 | 63.24 | 62.35 | 53.35 | 39.58 | 16.89 |
| GraphSAINT | GNN | 82.00 | 70.30 | 77.36 | 29.55 | 48.65 | 63.78 | 61.96 | 53.55 | 35.26 | 17.67 |
| Pairnorm | DeepGNN | 66.24 | 44.20 | 72.12 | 24.33 | 40.68 | 41.08 | 52.94 | 54.58 | 31.41 | 22.56 |
| Nodenorm | DeepGNN | 80.14 | 65.74 | 78.64 | 29.74 | 40.00 | 66.49 | 48.24 | 54.22 | 44.11 | 16.33 |
| Meannorm | DeepGNN | 79.54 | 72.16 | 73.06 | 25.46 | 25.41 | 61.62 | 52.94 | 20.37 | 42.40 | 19.67 |
| DropEdge | DeepGNN | 81.69 | 71.43 | 79.06 | 26.38 | 52.97 | 64.86 | 60.78 | 39.23 | 32.11 | 18.33 |
| GCNII | DeepGNN | 85.34 | 73.24 | 79.88 | 34.64 | 74.61 | 69.19 | 70.31 | 51.91 | 41.79 | 8.78 |
| ShadowGNN | DeepGNN | 82.32 | 70.06 | 77.30 | 29.45 | 51.35 | 64.32 | 62.35 | 53.35 | 37.59 | 17.00 |
| GT | GT | 77.58 | 66.96 | 76.48 | 37.15 | 61.62 | 74.60 | 71.76 | OOM | OOM | 15.57 |
| SAN | GT | 77.60 | 68.64 | 76.62 | 37.79 | 63.24 | 75.14 | 77.25 | OOM | OOM | 13.00 |
| Graphormer | GT | 63.08 | 61.08 | OOM | OOM | 62.70 | 76.76 | 72.16 | OOM | OOM | 15.40 |
| ANS-GT | GT | 77.68 | 64.16 | 77.98 | 38.29 | 74.92 | 76.22 | 76.47 | 41.83 | 21.86 | 13.22 |
| Nodeformer | GT | 79.02 | 69.66 | 76.06 | 34.80 | 68.11 | 77.84 | 76.47 | 39.47 | 40.31 | 13.11 |
| NAGphormer | GT | 79.51 | 67.34 | 78.32 | 37.33 | 63.78 | 71.89 | 66.27 | 52.00 | 38.59 | 13.44 |
| GOAT | GT | 83.18 | 71.99 | 79.13 | 37.66 | 64.32 | 76.76 | 73.33 | 52.46 | 35.53 | 9.11 |
| Exphormer | GT | 82.77 | 71.63 | 79.46 | 35.53 | 62.16 | 75.68 | 70.98 | 41.12 | 22.79 | 12.67 |
| SGFormer | GT | 82.38 | 71.82 | 80.64 | 37.80 | 68.65 | 78.92 | 80.00 | 45.73 | 40.13 | 7.22 |
| DIFFormer | GT | 83.32 | 74.46 | 78.16 | 34.51 | 60.00 | 68.11 | 63.92 | 53.60 | 44.25 | 10.56 |
| TDGNN | GT | 85.35 | 73.78 | 80.20 | 32.84 | 35.68 | 61.35 | 46.86 | OOM | 38.25 | 15.00 |
| GraphMamba | Mamba | 54.36 | 58.98 | 70.90 | 36.05 | 74.05 | 77.29 | 80.39 | 33.59 | 42.30 | 13.89 |
| Ours | Forest | **85.46** | 74.42 | 81.00 | **39.88** | **83.24** | **91.89** | **86.27** | **56.47** | **47.22** | **1.22** |

## 4.6 THEORETICAL DISCUSSION

In this subsection, we provide theoretical justification for a rigorous asymptotic relationship between the accuracy of the edge-homophily estimator and the quality of the induced tree distribution. Formally, we define $P_{\widehat{G}}(T) = \prod_{e_{i,j} \in T} s(e_{ij}) / \sum_{T \subseteq \widehat{G}} \prod_{e_{i,j} \in T} s(e_{ij})$, where the edge score is given by $s(e_{ij}) = p$ if nodes $i$ and $j$ share the same label (a homophilous edge), and $s(e_{ij}) = q$ otherwise (a heterophilous edge). Based on this formulation, we establish the following result:

**Theorem 2.** *Let $\widehat{G}$ be any connected graph, and define the expected edge homophily ratio under the score ratio $\Delta = p/q > 0$ as:*

$$R_{\widehat{G}}(\Delta) := \mathbb{E}_{T \sim P_{\widehat{G}}^{(p,q)}}[h(T)],$$

*where $h(T)$ is the edge homophily ratio of tree $T$. Then there exists a $\Delta_0 > 0$ such that:*

- **Monotonicity.** *If $\Delta > \Delta' \geq \Delta_0$, then $R_{\widehat{G}}(\Delta) > R_{\widehat{G}}(\Delta')$.*

- **Upper Bound.** *For all $\Delta \geq \Delta_0$, $R_{\widehat{G}}(\Delta) \leq 1 - \frac{\text{NHCC}(\widehat{G})-1}{n-1}$, where $\text{NHCC}(\widehat{G})$ denotes the number of homophilous connected components of $\widehat{G}$.*

- **Asymptotic Tightness.** *As $\Delta \to +\infty$, $R_{\widehat{G}}(\Delta) \to 1 - \frac{\text{NHCC}(\widehat{G})-1}{n-1}$.*

Theorem 2 shows that, for a given graph $\widehat{G}$, as the ratio $\Delta = p/q$ increases, $P_{\widehat{G}}(T)$ gradually shifts toward homophilous trees. Moreover, the upper bound of $R_{\widehat{G}}(\Delta)$ is determined by the number of homophilous connected components in $\widehat{G}$, which reflects the inherent structural limitation of the graph. In the limit $\Delta \to +\infty$, $R_{\widehat{G}}(\Delta)$ approaches this structural bound. In other words, assigning a higher score $p > 0$ to homophilous edges and a lower score $q > 0$ to heterophilous edges drives $P_{\widehat{G}}(T)$ toward the maximum level of edge homophily permitted by the graph.

## 5 EXPERIMENTS

This section verifies the effectiveness of the proposed method in the semi-supervised node classification task via extensive experiments. Due to space limits, some experimental details such as environments, dataset statistics, algorithm implementation details, hyperparameter optimization strategy and configurations, and some visualizations are moved to Sec. K of Appn.

**Benchmarks and Baselines** The experiments include nine real-world benchmarks, covering two types: (1) homophilous graphs: Cora, Citeseer, Pubmed (Sen et al., 2008), and OGBN-ArXiv (Hu et al., 2020) at a large node scale; (2) heterophilous graphs: Flickr (Zeng et al., 2019), Texas, Wisconsin, Cornell (Pei et al., 2020a), and Actor (Tang et al., 2009). Their full statistics are detailed in Tab. 7 of Appn. For a fair comparison, semi-supervised data splits are adopted for OGBN-ArXiv

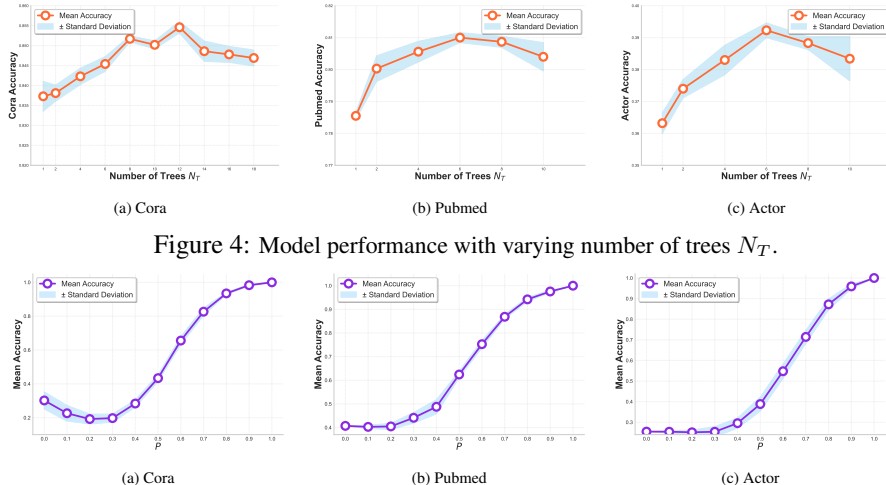

Figure 4: Model performance with varying number of trees $N_T$.

(a) Cora     (b) Pubmed     (c) Actor

Figure 5: Effect of homophily estimator accuracy ($p$ is the average score assigned to homophilous edges).

and Flickr (Sec. K.2), and other datasets strictly follow the standard public splits in (Kipf & Welling, 2017). Twenty-six counterparts are selected for a thorough comparison, including: **(1)** *classic method*: MLP; **(2)** *seven GNNs*: GCN (Li et al., 2019), GAT (Veličković et al., 2018), GraphSAGE, SuperGAT$_{SD}$ (Kim & Oh, 2021), APPNP (Gasteiger et al., 2019a), ClusterGCN (Chiang et al., 2019) and GraphSAINT (Zeng et al., 2019); **(3)** *six Deep GNNs*: Pairnorm (Zhao & Akoglu, 2020), Nodenorm (Zhou et al., 2021), Meannorm (Yang et al., 2020), DropEdge (Rong et al., 2020b), GCNII (Chen et al., 2020a) and ShadowGNN (Zeng et al., 2021); **(4)** *eleven Graph Transformers*: GT (Dwivedi & Bresson, 2020), SAN (Kreuzer et al., 2021), Graphormer (Ying et al., 2021), ANS-GT (Zhang et al., 2022), NodeFormer (Wu et al., 2022), GOAT (Kong et al., 2023), NAGphormer (Chen et al., 2022b), Exphormer (Shirzad et al., 2023), SGFormer (Wu et al., 2024), DIFFormer (Wu et al., 2023), and TDGNN (Qu et al., 2020); **(5)** *Mamba*: GraphMamba (Wang et al., 2024a),

**Comparative Experiments** All experiments run with ten different initializations. We report mean accuracy in Tab. 1 with also their standard deviations in Tab. 10 of Appn. We empirically show our framework has significant advantages for both homophilous and heterophilous datasets: against GT, DIFFormer, GCN, and GCNII, the mean accuracy is relatively increased by 16.2%, 16.1%, 24.5% and 11.9%, respectively. Particularly on Wisconsin, we obtain 20.2%, 35.0%, 50.7%, and 22.7% relative gains. Against recent models like TDGNN, ShadowGNN, and GraphSAINT, our framework also shows significant relative gains of 39.3%, 24.8%, and 27.0%, respectively. These performance gains are attributed to our ability to effectively capture long-distance knowledge, thus highlighting the potential of the proposed forest-based paradigm, even under label scarcity.

**Ablation Studies** We conduct ablation studies in Tab. 3 and drop or substitute key parts. For convenience, we refer to Eq. 9 and Eq.10 as Local and Global Submodules, respectively. We **(1)** drop Global Submodules to verify its long-range modeling capability; **(2)** drop Local Submodules to test the effects of supplementing local knowledge; **(3)** Sample trees from a uniform distribution and apply the attention weighting mechanism from Eq 7-8.; **(4)** sample only a single tree to explore the potential of multi-tree fusion. Comparing **(4) vs. (3)** reveals that sampling a single tree from the homophily-guided distribution outperforms multiple random trees, emphasizing the importance of homophily-based tree sampling. Comparing **(1)(2) vs. (5)** shows the significance of each submodule. Comparing (5) vs. (4) shows sampling multiple trees (a forest) can consistently surpass a single tree from our distribution, confirming that a forest can effectively capture more comprehensive and complementary topological knowledge.

Table 2: Running time comparison (sec/epoch)

| Method | Cora | Citeseer | Pubmed | Flickr | ArXiv |
|---|---|---|---|---|---|
| GT | 0.011 | 0.014 | 0.254 | OOM | OOM |
| SAN | 0.165 | 0.154 | 0.241 | OOM | OOM |
| Graphormer | 0.433 | 0.639 | OOM | OOM | OOM |
| ANS-GT | 1.453 | 2.973 | 3.433 | 7.796 | 24.540 |
| Nodeformer | 0.188 | 0.217 | 0.292 | 0.838 | 1.360 |
| NAGphormer | 0.022 | 0.044 | 0.031 | 0.835 | 1.560 |
| GOAT | 1.026 | 1.045 | 1.450 | 28.281 | 58.772 |
| Exphormer | 0.086 | 0.175 | 0.348 | 1.112 | 1.948 |
| SGFormer | 0.010 | 0.011 | 0.021 | 0.051 | 0.114 |
| DIFFormer | 0.029 | 0.030 | 0.047 | 0.297 | 0.545 |
| GraphSAINT | 0.013 | 0.022 | 0.030 | 0.658 | 0.951 |
| Pairnorm | 0.053 | 0.071 | 0.647 | 0.320 | 1.387 |
| Nodenorm | 0.013 | 0.032 | 0.285 | 0.310 | 1.357 |
| Meannorm | 0.012 | 0.030 | 0.279 | 0.296 | 1.461 |
| Dropedge | 0.017 | 0.017 | 1.231 | 1.244 | 1.491 |
| GCNII | 0.066 | 0.033 | 1.306 | 1.373 | 2.843 |
| **Ours** | 0.005 | 0.019 | 0.020 | 0.079 | 0.246 |

**Hyper-Parameter Studies** We conduct several hyper-parameter studies in Sec. J.1 Here, due to space limits, we focus only on the impact of the tree number $N_T$ on performance in Fig. 4, which reveals an optimal range of 6 to 10 trees across different datasets, highlighting our efficient coverage of global knowledge. In Fig. 4, the performance first consistently rises and then fluctuates or decreases, meaning that our framework covers the essence of the graph structure with only a few trees, and

Table 3: The results of ablation studies.

| No. | Method | Cora | Citeseer | Pubmed | Actor | Cornell | Texas | Wisconsin | ArXiv | Flickr |
|---|---|---|---|---|---|---|---|---|---|---|
| **(1)** | *w.o. Global Submodule* | 80.00 | 71.63 | 76.13 | 34.73 | 75.68 | 82.88 | 83.92 | 55.05 | 39.63 |
| **(2)** | *w.o. Local Submodule* | 82.18 | 71.55 | 77.48 | 35.08 | 74.77 | 69.93 | 75.49 | 54.92 | 32.17 |
| **(3)** | *Uniform Tree Sampling* | 83.63 | 72.32 | 78.45 | 36.13 | 72.97 | 82.58 | 84.80 | 55.11 | 42.77 |
| **(4)** | *Single Homophily-guided Tree* | 83.73 | 72.58 | 78.55 | 36.32 | 76.35 | 84.83 | 85.29 | 55.17 | 42.96 |
| **(5)** | `FGL - Ours` | **85.46** | **74.42** | **81.00** | **39.88** | **83.24** | **91.89** | **86.27** | **56.47** | **47.22** |

more trees provide marginal benefits and risk redundancy, highlighting our efficiency due to *a lower number of structures* in the calculation of the total cost, *i.e.*, Eq. 1.

**Efficiency Comparison**  Besides the theoretical complexity analysis in Sec. 4.5, we compare the practical running time in Tab. 2, where our method runs faster than baselines in most cases. For example, compared with recent GTs like ANS-GT and GOAT, which require over 1 second per epoch on small graphs and dozens of seconds on large graphs, our method runs in under 0.02 seconds on small graphs and 0.246 seconds on ArXiv. Even against efficient GTs like DIFFormer and deep GNNs like GCNII, our method shows 2 to 5 times speedup. While a few baselines run slightly faster than ours, their performance is generally worse than ours, since they overlook some critical structural knowledge due to over-simplified designs. Compared with these baselines with strong performance, we have the highest efficiency, highlighting the advantages of the linear complexities and higher parallelizability of the proposed forest-based learning paradigm.

**Homophily Estimator Comparison**  To explore the effects of different homophily estimators, we compare six variants in Tab. 4: (A) Non-attention auxiliary module (NAAM) via single-layer GCN for homophilous graphs or MLP for heterophilous graphs to generate pseudo-labels; (B) Naive attention based estimator via a single local graph transformer layer where attention coefficients serve as bidirected average edge homophily scores; (C) 2-stage homophily estimation that first generates pseudo-labels via non-attention estimator, then uses these labels to guide the training of attention-based estimator for more stable homophily scores; (D) `FGL` (Uniform) as baseline that samples trees uniformly; (E) `FGL` (Naive attention estimator) that uses attention scores from (B) to guide tree sampling; (F) `FGL` (2-stage estimator) - Ours, incorporating the full two-stage estimation process for robust homophily-guided tree sampling. Comparing (B) vs. (E), `FGL` using an attention-based estimator performs competitive or better than the standalone attention estimator, demonstrating `FGL`'s effective utilization of homophily scores through structured tree aggregation. Comparing (C) vs. (E), two-stage estimation significantly outperforms `FGL` with only attention-based estimation in most cases, confirming that pseudo-labels from non-attention estimators provide valuable supervision to improve homophily estimation quality, especially under label scarcity. These empirical observations further support our theoretical analysis (Theorem 2) and directly confirm the accuracy of the edge homophily estimator has a positive impact on our final results.

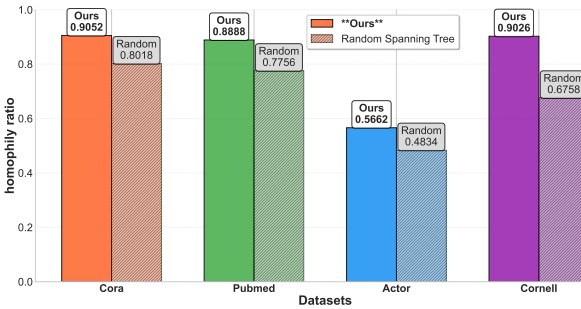

Figure 6:  homophily ratio comparison based on different sampling strategies

**Interpretability Studies**  We propose a strategy to design our tree distribution, which is justified by Theorem 2. Here, we provide some empirical evidence to understand our performance gains. Fig. 5 reveals that as the accuracy of homophily estimator increases, model performance consistently improves across all datasets, with perfect estimation (accuracy is 1) leading to perfect classification, demonstrating no performance bottleneck and motivating the pursuit of high-quality homophily estimators. To further understand the mechanism, we introduce a global homophily metric (Sec. J.2). Fig. 6 shows that trees sampled from our homophily-guided distribution significantly facilitate higher long-range homophilous information propagation compared to uniform sampling. Such trees allow the subsequent tree aggregator much easier to capture and exploit beneficial distant graph information, which fundamentally interprets our performance gains.

## 6    CONCLUSION

To break the dilemma of existing graph techniques, *i.e.*, the challenging trade-off between complexities and comprehensive long-distance knowledge, we fundamentally analyze its root cause and propose a

Table 4: Comparison of different homophily estimators.

| No. | Model | Cora | Citeseer | Pubmed | Actor | Cornell | Texas | Wisconsin | ArXiv | Flickr |
|-----|-------|------|----------|--------|-------|---------|-------|-----------|-------|--------|
| **(A)** | Non-attention auxiliary module (NAAM) | 78.42 | 69.62 | 76.64 | 35.33 | 72.97 | 72.97 | 82.35 | 47.65 | 38.36 |
| **(B)** | Naive attention based estimator | 75.18 | 65.78 | 74.32 | 34.87 | 70.27 | 75.00 | 73.04 | 53.45 | 40.90 |
| **(C)** | Two-stage (NAAM + attention) estimator | 81.40 | 70.30 | 78.68 | 36.20 | 78.38 | 83.78 | 82.75 | 53.99 | 43.30 |
| **(D)** | FGL (Uniform) | 78.40 | 73.13 | 71.54 | 34.47 | 71.62 | 70.27 | 74.51 | 52.30 | 41.05 |
| **(E)** | FGL (Naive attention) | 81.60 | 73.38 | 75.10 | 35.56 | 74.32 | 75.00 | 76.75 | 53.63 | 41.61 |
| **(F)** | FGL (2-stage) - Ours | **85.46** | **74.42** | **81.00** | **39.88** | **83.24** | **91.89** | **86.27** | **56.47** | **47.22** |

novel forest-based graph learning paradigm. The key insight is to understand a graph as a fusion of some sampled spanning trees, similar to bagging, since a tree can connect all nodes economically. We provide a technical framework, where we first induce a tree distribution proven biased towards homophily, and then efficiently conduct all node-pair interactions in each tree via a general tree aggregator with linear complexities and higher parallelizability. Compared with deep GNNs or GTs, our framework has better global coverage and structural understanding, with higher efficiency. Extensive experiments on semi-supervised node classifications show we can achieve competitive or even better results than state-of-the-art counterparts. We believe our forest-based paradigm is a significant step towards the future development of long-distance graph learning.

ACKNOWLEDGMENTS

This work is partly supported by the National Key Research and Development Program of China (No. 2023YFF0725001), the National Natural Science Foundation of China (No. 92370204, 62306255), the Guangdong Basic and Applied Basic Research Foundation (No. 2024A1515011839). the Guangdong Basic and Applied Basic Research Foundation (Grant No.2023B1515120057), the Key-Area Special Project of Guangdong Provincial Ordinary Universities(2024ZDZX1007).

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

## A    DISCUSSIONS

In this section, we will provide extensive discussions on some different aspects of our framework, including some intuitions and motivations as well as theoretical insights.

### A.1    DISCUSSION ON DEGREE IMBALANCE AND THE MERITS OF LONG-DISTANCE KNOWLEDGE

In real-world graphs, *degree/density imbalance* is a common phenomenon, where a small number of nodes have many connections while most nodes have only a few connections, especially in, *e.g.*, social networks, citation networks, and biological graphs. For example, in social media, a few influential users may have thousands of followers, but most of the others have limited neighbors. Degree imbalance biases graph learning algorithms to focus on highly connected nodes and ignore low-degree nodes. More severely, it fundamentally worsens overfitting in graph learning. Low-degree nodes in the training set face a significant issue. Constrained by training objectives, they must fit their labels, but their limited local knowledge may not be enough. With sufficient expressivity, they will use noisy information for fitting to achieve their learning goal and reduce their training losses. Since noise varies among nodes, classification rules learned from noise cannot generalize to unseen nodes, thus causing overfitting issues. On the other hand, for low-degree or low-density nodes in the test set, the limitation of local knowledge poses another challenge. Graph learning models rely on training data to learn classification rules to generalize. However, for those unseen low-degree nodes in the test set, even well-trained models struggle to generalize, since their valuable knowledge on which the generalizable rule is based is insufficient, directly causing misclassification.

Therefore, distant information becomes crucial. By supplementing the scarce local information of low-degree nodes, it helps models understand these nodes better, capture global graph context, and learn general patterns, thus reducing overfitting. In summary, distant knowledge is critical in graph learning, especially under degree or density imbalance. Even when local information seems sufficient, integrating distant knowledge can partly improve performance, which is currently underestimated.

### A.2    DISCUSSION ON AGGREGATING BEHAVIOR COMPARISON AMONG DIFFERENT PARADIGMS

We find that our tree-based paradigm and some other counterparts can be rewritten in a united path-decomposition form with different path weighting strategies, which is shown below:

**Definition 1.** *Define the path-decomposition of a graph learner or aggregator as follows:*

$$H'_v = \text{Agg}\left(\{H_u\}_{u \in V}\right) = \sum_{u \in V} \sum_{\substack{\vec{\mathbf{p}}(u \to v) \in G: \\ p_0 = u \to p_1 \to \cdots \to p_k = v}} H_u \cdot w\left(\vec{\mathbf{p}}\right) \cdot \text{PE}\left(\vec{\mathbf{p}}\right), \tag{12}$$

*where $w\left(\vec{\mathbf{p}}\right) = \prod_{i=1}^{k} w_{p_{i-1} \to p_i}$ with $w_{x \to y}$ is the weight of directed edge $x \to y$ and $\text{PE}\left(\vec{\mathbf{p}}\right)$ is an extra path-based positional encoding beyond vanilla pair-wise relative positional encoding.*

We find that: (1) Infinite-layer deep local SGC, deep local GT, or infinite-step random walk aggregation all have this form, with $w_{x \to y}$ as values in the normalized adjacency matrix/transition matrix/layer-shared attention coefficients, and $\text{PE}\left(\vec{\mathbf{p}}\right) = \prod_{i=0}^{k} \text{PE}\left(p_i\right)$, where $\text{PE}\left(p_i\right)$ is the sum of the discounted edge weight product in all circles of any length. This shows that these methods focus more on *local environmental importance* of a path, *e.g.*, densities or degrees of nodes contained in it. (2) Tree-Set (*i.e.*, Forest) Layer also has this form, with $\text{PE}\left(\vec{\mathbf{p}}\right)$ as the sum of weight products of all spanning trees of the graph obtained by merging path $\vec{\mathbf{p}}$ into a single node. This shows that our paradigm focuses more on *global transport importance* of a path, *i.e.*, how connectivity or communication this path can facilitate if it is built as a highway with no communication cost along it.

We provide the detailed derivations in Sec. B.4 of Appn.

### A.3    DISCUSSION ON OVER-SMOOTHNESS ALLEVIATION OF OUR PARADIGM

In this subsection, we provide a theoretical discussion on the relationship between our graph learning paradigm and the over-smoothing issues. Our analysis can be divided into two parts: (1) Analysis on

the over-smoothness of fixed-distance aggregation; (2) Analysis on the over-smoothness of infinite-distance (*i.e.*, comprehensively global) aggregation. This analysis not only highlights one of the merits of our paradigm, but also provides some novel insights for alleviating over-smoothing issues from the perspective of the aggregating operators themselves as well as their adaptive aggregating scopes.

We first consider the first case, *i.e.*, fixed distance, which is based on similar theoretical evidence as those deep GNNs from, *e.g.*, Rong et al. (2020b) or Chung (1997).

**Lemma 1** (Chung (1997)). *Let $G = (V, E)$ be a connected graph with its diameter $D(G) \geq 4$. Then the second smallest eigenvalue $\lambda_2(G)$ of its normalized Laplacian matrix satisfies:*

$$\lambda_2(G) \leq 1 - 2 \cdot \frac{\sqrt{(\max_{v \in V} d_v) - 1}}{\max_{v \in V} d_v} \left( 1 - \frac{2}{D(G)} \right) + \frac{2}{D(G)}, \tag{13}$$

*where $\lambda_2(G)$ is also known as the spectral gap of the graph $G$.*

This lemma provides an upper bound for the spectral gap for a graph $G$. The next lemma shows how $\lambda_2(G)$ can connect to the over-smoothness.

**Lemma 2** (Chung (1997)). *Let $\mathbf{P}$ be an ergodic random walk transition matrix, where $G$ is connected and non-bipartite, let $\boldsymbol{\pi}$ be its stationary distribution, and let $\mathbf{f}$ be any initial distribution. For any $s \in \mathbb{N}^+$, we have:*

$$\|\mathbf{f}^\top \mathbf{P}^s - \boldsymbol{\pi}\| \leq e^{-s\lambda'} \frac{\max_i \sqrt{d_i}}{\min_j \sqrt{d_j}}, \tag{14}$$

*where $\lambda' = \lambda_2(G)$ if $1 - \lambda_2 \geq \lambda_N(G) - 1$, and $2 - \lambda_N$ otherwise. $\mathbf{P} = \mathbf{D}^{-1}\mathbf{A}$ is the random walk transition matrix. For any initial node distribution $f : \mathcal{V} \to \mathbb{R}$ with $\sum_{v \in \mathcal{V}} f(v) = 1$, the node distribution after $k$ steps is given by $\mathbf{f}^\top \mathbf{P}^k$, where $\mathbf{f} \in \mathbb{R}^{N \times 1}$ is the vector of initial distributions such that $\mathbf{f}(i)$ is the function evaluated on the ith node. The random walk is ergodic when there is a unique stationary distribution $\boldsymbol{\pi}$ satisfying that $\lim_{s \to \infty} \mathbf{f}^\top \mathbf{P}^s = \boldsymbol{\pi}$ Chung (1997).*

*Therefore, we can compute the value of $s$ such that $\|\mathbf{f}^\top \mathbf{P}^s - \boldsymbol{\pi}\| \leq \epsilon$ as follows:*

$$s \geq \frac{1}{\lambda' \log \left( \frac{\max_i \sqrt{d_i}}{\epsilon \min_j \sqrt{d_j}} \right)}, \tag{15}$$

*where we can always add self-loops with weights $d_v$ for node $v \in V$ to make $\lambda' = \lambda_2(G)$.*

**Theorem 3.** *Let $G = (V, E)$ be a connected graph with vertex degrees $\{d_v\}_{v \in V}$, maximum degree $M = \max_{v \in V} d_v$, and diameter $D(G) \geq 4$. For any spanning tree $T = (V, E_T)$ of $G$, denote the degree of vertex $v$ in $T$ as $d_v^{(T)}$, its maximum degree as $M_T = \max_{v \in V} d_v^{(T)}$, and its diameter as $D(T) \geq D(G)$. Let $s(G)$ and $s(T)$ be the number of steps required for node distributions in $G$ and $T$ to be within $\epsilon$ of their stationary distributions, respectively. Based on Lemma 1 and Lemma 2, we have the lower bounds of $s(\cdot)$: (1) For graph $G$:*

$$s(G) \geq \frac{1}{\left( 1 - 2 \cdot \frac{\sqrt{M}-1}{M} \left( 1 - \frac{2}{D(G)} \right) + \frac{2}{D(G)} \right) \log \left( \frac{\max_i \sqrt{d_i}}{\epsilon \min_j \sqrt{d_j}} \right)}, \tag{16}$$

*and (2) for its spanning tree $T$:*

$$s(T) \geq \frac{1}{\left( 1 - 2 \cdot \frac{\sqrt{M_T}-1}{M_T} \left( 1 - \frac{2}{D(T)} \right) + \frac{2}{D(T)} \right) \log \left( \frac{\max_i \sqrt{d_i^{(T)}}}{\epsilon \min_j \sqrt{d_j^{(T)}}} \right)}. \tag{17}$$

*Since spanning tree $T$ satisfies: (1) **Degree constraint**: $d_v^{(T)} \leq d_v$ for all $v \in V$, implying $M_T \leq M$; (2) **Diameter extension**: $D(T) \geq D(G) \geq 4$, the monotonicity analysis of the bounds of $s$ with respect to $M$ yields: $T$ has a tighter (larger) lower bound of $s(T)$ against that of $G$. Since a larger $s$ indicates more steps are needed before distributions approach the stationary node distribution of the Markov Chain. Thus, for fixed-distance aggregation, the spanning tree structure $T$ can alleviate over-smoothing issues.*

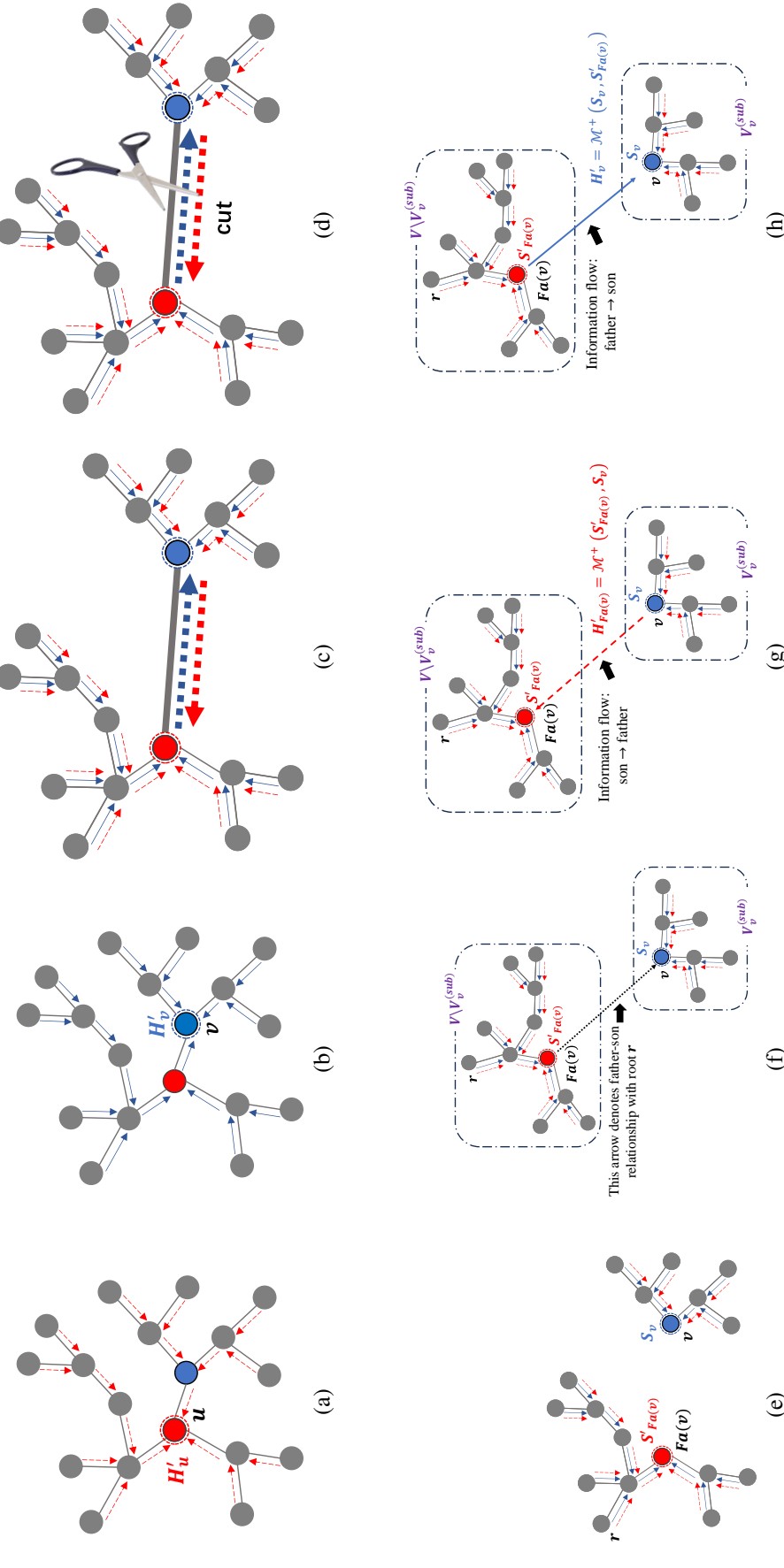

Figure 7: The detailed illustration of the recursion formulas (Eq. 30 and Eq. 31) described in Sec. B.1 of the main text. We demonstrate the derivation of these two relationships via several steps. Given a tree $T$, consider a pair of neighboring key nodes $u$ and $v$, which are colored *red* and *blue*, respectively. **(a)** With the *red* node $u$ as the root of $T$, all other nodes send global messages to $u$ (colored *red dashed line*), and we can clearly observe the information flow direction of each edge. Denote the expected total global messages of node $u$ with $H'_u$. **(b)** With the *blue* node $v$ as the root of $T$, all other nodes send global messages to $v$ (colored *blue solid line*), and we can clearly observe the information flow direction of each edge. Denote the expected total global messages of node $v$ with $H'_v$. **(c)** We put (a) and (b) together. From the comparison, we can clearly see that the only difference between these two distinct information flows is *the direction of a single key edge* (*i.e.*, the edge $u \leftrightarrow v$). In other words, the directions of all other edges remain consistent. **(d)** Then, we cut the key edge and observe the remaining message aggregation processes. **(e)** With another node $r \in V$ as the root of tree $T$, node $u$ will become the father of node $v$. After cutting the key edge $u \leftrightarrow v$, the remaining global messages flowing into node $u = \mathrm{Fa}(v)$ is denoted as $S'_{\mathrm{Fa}(v)}$ (red) while the remaining global messages flowing into node $v$ is denoted as $S_v$ (blue). **(f)** This sub-figure more clearly shows the father-son relationship between node $u$ and node $v$. The black dashed line explicitly highlights this relationship. Note that the black line does not represent any information flow or its direction. **(g)** If we add back the key edge with son → father (*i.e.*, $v \to \mathrm{fa}(v) = u$), we can restore the total global messages into node $u = \mathrm{Fa}(v)$, *i.e.*, $H'_{\mathrm{Fa}(v)}$, by merging the information from $v$ (*i.e.*, blue $S_v$) into that of $\mathrm{Fa}(v)$ (*i.e.*, red $S'_{\mathrm{Fa}(v)}$), which derives one formula (Eq. 31). Note that the long red dashed line $v \to u$ denotes the direction of the information flow. **(h)** If we add back the key edge with father → son (*i.e.*, $\mathrm{fa}(v) = u \to v$), we can restore the total global messages into node $v$, *i.e.*, $H'_v$, by merging the information from $u = \mathrm{Fa}(v)$ (*i.e.*, red $S'_{\mathrm{Fa}(v)}$) into that of $v$ (*i.e.*, blue $S_v$), which derives another formula (Eq. 30). Note that the long blue solid line $u \to v$ denotes the direction of the information flow. These figures detailedly show the derivations of two formulas, both of which utilize the operator $\mathcal{M}^+$ $(\cdot, \cdot)$ but with different or opposite merging directions.

The proof can be found in Sec. B.3 of Appn.

We then consider the second case, *i.e.*, infinite distance. Recall Eq. 30 and Eq. 31 provided in Sec. B.1 and the illustration on their derivations in Fig. 7:

$$H'_v = \mathcal{M}^+ \left( S_v, \ S'_{\mathrm{Fa}(v)} \right), \tag{18}$$

$$H'_{\mathrm{Fa}(v)} = \mathcal{M}^+ \left( S'_{\mathrm{Fa}(v)}, \ S_v \right), \tag{19}$$

where $\mathcal{M}^+ (\cdot, \cdot)$ is a directional merging operator, merging the right term into the left term. Note that the term $S_v$ and $S'_{\mathrm{Fa}(v)}$ denoting knowledge from different parts in the graph $G$ (the sub-tree of node $v$ and its complement, Fig. 7), which means that there is nearly no over-lapped (or intersected) information among them. Therefore, it is reasonable to suppose that: $\|S'_{\mathrm{Fa}(v)} - S_v\|_2$ is not small, which gives:

$$\|H'_{\mathrm{Fa}(v)} - H'_v\|_2 = \left\| \mathcal{M}^+ \left( S_v, \ S'_{\mathrm{Fa}(v)} \right) - \mathcal{M}^+ \left( S'_{\mathrm{Fa}(v)}, \ S_v \right) \right\|_2. \tag{20}$$

This equation implies that the difference between the node embeddings of a pair of neighboring nodes can be bounded by the extent of asymmetry. Since the $\mathcal{M}^+ (\cdot, \cdot)$ can be specifically designed by practitioners, we have the opportunity to directly alleviate the over-smoothness. For the simplest example, we set $\mathcal{M}^+ (a, \ b) = 3a + b$, and thus we obtain:

$$\|H'_{\mathrm{Fa}(v)} - H'_v\|_2 = \left\| \left( 3S_v + \ S'_{\mathrm{Fa}(v)} \right) - \left( 3S'_{\mathrm{Fa}(v)} + \ S_v \right) \right\|_2 \tag{21}$$

$$= 2 \cdot \left\| S'_{\mathrm{Fa}(v)} - \ S_v \right\|_2. \tag{22}$$

Since we assume that $\|S'_{\mathrm{Fa}(v)} - S_v\|_2$ is not small (*i.e.*, the two terms inside it have non-trivial differences because they aggregate information of completely different sets of nodes, illustrated in Fig. 7), the over-smoothness can be naturally controlled. Yet, for traditional deep GNNs with very deep layers, the differences between $H'_u$ and $H'_v$ are bounded by embeddings from the last layer, which already have too many similarities due to overlapping or even the same global scope, intuitively improving the risk of over-smoothness.

### A.4 DISCUSSION ON PROPAGATION BOTTLENECK ALLEVIATION OF OUR PARADIGM

In this subsection, we briefly discuss how our framework alleviates the propagation bottleneck. When the knowledge is propagated along the edge of a single tree, it would be discounted with distance or even blocked when the distance becomes too long. To address this issue, we integrate several trees ($N_T$ trees), *i.e.*, a forest, rather than a single tree. Furthermore, we can additionally integrate a simple local shallow GNN (*i.e.*, several local layers) before and/or after the proposed global layer (*i.e.*, Forest Layer in Fig. 2).

Suppose we have $N_T$ trees and add $K_L/2$ local sub-layers before and after our global layer, respectively. One message from node $v$ to node $u$ has only a single path in a tree with the distance $\mathrm{dist}^{(T)} (v, \ u)$. After improving, the number of candidate paths becomes $K \cdot \bar{d}^{2K_L}$, where $\bar{d}$ is assumed to be the average degree of all nodes on a graph. Furthermore, the distance can be significantly shortened as follows:

$$\mathrm{dist}^{(\{T_k\}_{k \in [1, \ K]})} (v, \ u) \tag{23}$$

$$= \min_{\substack{v': \ \mathrm{dist}^{(G)}(v,v') \leq K_L/2 \\ u': \ \mathrm{dist}^{(G)}(u,u') \leq K_L/2}} \left( \mathrm{dist}^{(G)}(v, v') + \mathrm{dist}^{(G)}(u, u') + \min_{k \in [1, \ K]} \mathrm{dist}^{(T_k)}(u', v') \right), \tag{24}$$

which means that the message from $v$ to $u$ can first select a $K_L/2$-order neighbor on the vanilla graph, then be quickly sent to another neighbor $u'$ of $u$, and finally be propagated from $u'$ to $u$. The overall path is: $v \to v' \to u' \to u$, which mimics the real-world transport strategies, where people walk into a highway system and walk outside it.

The distance $\mathrm{dist}^{(\{T_k\}_{k \in [1, \ K]})}$ can be shortened to the minimum value, *i.e.*, the length of their shortest path in the vanilla graph, *i.e.*, $\mathrm{dist}^{(G)} (u, v)$, since:

$$\mathrm{dist}^{(\{T_k\}_{k \in [1, \ K]})} (v, \ u) \geq \mathrm{dist}^{(G)}(v, v') + \mathrm{dist}^{(G)}(u, u') + \mathrm{dist}^{(G)}(u', v') \geq \mathrm{dist}^{(G)}(v, u), \tag{25}$$

where $u'$, $v'$ are the optimal nodes of the last equation. When improving the value of $K_L$, the above distance tends to approach the optimal value, if $K_L \geq D(G)$ and letting $u' = v'$. This case deteriorates into a traditional deep GNN, propagating knowledge only via stacking local layers. Thus, from the perspective of propagating distance, our framework can be viewed as an interpolation between deep GNNs and shallow global counterparts. Moreover, with some graph augmentation tricks, *e.g.*, Expander Shirzad et al. (2023) or a prediction-based one (Sec. L), the shortest path length on the vanilla graph can be further reduced, which means the distance can be adjusted into appropriate values between a large interval. Therefore, we can find a good $K_L$ and $N_T$ to adjust those distances to the most appropriate values. Note that too large $K_L$ and $N_T$ can nearly address the propagation bottleneck, but with extra computational overhead. Also, it would make over-smoothness severe due to their essential trade-off reported in Giraldo et al. (2023). Thus, we keep $K_L \leq 2$ and $N_T \leq 15$ ($N_T \leq 6$ for larger graphs), and empirically obtain competitive performance.

## A.5 Discussion on the Motivation of Our Attention-based Estimator

In this subsection, we will detail our motivation for the design of an attention-based edge weight or homophily estimator, as well as a discussion on its impact on future attention exploitation, which will essentially provide an explanation of our performance gain from a very high-level perspective, even though we utilize the most basic auxiliary models to guide the training of local attention learning.

We expect to sample spanning trees from a tree distribution defined via scores defined based on a homophily estimator. In the node classification tasks, the quality of a homophily estimator is assumed to be positively proportional to its performance (despite the fact that it is not the only factor related to the performance). Therefore, we hope the auxiliary models to have better performance. Attention coefficients tend to measure the utility of the information of one object on the learning of another one, which is intuitively a good implementation of the above-mentioned homophily estimator. However, under label scarcity in the semi-supervised settings, attention-based models risk over-fitting issues due to their strong expressivity, which would deteriorate their performance as well as their quality of attention. To address this issue, before learning effective attention coefficients, we expect to adjust or polish the way of pre-training the attention-based auxiliary models, *i.e.*, $\mathcal{T}_1$. Yet, how to better pre-train this auxiliary model $\mathcal{T}_1$ is quite a big problem, which may be significantly beyond our work. But we can consider this problem at a high level and utilize a simple trick: use another pre-training process before the training of $\mathcal{T}_1$ and inject some valuable knowledge $\mathbb{K}_0$ into the latter. The extra knowledge $\mathbb{K}_0$ can be viewed as some extra guidance, hints, or rules (*e.g.*, the pseudo-labels or predictions from $\mathcal{T}_0$). The attention learning will be generalized as a conditional training, *i.e.*, with $\mathbb{K}_0$, we extract new knowledge $\mathbb{K}_1$ from $\mathcal{T}_1$. We trust the ability of the model $\mathcal{T}_1$, but sometimes it may suffer from some issues due to some reasons. The outputs $\mathbb{K}_0$ have the potential to stabilize its training. Also, $\mathbb{K}_0$ can be extracted from another model $\mathcal{T}_0$. For our case, the $\mathbb{K}_0$ can be simply set to the predictions, *i.e.*, the node label predictions from the previous auxiliary model $\mathcal{T}_0$. The intuition is that this $\mathbb{K}_0$ can supplement the supervision, which has effects on alleviating over-fitting issues. As evidenced in our estimator comparison experiments (*i.e.*, Tab. 4), the introduction of $\mathbb{K}_0$ can effectively improve the performance of the attention-based model $\mathcal{T}_1$ in most cases.

## A.6 Discussion on the Generality of the Message Aggregator

Despite the restrictions posed on $f_{\text{Agg}}(\cdot)$ due to the discovered properties, we show $f_{\text{Agg}}(\cdot)$ can still be quite general. To see this, recall the fact that many aggregators are designed with a weighted sum of transformed embeddings (including attention) followed by a simple element-wise activation. If the activation is an identity map, then the linearity perfectly admits the properties (Eq. 4). Besides, if it is non-linear and invertible, we can first invert $S_v$ and $H'_{\text{Fa}(v)}$ easily in Eq. 34 (the first arguments in $\mathcal{M}^+$ and $\mathcal{M}^-$), and then repose non-linearity after $\mathcal{M}^+$. Moreover, for those non-invertible non-linear activation functions $\sigma(\cdot)$, we can use a trick, *i.e.*, only storing before-activation values into $S_v^{[b]}$ and $H'^{[b]}_{\text{Fa}(v)}$ to avoid invertibility and then reposing non-linearity as follows:

$$\forall\, u \in V, \quad S_u^{[b]} = f_{\text{Agg}}^{[b]} \left( \left\{ \sigma\left(S_v^{[b]}\right) \right\}_{v \in \text{Child}(u)} \cup \{g(H_u)\} \right), \quad \textbf{Recursion (I)}' \qquad (26)$$

$$\forall\, v \in V, \quad H'^{[b]}_v = \mathcal{M}^+ \left( S_v^{[b]}, \sigma\left( \mathcal{M}^- \left( H'^{[b]}_{\text{Fa}(v)}, \sigma\left(S_v^{[b]}\right) \right) \right) \right), \quad \textbf{Recursion (II)}' \qquad (27)$$

$$\forall\, v \in V, \quad H'_v = \sigma\left(H'^{[b]}_v\right), \quad S_v = \sigma\left(S'^{[b]}_v\right), \quad f^{[b]}_{\mathrm{Agg}}\left(\cdot\right) = \sigma\left(f_{\mathrm{Agg}}\left(\cdot\right)\right). \tag{28}$$

This insight allows almost all famous first-order local aggregators $f_{\mathrm{Agg}}\left(\cdot\right)$, including many local attention-based GNNs and typical RNNs, even beyond linearity.

### A.7 Discussion on the insights of Expressivity

Recall that the expressive power of GNNs typically refers to their ability to discriminate whether two given graphs are isomorphic or not. Thus, the first natural question is whether the proposed framework can successfully identify two given isomorphic graphs. Here, we consider two graphs, $G_1 = (V_1, E_1)$ and $G_2 = (V_2, E_2)$, with exactly the same number of nodes and edges (i.e., $|V_1| = |V_2|$ and $|E_1| = |E_2|$). Provided that they are isomorphic, there must exist a bijective function $\varphi : V_1 \to V_2$ such that $\mathbb{I}\{(u, v) \in E_1\} = \mathbb{I}\{(\varphi(u), \varphi(v)) \in E_2\}$ for any $u, v \in V_1$, where $\mathbb{I}\{\cdot\}$ is the indicator function. In other words, these two graphs, $G_1$ and $G_2$, look exactly the same, up to a node relabeling. Assume that our paradigm encodes a graph $G = (V, E)$ into an embedding $H(G) \in \mathbb{R}^d$, where $H(G) = \mathrm{Pool}\left(\{H_v(G)\}_{v \in V}\right)$ and $H_v(G) = \mathbb{E}_{T \sim P_T(G)}\left[f^T_{\mathrm{Agg}}(v)\right]$ with $f^T_{\mathrm{Agg}}(v) \in \mathbb{R}^d$ denoting the global message obtained via node $v$ by the tree $T$. Therefore, we can successfully identify that these two graphs $G_1$ and $G_2$ are isomorphic, since $H(G_1) = H(G_2)$. To see this, considering any spanning tree $T_1 = (V, E_{T_1}) \subseteq G_1$, there must exist a corresponding tree $T_2 \subseteq G_2$, where $T_2 = \varphi(T_1) = (V, \varphi(E_{T_1}))$, such that $T_1$ and $T_2$ are isomorphic and the position of node $v$ in $T_1$ is symmetrically the same as that of node $\varphi(v)$ in $T_2$. Thus, we have $f^{T_1}_{\mathrm{Agg}}(v) = f^{T_2}_{\mathrm{Agg}}(\varphi(v))$, and consequently we can obtain that for any node $v \in V$, $H_v(G_1) = H_{\varphi(v)}(G_2)$, due to the fact that $\mathbb{E}_{T_1 \sim P_T(G_1)}\left[f^{T_1}_{\mathrm{Agg}}(v)\right] = \mathbb{E}_{T_2 \sim P_T(G_2)}\left[f^{T_2}_{\mathrm{Agg}}(\varphi(v))\right]$. This concludes the proof of $H(G_1) = H(G_2)$, showing the ability of the proposed framework to identify isomorphic graphs.

Besides, we can further provide some insights into the ability to identify non-isomorphic graphs. We present two example graphs, $G_1$ and $G_2$, that are not isomorphic and can be successfully identified by our paradigm, yet fail to be identified by typical GNNs. One case is that: let $G_1$ be a six-node circle and $G_2$ be two three-node circles with all node features assigned scalars 1s (we call this case **Case A**). For typical GNNs, including GIN, their expressive/discriminative powers are restricted within 1-WL, which is a theoretical framework for encoding graphs based on iterative local aggregations and hashing. In this case, by mathematical induction, each node in each graph has intrinsically the same color set of first-order neighbors (e.g., $\{1, 1\}$), and thus they will be colored exactly the same after hashing at each iteration in the 1-WL test algorithm (e.g., all nodes are colored with 1), which implies that both of $G_1$ and $G_2$ obtains the same graph encoding, i.e., the multi-set $\{1, 1, \cdots, 1\}$, and cannot be distinguished. More high-levelly, since 1-WL gains global power by local stacking, it works well for local structures, but may fail at some simple global structural recognitions. We know that graphs can be viewed as approximated discrete manifolds. From the perspective of manifolds, local structures do not directly imply the global topology, and how such local structures are organized still matters. That is why 1-WL techniques fail to deal with Case A. In contrast, Graph Transformers (GTs) directly conduct global aggregations and have the potential to address this case. Yet, without sophisticated positional or structural encodings (PEs or SEs), GTs fail to capture such long-distance topological knowledge (e.g., connectivity, connected components, or communities). It would produce node embeddings 1s for all nodes (i.e., $H(G_1) = H(G_2) = \mathrm{Pool}(\{1, 1, \cdots, 1\})$ for each graph), and thus still fail to discriminate graphs in case A. Therefore, it necessitates a complex PE for GTs, such as Laplacian-based encodings, which, nevertheless, would result in a higher complexity to achieve such a stronger power. But in our paradigm, taking any spanning tree (with edges weights 1) for each connected component of the graphs, we can efficiently obtain $H(G_1) = \mathrm{Pool}(\{6, 6, \cdots, 6\})$ and $H(G_2) = \mathrm{Pool}(\{3, 3, \cdots, 3\})$, which easily distinguishes these two graphs $G_1$ and $G_2$ without the necessity of any hand-craft high-complexity PEs or SEs. This case can be generalized to any two non-isomorphic $k$-regular graphs $G_1$ and $G_2$ (**Case B**), where each node has the same degree $k$. Similarly, by mathematical induction, 1-WL will still produce the same encoding for both $G_1$ and $G_2$, and thus fails to discriminate them. And GTs still require sophisticated PEs/SEs to encode subtle structural differences. But our FGL can implicitly encode such differences into edge probabilities in a tree (i.e., $p(e)$, denoting how likely an edge $e$ would appear in a spanning tree), and then affect the probabilities of propagating paths. The sensitive probabilistic differences will naturally differentiate the passed node messages and make final graph encodings distinct. This insight shows the potential of our FGL to surpass the expressive power of typical GNNs.

### A.8 DISCUSSION ON DEALING WITH DENSE GRAPH

It is not practical for us to utilize only several trees (e.g., $K$ trees) to comprehensively capture all topological information of a densely connected graph, such as a graph with $n$ nodes and $O(n^2)$ edges. In other words, it would unavoidably cause some information loss in this situation. The intuition is that it seems not enough for these trees to well represent first-order neighborhoods, since $K$ trees cover only $K \cdot n$ degrees ($K \ll n$), but the total node degree positively correlates to the number of edges, i.e., $O(n^2)$ edges. However, we can straightforwardly mitigate the information loss by increasing the number of trees, $K$, with extra complexities no more than the number of edges $O(n^2)$. The reason is that in this situation, the graph itself would become the essential bottleneck of complexities, and thus, there are no efficient ways to process it without any information loss. Moreover, we empirically find that adding only a few trees may be sufficient to cover the main information. That is to say, the number of trees utilized for a dense graph can be sublinear in the number of nodes, $n$, with a very limited extra computational burden compared to the graph itself. To show this clearly, we introduce a dense graph and empirically test the relationship between the number of trees and the final performance. We construct the graph by adding many edges to the Cora dataset (100 extra edges for each node), while maintaining the vanilla edge homophily rate $p$ (adding a homophilous edge with probability $p$ and a heterophilous edge with probability $1 - p$). The results are shown in Sec. J.6. We observe that the addition of extra trees can further improve performance compared to using only a few. However, introducing too many trees cannot improve performance and may even slightly degrade it, as they introduce redundancy and would increase the risk of overfitting or over-smoothing issues. Therefore, even dealing with a very densely connected graph, a limited number of trees would be enough to encode the essential structural knowledge, without the need to introduce too many trees.

### A.9 DISCUSSION ON A CASE WHERE OUR PARADIGM MIGHT UNDERPERFORM

Here, we supplement a discussion with a case where our paradigm might underperform. The case would be a highly disconnected graph with too sparse edges. This graph has too many connected components ($O(n)$ components), and each of them is very small in size ($O(1)$ nodes per component). In this graph, our paradigm might fail to extract valuable long-distance knowledge and thereby degrade the final performance, possibly due to its heavy dependence on the pre-processing stage to address the high dis-connectivity. Yet, notably, deep GNNs still have this limitation, while GTs might have some merits in avoiding this issue.

## B PROOFS AND DERIVATIONS

### B.1 PROOF FOR THEOREM 1

Recall the design of our tree aggregator $f_{\text{Agg}}^{(T)}$, it is designed based on a general message aggregator $f_{\text{Agg}}(\cdot)$. Any $f_{\text{Agg}}(\cdot)$ can be applied here if it satisfies the following two sufficient properties:

$$\forall\, S, A, B,\ s.t.,\ S = B \cup A,\ B \cap A = \emptyset:$$
$$f_{\text{Agg}}(S) = \mathcal{M}^+\left(f_{\text{Agg}}(B),\, f_{\text{Agg}}(A)\right),\quad \textbf{Property (I)} \tag{29}$$
$$f_{\text{Agg}}(B) = f_{\text{Agg}}(S \setminus A) = \mathcal{M}^-\left(f_{\text{Agg}}(S),\, f_{\text{Agg}}(A)\right),\quad \textbf{Property (II)}$$

where $A, B, S$ are sets of messages (*e.g.*, node embeddings).

Two merging operators $\mathcal{M}^+\left(\vec{a},\, \vec{b}\right)$ and $\mathcal{M}^-\left(\vec{a},\, \vec{b}\right)$ denote adding (or deleting) vector $\vec{b}$ to (or from) vector $\vec{a}$. Note that the $\mathcal{M}^{+/-}(\cdot, \cdot)$ can utilize auxiliary information and thus are allowed to be unsymmetrical.

Then, we provide the proof of Theorem 1 as follows:

*Proof.* Let $f_{\text{Agg}}^{(T)} : H \mapsto H' \in \mathbb{R}^{n \times d}$ denote our general tree aggregator, where $H, H' \in \mathbb{R}^{n \times d}$ denote the node embeddings before and after aggregation. Let node $r \in V$ is the root node of tree $T$.

First, we prove Recursion (I). Consider a node $u \in V$: the notation $S_u$ denotes the combination of all messages from the subtree $T_u^{(sub)}$ (nodes $V_v^{(\text{sub})}$). Note that all such messages will either pass

through one of the children nodes $\text{Child}(u)$ (denoted as $v$) or be generated from $u$ itself (*i.e.*, $g(H_u)$). Provided that node $u$ has $K_u$ children on tree $T$, we can classify the aforementioned messages into $K_u + 1$ categories. At each one of the first $K_u$ categories (aussming passing through the child $v$), messages are merged into $S_v$. Merging these $K_u + 1$ catergories, *i.e.*, $\{S_v\}_{v \in \text{Child}(u)} \cup \{g(H_u)\}$, which naturally derives Recursion (I), concluding the proof. Therefore, one can easily pre-process all $S_v$ for all $v \in V$ with the simple recursion, *i.e.*, Recursion (I). Then, we will assume that all $S_v$ are known.

Second, we prove Recursion (II). Recalling the observation we found in Sec. 4.3 and the intuitive visualization in Fig. 7, we can obtain: the globally merged message at node $v$ (*i.e.*, $H'_v$) and that at its father node $\text{Fa}(v)$ (*i.e.*, $H'_{\text{Fa}(v)}$) differ only at the direction of one edge $e = (v, \text{Fa}(v))$. In other words, the messages arriving at node $v$ partially pass the edge $\text{Fa}(v) \rightarrow v$, and those at node $\text{Fa}(v)$ partially pass $v \rightarrow \text{Fa}(v)$. Thus, if we delete the edge $e$, the left messages at nodes $v$ and $\text{Fa}(v)$ (denoted as $S_v$ and $S'_{\text{Fa}(v)}$) are from a subtree $T_v^{(\text{sub})}$ (*i.e.*, nodes $V_v^{(\text{sub})}$) and from its complement set (*i.e.*, nodes $V \backslash V_v^{(\text{sub})}$), respectively. It means that $H'_v$ and $H'_{\text{Fa}(v)}$ are formed by exactly the same two parts with different merging directions, which can be formulated as follows:

$$H'_v = \mathcal{M}^+ \left( S_v, \ S'_{\text{Fa}(v)} \right), \tag{30}$$

$$H'_{\text{Fa}(v)} = \mathcal{M}^+ \left( S'_{\text{Fa}(v)}, \ S_v \right). \tag{31}$$

The above insight allows us to derive an recursion that directly connects $H'_v$ and $H'_{\text{Fa}(v)}$. Provided with $S_v$, according to Properties (I) and (II), we can first inverse Eq. 31 easily to calculate $S'_{\text{Fa}(v)}$ from $H'_{\text{Fa}(v)}$ by taking $\mathcal{M}^- (\cdot, \ S_v)$ at both of its sides:

$$S'_{\text{Fa}(v)} = \mathcal{M}^- \left( \mathcal{M}^+ \left( S'_{\text{Fa}(v)}, \ S_v \right), \ S_v \right) = \mathcal{M}^- \left( H'_{\text{Fa}(v)}, \ S_v \right). \tag{32}$$

Note that we can combine Properties (I) and (II), which implies that:

$$\mathcal{M}^- \left( \mathcal{M}^+ (p, \ q), \ q \right) = p. \tag{33}$$

In other words, if we first add $q$ into $p$ (obtaining $p + q$) and then subtract $q$ from $p + q$, then we will obtain $p$ itself, showing the invertability of $\mathcal{M}^+$ and $\mathcal{M}^-$.

Then, injecting Eq. 32 into Eq. 30 provides us with Recursion (II), *i.e.*, Eq. 34.

$$\forall \, v \in V, \quad H'_v = \mathcal{M}^+ \left( S_v, \ \mathcal{M}^- \left( H'_{\text{Fa}(v)}, S_v \right) \right). \qquad \textbf{Recursion (II)} \tag{34}$$

Putting these two parts together concludes the whole proof of this theorem. $\qquad \square$

## B.2 Proof for Theorem 2

Recall the theorem we present in Sec. 4.6 of the main text, which justifies the quality of the specifically designed tree distribution, *i.e.*, trees sampled from it tend to preserve the first-order homophily ratios. Next, we provide its proof.

**Theorem 2\*.** *Let $\widehat{G}$ be any connected graph, and define the expected edge homophily ratio under the score ratio $\Delta = p/q > 0$ as:*

$$R_{\widehat{G}}(\Delta) := \mathbb{E}_{T \sim P_{\widehat{G}}^{(p,q)}} \left[ h(T) \right],$$

*where $h(T)$ is the edge homophily ratio of tree $T$. Then there exists a $\Delta_0 > 0$ such that:*

- **Monotonicity.** *If $\Delta > \Delta' \geq \Delta_0$, then*

$$R_{\widehat{G}}(\Delta) > R_{\widehat{G}}(\Delta').$$

- **Upper Bound.** *For all $\Delta \geq \Delta_0$,*

$$R_{\widehat{G}}(\Delta) \ \leq \ 1 - \frac{\text{NHCC}(\widehat{G}) - 1}{n - 1},$$

*where $\text{NHCC}(\widehat{G})$ denotes the number of homophilous connected components of $\widehat{G}$.*

- **Asymptotic Tightness.** *As* $\Delta \to +\infty$,

$$R_{\widehat{G}}(\Delta) \to 1 - \frac{\text{NHCC}(\widehat{G}) - 1}{n - 1}.$$

*Proof.* Based on an introduced homophily indicator function $\mathbb{I}^{\text{homo}}(\cdot)$ denoting wether an edge is homophilous, we can define the first-order homophily ratio of a tree $T$ of size $n$ (assuming it is sampled from a graph $\widehat{G} = (V, E)$ of $n$ nodes and $m$ edges), *i.e.*, $h(T)$, as follows:

$$h(T) = \frac{1}{n-1} \cdot \sum_{e \in T} \mathbb{I}^{\text{homo}}(e), \tag{35}$$

where $\mathbb{I}^{\text{homo}}(e = (u \leftrightarrow v)) = 1$ if node labels $Y_u = Y_v$, $\mathbb{I}^{\text{homo}}(e = (u \leftrightarrow v)) = 0$ otherwise. Denote as $\mathbb{T}(G)$ the space/set of all spanning trees of the graph $G$, where we suppose $G$ is connected. For convenience, we also define $n^+(T), n^-(T) \in \mathbb{N}$ as the numbers of edges in the tree $T$ that are homophilous and heterophilous, respectively. Therefore, based on their definitions, we immediately have:

$$\forall T \in \mathbb{T}\left(\widehat{G}\right), \quad n^+(T) + n^-(T) = n - 1, \tag{36}$$

$$n^+(T) = \sum_{e \in T} \mathbb{I}(e). \tag{37}$$

Recall the definition of the designed tree distribution $P_{\widehat{G}}^{(p,\,q)}(T)$ $(0 < q \le p \le 1)$ conditioned on the graph $\widehat{G}$:

$$\forall T \in \mathbb{T}\left(\widehat{G}\right), \quad P_{\widehat{G}}^{(p,\,q)}(T) = \frac{\prod_{e \in T} p^{\mathbb{I}^{(\text{homo})}(e)} \cdot q^{1 - \mathbb{I}^{(\text{homo})}(e)}}{\sum_{T' \in \mathbb{T}(\widehat{G})} \prod_{e \in T'} p^{\mathbb{I}^{(\text{homo})}(e)} \cdot q^{1 - \mathbb{I}^{(\text{homo})}(e)}}, \tag{38}$$

where $p > 0$ is the score we assign for the homophilous edges, $q > 0$ is the score for the heterophilous edges, and we assume $q < p$. Then we can analyze the expected homophily ratio as follows:

$$\mathbb{E}_{T \sim P_{\widehat{G}}^{(p,\,q)}(T)}[h(T)] = \sum_{T \in \mathbb{T}(\widehat{G})} P_{\widehat{G}}^{(p,\,q)}(T) \cdot h(T) \tag{39}$$

$$= \sum_{T \in \mathbb{T}(\widehat{G})} P_{\widehat{G}}^{(p,\,q)}(T) \cdot \frac{n^+(T)}{n - 1} \tag{40}$$

$$= \frac{1}{n-1} \cdot \sum_{T \in \mathbb{T}(\widehat{G})} P_{\widehat{G}}^{(p,\,q)}(T) \cdot n^+(T) \tag{41}$$

$$= \frac{1}{n-1} \cdot \sum_{T \in \mathbb{T}(\widehat{G})} \frac{\prod_{e \in T} p^{\mathbb{I}^{(\text{homo})}(e)} \cdot q^{1 - \mathbb{I}^{(\text{homo})}(e)}}{\sum_{T' \in \mathbb{T}(\widehat{G})} \prod_{e \in T'} p^{\mathbb{I}^{(\text{homo})}(e)} \cdot q^{1 - \mathbb{I}^{(\text{homo})}(e)}} \cdot n^+(T) \tag{42}$$

$$= \frac{1}{n-1} \cdot \frac{\sum_{T \in \mathbb{T}(\widehat{G})} \prod_{e \in T} p^{\mathbb{I}^{(\text{homo})}(e)} \cdot q^{1 - \mathbb{I}^{(\text{homo})}(e)} \cdot n^+(T)}{\sum_{T' \in \mathbb{T}(\widehat{G})} \prod_{e \in T'} p^{\mathbb{I}^{(\text{homo})}(e)} \cdot q^{1 - \mathbb{I}^{(\text{homo})}(e)}}. \tag{43}$$

Due to the fact that:

$$\prod_{e \in T} p^{\mathbb{I}^{(\text{homo})}(e)} \cdot q^{1 - \mathbb{I}^{(\text{homo})}} = p^{n^+(T)} \cdot q^{n^-(T)}. \tag{44}$$

Injecting Eq. 44 into Eq. 43, we can obtain:

$$\mathbb{E}_{T \sim P_{\widehat{G}}^{(p,\,q)}(T)}[h(T)] = \frac{1}{n-1} \cdot \frac{\sum_{T \in \mathbb{T}(\widehat{G})} \cdot p^{n^+(T)} \cdot q^{n^-(T)} \cdot n^+(T)}{\sum_{T' \in \mathbb{T}(\widehat{G})} p^{n^+(T')} \cdot q^{n^-(T')}}. \tag{45}$$

According to Eq. 36, we have:

$$n^- (T) = n - 1 - n^+ (T).$$ (46)

Injecting this into Eq. 45, we can obtain the following formula:

$$\mathbb{E}_{T \sim P_{\widehat{G}}^{(p, q)}(T)} [h (T)] = \frac{1}{n - 1} \cdot \frac{\sum_{T \in \mathbb{T}(\widehat{G})} \cdot p^{n^+ (T)} \cdot q^{n-1-n^+ (T)} \cdot n^+ (T)}{\sum_{T' \in \mathbb{T}(\widehat{G})} p^{n^+ (T')} \cdot q^{n-1-n^+ (T')}}.$$ (47)

We can cancel out the term $q^{n-1}$ and then get:

$$\mathbb{E}_{T \sim P_{\widehat{G}}^{(p, q)}(T)} [h (T)] = \frac{1}{n - 1} \cdot \frac{\sum_{T \in \mathbb{T}(\widehat{G})} \cdot p^{n^+ (T)} \cdot q^{-n^+ (T)} \cdot n^+ (T)}{\sum_{T' \in \mathbb{T}(\widehat{G})} p^{n^+ (T')} \cdot q^{-n^+ (T')}}$$ (48)

$$= \frac{1}{n - 1} \cdot \frac{\sum_{T \in \mathbb{T}(\widehat{G})} \cdot \left(\frac{p}{q}\right)^{n^+ (T)} \cdot n^+ (T)}{\sum_{T' \in \mathbb{T}(\widehat{G})} \left(\frac{p}{q}\right)^{n^+ (T')}}.$$ (49)

Let $\lambda = \frac{p}{q} > 1$, then we have:

$$\mathbb{E}_{T \sim P_{\widehat{G}}^{(p, q)}(T)} [h (T)] = \frac{1}{n - 1} \cdot \frac{\sum_{T \in \mathbb{T}(\widehat{G})} \cdot \lambda^{n^+ (T)} \cdot n^+ (T)}{\sum_{T' \in \mathbb{T}(\widehat{G})} \lambda^{n^+ (T')}}.$$ (50)

Observing this formula, we find that we can treat both the numerator and the denominator as polynomials of the variable $\lambda > 1$. Moreover, they have different coefficients for different terms: all the terms of the denominator have exactly the same coefficient, *i.e.*, 1, while the term $\lambda^{n^+ (T)}$ of the numerator has the coefficient $n^+ (T)$, which hints at us to disentangle the constant.

Denote the maximum value of $n^+ (T)$ as $n_{\max}^+$ and the number of $n^+ (T)$ among all trees $T$ in $\mathbb{T}\left(\widehat{G}\right)$ as $N (n^+ (T))$, formally:

$$N (n_0) = \# \left\{ T : T \in \mathbb{T}\left(\widehat{G}\right), \ n^+ (T) = n_0 \right\},$$ (51)

$$n_{\max}^+ = \max_{T \in \mathbb{T}(\widehat{G})} n^+ (T).$$ (52)

Based on these notations, we can reformulate Eq. 50 as follows:

$$\mathbb{E}_{T \sim P_{\widehat{G}}^{(p, q)}(T)} [h (T)] = \frac{1}{n - 1} \cdot \frac{\sum_{k \geq 0} \lambda^k \cdot k \cdot N (k)}{\sum_{k \geq 0} \lambda^k \cdot N (k)}.$$ (53)

For $N(k) > 0$, $k_{\max} = N_{\max}^+$. We can disentangle the constant $\frac{k_{\max} \cdot N(k_{\max})}{N(k_{\max})} = k_{\max} = N_{\max}^+$ from the above equations:

$$\mathbb{E}_{T \sim P_{\widehat{G}}^{(p, q)}(T)} [h (T)] = \frac{1}{n - 1} \cdot \left( N_{\max}^+ - N_{\max}^+ + \frac{\sum_{k \geq 0} \lambda^k \cdot k \cdot N (k)}{\sum_{k \geq 0} \lambda^k \cdot N (k)} \right)$$ (54)

$$= \frac{1}{n - 1} \cdot \left( N_{\max}^+ + \frac{\sum_{k \geq 0} \lambda^k \cdot k \cdot N (k) - \sum_{k \geq 0} \lambda^k \cdot N (k) \cdot N_{\max}^+}{\sum_{k \geq 0} \lambda^k \cdot N (k)} \right)$$ (55)

$$= \frac{1}{n - 1} \cdot \left( N_{\max}^+ - \frac{\sum_{k \geq 0} \lambda^k \cdot (N_{\max}^+ - k) \cdot N (k)}{\sum_{k \geq 0} \lambda^k \cdot N (k)} \right)$$ (56)

$$= \frac{1}{n - 1} \cdot \left( N_{\max}^+ - \frac{\sum_{0 \leq k < N_{\max}^+} \lambda^k \cdot (N_{\max}^+ - k) \cdot N (k)}{\sum_{0 \leq k \leq N_{\max}^+} \lambda^k \cdot N (k)} \right).$$ (57)

Observed from the above equation, the order of the numerator $\sum_{0 \leq k < N_{\max}^+} \lambda^k \cdot (N_{\max}^+ - k) \cdot N (k)$ is less than $N_{\max}^+$, but the order of the denominator $\sum_{0 \leq k \leq N_{\max}^+} \lambda^k \cdot N (k)$ is exactly $N_{\max}^+$. Therefore,

the order of the numerator is less than the order of the denominator. Thus, intuitively, as $\lambda \to +\infty$, the growth rate is much faster than that of the numerator. There must exist a constant $\lambda_0$ satisfying that when $\lambda > \lambda_0$, the fraction part of the above formula $\frac{\sum_{0 \le k < N_{\max}^+} \lambda^k \cdot \left(N_{\max}^+ - k\right) \cdot N(k)}{\sum_{0 \le k \le N_{\max}^+} \lambda^k \cdot N(k)}$ decreases as $\lambda$ increases. Thus, the proof of part (1) of the theorem concludes.

To prove parts (2) and (3) of the above theorem, we should further prove that:

$$\frac{N_{\max}^+}{n-1} = 1 - \left(\text{NHCC}\left(\widehat{G}\right) - 1\right) / (n-1). \tag{58}$$

It immediately follows if and only if:

$$N_{\max}^+ = n - \text{NHCC}\left(\widehat{G}\right). \tag{59}$$

Suppose the graph $\widehat{G}$ has $N_H = \text{NHCC}\left(\widehat{G}\right)$ numbers of homophilous connected components (*i.e.*, considering only the homophilous edges) $C_1, \cdots, C_{N_H}$. $N_{\max}^+$ is the maximum value of $N^+(T) = \sum_{e \in T} \mathbb{I}^{(\text{homo})}(e)$. This fact means that $N_{\max}^+$ is the sum of weights contained in the maximum spanning tree of the graph $\widehat{G}$ with the weights $\mathbb{I}^{(\text{homo})}(e)$. Considering the Kruskal's algorithm [2], we sort all edges of $\widehat{G}$ according to their weights, which means that all edges $e$ with $\mathbb{I}^{(\text{homo})}(e) = 1$ will be considered first. The rest of the edges with $0$ weights have no influence on the answer to the maximum spanning tree question.

All edges with weights $1$ are divided into $N_H$ components. Each component has a mutually independent solution. The answer of the component $C_i$ ($i \in [1, N_H]$) is obviously $|C_i| - 1$, since adding any further edge will cause a circle. Therefore, we have:

$$N_{\max}^+ = \sum_{k=1}^{N_H} (|C_k| - 1) = \sum_{k=1}^{N_H} |C_k| - N_H = n - N_H = n - \text{NHCC}\left(\widehat{G}\right), \tag{60}$$

which shows that Eq. 59 follows.

Injecting Eq. 59 into Eq. 57 concludes the proof of the theorem. $\qquad\square$

### B.3 Proof for Theorem 3

In this subsection, we provide the rigorous proof for Theorem 3, which is heavily based on the monotonicity analysis of $s$ with Respect to Maximum Degree $M$.

*Proof.* According to Lemma 1 and Lemma 2, we can derive the following formula for the lower bound of $s(\cdot)$:

$$s \ge \frac{1}{\left(1 - 2 \cdot \frac{\sqrt{M} - 1}{M}\left(1 - \frac{2}{D}\right) + \frac{2}{D}\right) \log\left(\frac{\max_i \sqrt{d_i}}{\epsilon \min_j \sqrt{d_j}}\right)} \tag{61}$$

where $M = \max_{v \in V} d_v$ and $D = D(G) \ge 4$.

First, simplify the denominator part involving $M$:

$$1 - 2 \cdot \frac{\sqrt{M} - 1}{M}\left(1 - \frac{2}{D}\right) + \frac{2}{D}$$

$$= 1 - \frac{2(\sqrt{M} - 1)}{M}\left(1 - \frac{2}{D}\right) + \frac{2}{D}$$

$$= \frac{M - 2(\sqrt{M} - 1)}{M} + \frac{4(\sqrt{M} - 1)}{MD} + \frac{2}{D}$$

$$= \frac{M - 2\sqrt{M} + 2}{M} + \frac{4\sqrt{M} - 4 + 2M}{MD}$$

---

[2] https://en.wikipedia.org/wiki/Kruskal%27s_algorithm

$$= \frac{D(M - 2\sqrt{M} + 2) + 4\sqrt{M} - 4 + 2M}{MD}.$$

Thus, the formula for $s$ can be rewritten as:

$$s \geq \frac{MD}{(D(M - 2\sqrt{M} + 2) + 4\sqrt{M} - 4 + 2M) \log\left(\frac{\max_i \sqrt{d_i}}{\epsilon \min_j \sqrt{d_j}}\right)}. \tag{62}$$

To analyze the monotonicity of $s$ with respect to $M$, define:

$$y = \frac{MD}{D(M - 2\sqrt{M} + 2) + 4\sqrt{M} - 4 + 2M}, \tag{63}$$

and compute its derivative $\frac{dy}{dM}$.

Using the quotient rule $\left(\frac{u}{v}\right)' = \frac{u'v - uv'}{v^2}$, where $u = MD$ and $v = D(M - 2\sqrt{M} + 2) + 4\sqrt{M} - 4 + 2M$:

First, compute $v'$:

$$v' = \frac{d}{dM}\left[D(M - 2\sqrt{M} + 2) + 4\sqrt{M} - 4 + 2M\right]$$
$$= D\left(1 - \frac{1}{\sqrt{M}}\right) + \frac{2}{\sqrt{M}} + 2.$$

Then, compute $y'$:

$$y' = \frac{D \cdot v - MD \cdot v'}{v^2}$$
$$= \frac{D\left[D(M - 2\sqrt{M} + 2) + 4\sqrt{M} - 4 + 2M\right] - MD\left[D\left(1 - \frac{1}{\sqrt{M}}\right) + \frac{2}{\sqrt{M}} + 2\right]}{v^2}.$$

Simplifying the numerator:

$$\text{Numerator} = D^2 M - 2D^2\sqrt{M} + 2D^2 + 4D\sqrt{M} - 4D + 2DM$$
$$- MD^2 + MD \cdot \frac{D}{\sqrt{M}} - 2MD \cdot \frac{1}{\sqrt{M}} - 2MD$$
$$= (D^2 - D^2)M + (-2D^2 + D^2 + 4D - 2D)\sqrt{M} + 2D^2 - 4D$$
$$= (-D^2 + 2D)\sqrt{M} + 2D^2 - 4D$$
$$= -D(D - 2)\sqrt{M} + 2D(D - 2)$$
$$= D(D - 2)(2 - \sqrt{M}).$$

Thus, the derivative simplifies to:

$$y' = \frac{D(D - 2)(2 - \sqrt{M})}{(D(M - 2\sqrt{M} + 2) + 4\sqrt{M} - 4 + 2M)^2}. \tag{64}$$

Given $D \geq 4$, we have $D - 2 > 0$ and $D > 0$. The sign of $y'$ is determined by $2 - \sqrt{M}$:

- When $M < 4$, $2 - \sqrt{M} > 0 \implies y' > 0$, so $y$ is increasing.

- When $M = 4$, $2 - \sqrt{M} = 0 \implies y' = 0$, so $y$ reaches a critical point.

- When $M > 4$, $2 - \sqrt{M} < 0 \implies y' < 0$, so $y$ is decreasing.

Therefore, the lower bound of $s$ first increases and then decreases with respect to the maximum degree $M$, reaching its peak at $M = 4$. In our case, we typically have $M \geq 4$ since there exist some nodes with larger degrees than other nodes. Besides, we have a graph augmentation before graph learning, which has an effect on improving the maximum degree of nodes, making it more likely to satisfy $M \geq 4$.

Then, to analyze monotonicity of $y$ on $D$, write:

$$y = \frac{MD}{D(M - 2\sqrt{M} + 2) + 2M + 4\sqrt{M} - 4}. \tag{65}$$

By the quotient rule, the derivative is:

$$\frac{dy}{dD} = \frac{M(2M + 4\sqrt{M} - 4)}{\left[D(M - 2\sqrt{M} + 2) + 2M + 4\sqrt{M} - 4\right]^2}. \tag{66}$$

For $M \geq 1$, since $2M + 4\sqrt{M} - 4 \geq 2 > 0$, it follows that $\frac{dy}{dD} > 0$. Therefore, $y$ is monotonically increasing in $D$.

Putting two monotonic analyses together concludes the proof of the theorem. □

### B.4 DERIVATIONS OF PATH DECOMPOSITIONS FOR SOME PARADIGMS

Recall several formulas in Sec. A.2 of Appn.:

$$H'_v = \text{Agg}(\{H_u\}_{u \in V}) = \sum_{u \in V} \sum_{\substack{\tilde{p} \in \mathcal{P}_{u \to v}(G): \\ \tilde{p} = (p_0 \to p_1 \to \cdots \to p_k = v) \\ \text{s.t. } p_0 = u, p_k = v}} H_u \cdot w(\tilde{p}) \cdot \text{PE}(\tilde{p}). \tag{67}$$

Here, $w(\tilde{p})$ is the path weight for a path $\tilde{p} = (p_0, p_1, \ldots, p_k)$:

$$w(\tilde{p}) = \prod_{j=1}^{k} w_{p_{j-1} \to p_j}, \tag{68}$$

where $w_{x \to y}$ is the weight of the directed edge $x \to y$. And $\text{PE}(\tilde{p})$ is an extra path-based positional encoding beyond vanilla pair-wise relative positional encoding:

$$\text{PE}(\tilde{p}) = f_{\text{path-PE}}(\tilde{p}, G, \{w_{xy}\}). \tag{69}$$

#### B.4.1 OTHER PARADIGMS

**Infinite-layer deep local SGC, deep local GT, or infinite-step random walk aggregation all have this form (Equation equation 67)**. The "infinite-layer" or "infinite-step" nature implies that the summation $\sum_{\tilde{p} \in \mathcal{P}_{u \to v}(G)}$ in Equation equation 67 considers all paths (walks) of any length $k$ from $u$ to $v$. For these models, the components are identified as:

(1) The edge weights $w_{x \to y}$ used in Equation equation 68 are specified as:

- For **Infinite-layer SGC (Simple Graph Convolution)**: $w_{x \to y} = S_{xy}$, where $S$ is the normalized adjacency matrix (e.g., $S = \tilde{D}^{-1/2} \tilde{A} \tilde{D}^{-1/2}$ with $\tilde{A} = A + I$). The path weight becomes $w_{\text{SGC}}(\tilde{p}) = \prod_{j=1}^{k} S_{p_{j-1} p_j}$. The full aggregation often takes a form like $H' = (\sum_{\ell=0}^{\infty} \beta^\ell S^\ell) H_{\text{in}}$ or $(I - \beta S)^{-1} H_{\text{in}}$. The sum over paths in Def. 1, when combined with the PE term, captures this.

- For **Infinite-layer deep local GT (Graph Transformer/Attention)**: $w_{x \to y} = \alpha_{yx}^{\text{shared}}$, representing a layer-shared attention coefficient for information flowing from $x$ to $y$ (where $y$ is the target node in the attention mechanism). The path weight is $w_{\text{GT}}(\tilde{p}) = \prod_{j=1}^{k} \alpha_{p_j p_{j-1}}^{\text{shared}}$.

- For **Infinite-step random walk aggregation**: $w_{x \to y} = T_{xy}$, an entry from the probability transition matrix $T$, where $T_{xy} = P(\text{next node} = y | \text{current node} = x)$. The path weight becomes $w_{\text{RW}}(\tilde{p}) = \prod_{j=1}^{k} T_{p_{j-1} p_j}$. This corresponds to the probability of traversing the path $\tilde{p}$. Aggregations like those based on Personalized PageRank $\sum_{\ell=0}^{\infty} \beta^\ell T^\ell$ inherently sum over weighted paths.

(2) The path-based positional encoding $\text{PE}(\tilde{p})$ for a path $\tilde{p} = (p_0, p_1, \ldots, p_k)$ is given by a product of node-specific positional encodings for all nodes along the path:

$$\text{PE}(\tilde{p}) = \prod_{i=0}^{k} \text{PE}(p_i). \tag{70}$$

Each node-specific positional encoding $\text{PE}(p_i)$ is defined as the sum of the discounted edge weight product in all circles (cycles) $C$ of any length $L_C$ that pass through node $p_i$. Let $\mathcal{C}(p_i, G)$ be the set of all simple cycles containing node $p_i$ in graph $G$. It can be formulated as follows:

$$\text{PE}(p_i) = \sum_{C \in \mathcal{C}(p_i, G)} \delta(L_C) \left( \prod_{(x,y) \in E(C)} w_{x \to y} \right), \tag{71}$$

where $E(C)$ are the directed edges forming the cycle $C$, $L_C$ is the length (number of edges) of cycle $C$, and $\delta(L_C)$ is a discount factor that depends on the length of the cycle (e.g., $\delta(L_C) = \gamma^{L_C}$ for some $0 < \gamma \le 1$). The edge weights $w_{x \to y}$ in this context are the same as those defined above for SGC, GT, or RW respectively. This structure (Equations equation 70 and equation 71) demonstrates that these methods focus more on local environmental importance of a path, e.g., densities or degrees of nodes contained in it, as captured by the properties of cycles involving nodes on the path.

### B.4.2    OUR PARADIGM

**Tree-Set (i.e., Forest) Layer also has this form (Equation equation 67)**, with $\text{PE}(\tilde{p})$ as the sum of weight products of all spanning trees of the graph obtained by merging path $\tilde{p}$ into a single node.

Let $G'_{\tilde{p}}$ be the graph obtained by contracting the path $\tilde{p} = (p_0, \ldots, p_k)$ into a single *supernode* $v_{\tilde{p}}$. The edges incident to $v_{\tilde{p}}$ are derived from edges incident to any $p_i \in \tilde{p}$ in $G$. Let $\mathbb{T}(G'_{\tilde{p}})$ be the set of all spanning trees in $G'_{\tilde{p}}$. For each spanning tree $T \in \mathbb{T}(G'_{\tilde{p}})$, its weight product is $W(T) = \prod_{e \in E(T)} w'_e$, where $w'_e$ are the (possibly re-defined) edge weights in $T$.

The path positional encoding is:

$$\text{PE}(\tilde{p}) = \sum_{T \in \mathbb{T}(G'_{\tilde{p}})} W(T) = \sum_{T \in \mathbb{T}(G'_{\tilde{p}})} \left( \prod_{e \in E_T} w'_e \right). \tag{72}$$

This shows that this paradigm focuses more on the global transport importance of a path, *i.e.*, how connectivity or communication this path can facilitate if it is built as a highway with no communication cost along it.

## C    EXTENSIONS OF OUR TREE AGGREGATOR

In this section, we will introduce some extensions of our tree aggregator in Sec. 4.3, which not only shows the potential of the proposed general aggregator, but also reveals how it can be technically combined with other popular techniques, exhibiting some possible future research directions on general graph learning.

### C.1    INTEGRATED WITH GLOBAL ATTENTIONS

Based on kernel decomposition, *i.e.*, $k(x, y) = f(x)^T \cdot f(y)$. We can easily inject global linear and even general attention paradigm into our aggregator. In the attention mechanism, $k(Q, K)V = f(Q)f(K)^T V$. Thus, we can treat $f(K)^T V$ as messages fed into the aggregator. Note that $Q$ need to conduct tensor product with $f(K)^T V$. Dimension reduction can be further utilized to reduce the memory consumption.

## C.2 Fine-Grained Propagation Control

In this subsection, we design fine-grained propagation control, such as discounting or truncating the distance, which can make the tree aggregators more flexible and better filter the possible distant noise.

We can discount the distance by introducing a discounting weight and multiplying it by the edge weight. We can truncate the distance by introducing an extra variable to store how distant the currently embedding aggregates information.

## C.3 Generalize Forests

In this subsection, we generalize forests to eliminate the need for Recursion (II). We can generalize the forests to the Directed Acyclic Graphs (DAGs) forests. Each time, we randomly sample one *directing solution* for each edge and form a DAG. And use a similar tree aggregator to aggregate distant messages and conduct fusion for them. Yet, it needs extra sampling strategy and tools for DAGs, which is still under-developped.

# D  Accelerations of Our Tree Aggregator

In this section, we introduce some tricks to accelerate our tree aggregator.

## D.1 Selecting Centroid as Root

We can select the centroid node as the root to conduct the recursions on trees rather than choose the node 0, since the width of the rooted tree would be larger and the depth of the rooted tree would be smaller. This would facilitate higher parallelizability. For example, when we conduct leaf-to-root recursion, multiple threads can be better utilized. The centroid node can be found via two Depth-First Searches, which are efficient.

## D.2 Different Greedy Strategies for Different Recursions

Our recursions can use greedy strategies to improve parallelizability. When conducting leaf-to-root recursion, we can calculate the depth $\text{dep}(v)$ and sub-tree depth $\text{dep}^{(\text{sub})}(v)$ of each node $v$. We treat $\text{dep}^{(\text{sub})}(v)$ as the first key-words and treat $\text{dep}(v)$ as the second key-words to sort all nodes. When conducting root-to-leaf recursion, we can use $\text{dep}(v)$ as the only keyword. We can assign nodes for threads one by one based on $i \mod N$, where $N$ is the number of threads.

# E  Details of Block Acceleration of Tree Sampler

As described in Sec. 4.2 in the main text and Sec. G of Appn. as well as Algorithm 2, we follow some Theoretical Computer Science previous works Wilson (1996) and adopt a random walk-based spanning tree sampler. It can sample a random tree exactly from the given tree distribution with linear running times in most cases. However, when dealing with larger graphs, a linear time complexity is sometimes not enough to satisfy practical requirements. The essential reason is that the parallelizability is based on the specific input graph structure. When the graph becomes more dense, a single thread may have a shorter life period, and more threads are needed. Yet, when more threads simultaneously work at nearby locations in a graph, they may interfere with each other. For example, supposing that two threads produce two paths $p_1$ and $p_2$ after circle/loop stripping, $p_1$ and $p_2$ can intersect many times. To maintain the correctness of this sampler, we must drop some parts of these paths, which definitely causes computational waste to some extent. More severely, as the multi-threaded working paradigm proceeds, the under-explored area in the graph becomes much localler (akin to some smaller connected components), and in a nearby environment, the number of threads must be reduced (otherwise the useless walk/path would frequently occur), which reduces the parallelizability.

In this section, we propose a simple trick (*i.e.*, Algo 3), which calls the vanilla Tree Sampler (*i.e.*, Algo 2) with some extra tools and intuitively improves its parallelizability with only limited precision sacrifice. In other words, we propose to conduct approximate sampling with ignorable deviations, yet

make better use of modern architectural advantages. We describe some specific details in Algo. 3 for the convenience of implementation. Furthermore, we provide detailed explanations of the algorithmic pseudo-code in Sec. E of Appn.

The key idea is to identify some unimportant edges in the original graph and attempt to drop them or at least some of them. If the graph can be divided into several disconnected parts, we are then allowed to conduct a separate tree sampling for each of them.

How to integrally achieve this end poses a challenge, especially considering we expect to minimize the number of dropped edges. To demonstrate our motivation more clearly, we consider solving the problem starting from a simple greedy strategy. Since we already know the importance score of each edge, *i.e.*, $\mathbf{s}$ (recall Sec. 4.2), a natural heuristic greedy strategy is to sort all edges by the edge scores ascendingly and then drop the edges one by one until the number of connected components achieving the pre-defined number, *e.g.*, $K_B$. However, the significant issue of the greedy strategy is that to achieve the number $K_B$, we may risk dropping too many edges. Despite the fact that some of these dropped edges are low-score, the cumulative effect cannot be ignorable. To understand this, consider the following example with $K_B = 2$. Suppose that we have an $n$-sized random tree $T$ which is uniformly sampled from an $n$-sized complete graph. We label all edges in $T$ with edge scores 1. Assuming $n$ is even, we can find the two centroid nodes of the tree $T$ and denote them as nodes $u$ and $v$. Cutting the edge $u \leftrightarrow v$ would cause the tree $T$ to become two parts. We denote the left node set and the right node set with notations $V_{\text{left}} \subseteq V$ and $V_{\text{right}} \subseteq V$. We then continuously add edges with uniform random edge scores between 0 and $1 - \epsilon$ (*i.e.*, $s(e) \sim \text{Uniform}(0, 1)$ and $0 < \epsilon \ll 1$) to the $V_{\text{left}}$-induced subgraph (denoted $G_{\text{left}}$) and the $V_{\text{right}}$-induced subgraph (denoted $G_{\text{right}}$) until both of them becoming complete graphs. Next, we consider the solution derived from the above greedy strategy for the constructed graph $G$. It will drop all the edges with scores smaller than 1, *i.e.*, all the edges that are added after sampling the initial tree $T$, and then drop one of the edges in $T$. An obvious issue of this method is that the sum of all the dropped edges would be too large to ignore their cumulative effects. In fact, the expected sum of the dropped edges is $\mathcal{O}(n^2)$. On the other hand, the optimal solution of the constructed graph $G$ is obvious, *i.e.*, deleting the single edge $u \leftrightarrow v$ with the total edge scores equaling 1 (*i.e.*, $\mathcal{O}(1)$). From another perspective, we expect the obtained two components are relatively balanced, *i.e.*, $|V_{\text{left}}| \approx |V_{\text{right}}|$. The optimal solution follows this requirement, yet the greedy strategy may fail to achieve this, due to the randomness of the sorting algorithm for those edges with exactly the same scores.

Therefore, based on the above analysis, we can outline the conditions that we expect the graph division technique to holistically satisfy: (1) The divided parts are expected to be maximally balanced. (2) The number of dropped edges is expected to be minimal. (3) The sum of scores of the dropped edges is expected to be minimal. We notice that these conditions are almost the merits of some graph cut techniques, *e.g.*, METIS library Karypis & Kumar (1998), which is treated as an operator (denoted as $\text{GraphCut}(\cdot)$) called in the first step of our algorithm.

After obtaining the several components split via $\text{GraphCut}(\cdot)$, we call Algo. 2 to separately sample a tree for each component and then we obtain several trees $T_1, \cdots, T_{K_B}$. However, how to merge them and how to consider the dropped edges remain challenging. To this end, we propose to collapse each component to a single node labeled with the component label (*i.e.*, the block number, labeled, *e.g.*, between 1 and $K_B$). Then all edges between these components (*i.e.*, blocks) can be re-labeled with their end nodes' block numbers and will become multi-edges connecting two newly labeled nodes. For example, a vanilla edge $73 \leftrightarrow 254$ would become $4 \leftrightarrow 7$, where 73 and 254 are two vanilla node labels as well as 4 and 7 are two block numbers (*i.e.*, two new labels of merged/collapsed new nodes, 73 comes from 4-th block, and 254 comes from 7-th block). Along this way, we can obtain a highly collapsed small graph with exactly $K_B$ nodes and many edges coming from vanilla edges connecting two different blocks/components. We call this new graph $G'$. It contains no self-loops since we drop all intra-block vanilla edges (note that when sampling $T_1, \cdots, T_{K_B}$, we retain only these edges and drop all the other edges). It also contains many edges that connect two same pair of nodes (new nodes), which we call *multi-edges*. We then merge a multi-edge (containing many edges) into a single edge, with its edge weight as the sum of vanilla edge weights. Here, our idea is to sample a tree for this new graph $G'$, and then down-sample a selected tree edge (corresponding to a multi-edge before merging) into a vanilla edge contained in the corresponding multi-edge. This process can be proven equivalent to directly sampling a tree for the new graph with repeating edges that have different edge weights, where the latter cannot be conveniently implemented via directly calling Algo. 2.

The edge down-sampling process requires us to simultaneously conduct multiple (exactly $K_B$) sampling operations from variable-length categorical distributions. For example, we are required to parallelizably sample a $x$ from the categorical distribution $[\frac{1}{2}, \frac{1}{4}, \frac{1}{4}]$ and at the same time, sample a $y$ from another distribution with different length, $e.g.$, $[\frac{3}{7}, \frac{4}{7}]$. This makes GPU-level parallelization inconvenient. The challenge can be addressed via a Gumbel trick, described below. We first normalize the edge weights in each first-order neighborhood to construct each categorical distribution. This step can be done via a $\mathrm{Scatter\_Add}$ operator to efficiently calculate the sum of first-order edge weights. Then, we compute the $\log$ values for all probabilities and add element-wise *Gumbel* random variables, $i.e.$, $x = -\log\left(-\log\left(t\right)\right),\ t \sim \mathrm{Uniform}\left(0,\ 1\right)$. Next, the most key step is to conduct a $\mathrm{Scatter\_ArgMax}\left(\cdot\right)$ operator to allow the sampling distribution to have different lengths, which is significantly beyond vanilla $\mathrm{ArgMax}\left(\cdot\right)$ after some operation such as $*.\,\mathrm{reshape}\left(\cdot\right)$. Through these steps, we can efficiently conduct edge down-sampling and obtain a new tree with vanilla node labels (between 1 and $n$). We call this tree $T_0$. Merged with other obtained trees, $i.e.$, $T_1,\ ,\cdots,\ T_{K_B}$, we can obtain the final tree $T = \mathcal{M}\left(\{T_k\}_{k \in [0,\ K_B]}\right)$.

# F  Details of Algorithms

In this section, we provide several algorithmic pseudo-codes to support the detailed implementations of our main framework, as well as some discussions in other sections of this Supplementary Materials. In each algorithm, we will shortly provide a description of this algorithm, the input/output, important hyper-parameters, learnable parameters, and other notes on, $e.g.$, some extra definitions or clarifications of symbols or operators, as well as some tools called from another library. In the corresponding subsection of one algorithm, we will provide line-by-line explanations to detail the specific implementation, including some notable points.

## F.1  Algorithm of Our Framework

We provide a detailed introduction part by part in Sec. 4 of the main text. Next, we will provide an algorithmic pseudo-code ($i.e.$, Algo. 1) to connect these parts from a more holistic perspective for a better understanding.

Furthermore, we will provide a line-by-line description for this algorithmic pseudo-code ($i.e.$, Algo. 1) as follows:

- Line 1 extracts some knowledge from the non-attention-based model $\mathcal{T}_0$ via pretraining. In our implementation, for simplicity, we select a single-layer GCN/MLP for our $\mathcal{T}_0$ to show our true potential (rather than based on powerful candidates). Also, $\mathbb{K}_0$ is treated as direct node predictions, $i.e.$, node label prediction probabilities. Other implementations may also be acceptable or better.

- Line 2 augments the vanilla graph into an augmented variant, based on the extracted predictions $\mathbb{K}_0$ (note it is optional). The main motivation is to make the graph connected and to improve the initial homophily ratio or its NHCC ( especially for heterophilous graphs ). A simple top-k augmentation is enough to achieve this end (Sec. L).

- Line 3 extracts new knowledge $\mathbb{K}_1$ from the attention-based model $\mathcal{T}_1$ based on the old yet necessary auxiliary information $\mathbb{K}_0$. The detailed motivation for why we need additional old information $\mathbb{K}_0$ to extract new knowledge $\mathbb{K}_1$ can be found in Sec. 4.2 of the main text and Sec. A.5. In our implementation, the new knowledge is modeled as attention coefficients of the module $\mathcal{T}_1$, and thus it can also be viewed as a kind of attention learning.

- Line 4, based on the knowledge $\mathbb{K}_1$, defines a tree distribution conditioned on the graph $\widehat{G}$, $i.e.$, $P_{\widehat{G}}\left(T\right)$.

- Line 5 $\rightarrow$ 7 separately sample a tree from the tree distribution $P_{\widehat{G}}\left(T\right)$. We sample $N_T$ trees in total and these trees can be viewed a *iid.* sample of size $N_T$ from the distribution.

- Line 8 inputs the vanilla graph $G$ as well as its feature matrix $X$ into our Local Submodule (Sec. 4.4) of our architecture, and we obtain the node embedding matrix $H \in \mathbb{R}^{n \times d}$.

- Line 9 $\rightarrow$ 11 separately fed the sampled tree $T_k$ as well as the embedding matrix $H$ into the tree aggregator, and obtain new embeddings $H^{'(k)} \in \mathbb{R}^{n \times d}$. The tree aggregator will

aggregate distant messages on each tree independently. In other words, $H^{'(k)}$ depends only on the $k$-th tree.

- Line 12 fuses all these node embeddings from different trees into single ones, *i.e.*, $H' \in \mathbb{R}^{n \times d}$. We directly implement the operator $\mathrm{Fuse}\,(\cdot)$ via a simple post-hoc mean fusion. Yet, note that other sophisticated alternatives are options, too.

- Line 13 represents a residual connection governed by a hyper-parameter $\gamma \in [0, 1]$, which controls the trade-off between the local knowledge and global knowledge.

---

**Algorithm 1** Algorithm of Our Framework

---

**Description**: The holistic architecture of our FGL framework mainly contains four critical steps.

**Input**: an input graph $G = (V,\ E)$ with feature matrix $X \in \mathbb{R}^{n \times f}$ and normalized adjacency matrix $\widehat{A}$, the node labels in the training set $Y_L$, a graph augmenter $\mathrm{Aug}\,(G;\ \mathbb{K})$ (to augment graph with some auxiliary information $\mathbb{K}$), a non-attention-based graph layer $\mathcal{T}_0$, an local attention-based graph layer $\mathcal{T}_1$, training operator $\mathrm{Train}\,(\mathcal{T} \mid \mathbb{K})$ returning the outputs or predictions of the trained auxiliary modules (*e.g.*, $\mathcal{T}_0$ and $\mathcal{T}_1$) where auxiliary information $\mathbb{K}$ may contain both training labels and training inputs as well as testing inputs, a tree distribution definer $\mathrm{Define}\,(\cdot)$, and some key technical components including Tree Sampler $\mathrm{TreeSampler}\,(\cdot)$, Tree Aggregators $f_{\mathrm{Agg}}^{(T)}\,(\cdot)$, and Tree Fuser $\mathrm{Fuse}\,(\cdot)$.

**Output**: node embeddings $H'' \in \mathbb{R}^{n \times d}$

**Hyper-Parameters**:

  $\beta_1,\ \beta_2$, the hyper-parameters in our Local Submodule (Eq. 9);

  $\gamma$, the residual coefficient;

  $K_L \le 2$, the number of sub-layers of our Local Submodule;

  $N_T$, the number of sampled trees

**Trainable-Parameters**: learnable parameters in Local and Global Submodules as well as auxiliary modules (layers), *i.e.*, $\mathcal{T}_0$ and $\mathcal{T}_1$

**Note**: For brevity, here we aim to present the *high-level idea*, and detailed text descriptions are provided in Sec. 4 of the main text.

1: $\mathbb{K}_0 \leftarrow \mathrm{Train}\,(\mathcal{T}_0 \mid G,\ Y_L)$

2: $\widehat{G} \leftarrow \mathrm{Aug}\,(G;\ \mathbb{K}_0)$

  ▷ The main motivation is to make the graph connected and to improve the initial homophily ratio or its NHCC. A simple top-k augmentation is enough to achieve this end.

3: $\mathbb{K}_1 \leftarrow \mathrm{Train}\left(\mathcal{T}_1 \mid \mathbb{K}_0,\ \widehat{G},\ Y_L\right)$

4: $P_{\widehat{G}}\,(T) \leftarrow \mathrm{Define}\,(\mathbb{K}_1)$

  ▷ Define a tree distribution based on the knowledge $\mathbb{K}_1$ (Sec. 4.2).

5: **for** each $k \in [1,\ N_T]$ **do**

6:   $T_k \leftarrow \mathrm{TreeSampler}\left(P_{\widehat{G}}\,(\cdot)\right)$

7: **end for**

8: $H \leftarrow \mathrm{Local}\,(X;\ \beta_1,\ \beta_2,\ G,\ \mathbb{K}_1,\ K_L)\ \in \mathbb{R}^{n \times d}$

9: **for** $k \in [1,\ N_T]$ **do**

10:   $H^{'(k)} \leftarrow f_{\mathrm{Agg}}^{(T_k)}\,(H)\ \in \mathbb{R}^{n \times d}$

11: **end for**

12: $H' \leftarrow \mathrm{Fuse}\left(\left\{H^{'(k)}\right\}_{k \in [1,\ N_T]}\right)\ \in \mathbb{R}^{n \times d}$

  ▷ We directly implement the operator $\mathrm{Fuse}\,(\cdot)$ via a simple post-hoc mean fusion. Yet, note that other sophisticated alternatives are options, too.

13: $H'' \leftarrow \gamma \cdot H' + (1 - \gamma) \cdot H\ \in \mathbb{R}^{n \times d}$

14: **return** $H''\ \in \mathbb{R}^{n \times d}$

---

## F.2 ALGORITHM OF TREE SAMPLER

Recall the brief description of the utilized tree sampler in Sec. 4.2 and a more detailed introduction in Sec. G of Appn. To facilitate a better understanding, in this subsection, we will provide an algorithm pseudo-code as well as line-by-line explanations for the convenience of implementations or some possible future extensions and improvements.

We give an explanation of Algorithm 2 as follows:

- Lines 1 computes the transition matrix of a Markov Chain defined based on edge weights $\mathbf{s} = \{s_e\}_{e \in E}$ according to Eq. 74.

- Lines 2 randomly generate a node permutation $a[\cdot]$ with its first element as the root of the sampled tree in the graph.

- Lines $3 \rightarrow 5$ initialize the arrays $\mathrm{Next}[\cdot]$ and $\mathrm{InTree}[\cdot]$. The former denotes a temporary linked array pointing from the leaf nodes to the root node of the sampled tree (we treat the temporary tree as a rooted tree with root node $a[0]$). The latter denotes whether a node $v$ is contained in the current (rooted) sampled tree.

- Line 6 takes the root node $a[0]$.

- Line $7 \rightarrow 23$ finds a node $v \in V$ that is not in the current tree and starts a new walk (recall the above-mentioned Markov Chain) from the node $v$ (until encountering a node already in the current tree). The walk would contain some repeated nodes, and we should strip all circles or loops to obtain a path. Finally, we take all the nodes and edges, and merge them into the tree node and tree edge sets, respectively.

- Line $8 \rightarrow 11$ attempt to identify a new node that is not in the current tree to start a new walk.

- Line $12 \rightarrow 16$ sample a walk step by step (each time sampling a node in a categorical distribution) and record this walk in the way of $\mathrm{Next}$ pointers. Notably, the iteratively sampling operations implicitly strip all the loops or circles by substituting the old $\mathrm{Next}\,(u)$ with a new value, considering this point from the perspective of a linked array, and obtain a directed path. The walk (or the directed path) will end at a node that is already contained in the current tree, *i.e.*, $\mathrm{InTree}[v] = \mathrm{True}$.

- Line $17 \rightarrow 22$ add all nodes and edges in the path into the tree node set and the tree edge set, respectively.

- Line 24 returns the sampled tree.

---

**Algorithm 2** Algorithm of Tree Sampler

---

**Description**: Given a positively weighted directed graph $G$, define a tree distribution conditioned on graph $G$ as $P_G(T)$, with the unnormalized score equaling to the product of all edge weights in a tree (Recall Eq. 2 in the main text). Return a spanning tree of graph $G$ sampled from $P_G(T)$. Check Sec. F.2 for the line-by-line explanations of this algorithm.

**Input**: a graph $G = (V, E)$ with first-order neighborhoods $\{N_v\}_{v \in V}$ (or equivalently the adjacency list of $G$), edge weights $\mathbf{s} = \{s(e)\}_{e \in E}$ where $e \in E$ is a undirected edge of graph $G$

**Output**: the tree $T \sim P_G(T)$ with $T = (V_T, E_T)$, where $V_T$ and $E_T$ denote its node set and its (undirected) edge set, respectively

**Note**: (1) Next $[v]$ is a "*next*" pointer directing the next node after node $v$, noting that it implicitly includes a loop-striping process. The Next $[\cdot]$ operator with a Head node defines a linked array, representing a random walk. Since it implicitly includes a loop-stripping process, the above random walk is essentially a path. InTree $[v] \in \{\text{True}, \text{False}\}$ denotes whether node $v$ is in the current tree.

(2) This tree sampler can be highly parallelized via, *e.g.*, the OpenMP library at https://www.openmp.org/. Besides, it can naturally support simultaneously sampling multiple trees. We also provide a block acceleration trick (Algo. 3), which can further improve its parallelizability with only an ignorable deviation from standard sampling.

1:  Calculate a Markov Chain (transition matrix) based on the edge weights $\mathbf{s}$, recalling Eq. 74:

$$p_{i \to j} = \frac{s(e = (i \leftrightarrow j))}{\sum_{e' = (i \leftrightarrow k) \in \widehat{G}} s(e')} \in [0, 1]. \tag{73}$$

2:  Generate a random permutation of $V = \{v_0, v_1, \cdots, v_{n-1}\}$, *i.e.*, $a[\cdot]$, where $n = |V|$.

3:  **for** each $v \in V$ **do**

4:      Next $[v] \leftarrow -1$,    InTree $[v] \leftarrow$ False

5:  **end for**

6:  InTree $[a[0]] \leftarrow$ True,    $V_T \leftarrow \{a[0]\}$,    $E_T \leftarrow \emptyset$

7:  **for** each $t \in [0, n-1]$ **do**

8:      $v \leftarrow a[t]$,    $v_0 \leftarrow v$

9:      **if** InTree $[v]$ **then**

10:          Continue

11:      **end if**

12:      **while** InTree $[v]$ = False **do**

13:          $u \leftarrow$ Sample $\left(u; \{p_{v \to u}\}_{u \in N_v}\right)$

14:          Next $[v] \leftarrow u$

15:          $v \leftarrow u$

16:      **end while**

17:      **while** InTree $[v_0]$ = False **do**

18:          InTree $[v_0] \leftarrow$ True

19:          $V_T \leftarrow V_T \cup \{v_0\}$

20:          $E_T \leftarrow E_T \cup \{v_0 \leftrightarrow$ Next $[v_0]\}$

21:          $v_0 \leftarrow$ Next $[v_0]$

22:      **end while**

23:  **end for**

24:  **return** $T \leftarrow (V_T, E_T)$

### F.3 Algorithm of Block Acceleration of Tree Sampler

As introduced in Sec. E, we propose a *block acceleration* tree to improve the parallelizability of the tree sampler (see Algorithm 2), which can be efficiently and conveniently implemented via Tensor operations (using Torch library https://pytorch.org/) as well as some other tools.

We give an explanation of Algorithm 3 as follows:

- Line 1 call the $\mathrm{GraphCut}$ operator to cut the graph into several blocks and return the block numbers of all $n$ nodes in $V$, where $\mathrm{BlockNo}\,(v)$ denotes the block number of node $v \in V$. The motivation can be found in Sec. E for details.

- Line 2 calculates the new edge index of the graph $G$ with new node labels, *i.e.*, block numbers.

- Lines $3 \rightarrow 6$ obtain the edge index and the corresponding edge weights inside each block.

- Line 7 obtains the edge mask indicating all edges between blocks. These edges are essentially edges in $G'$, but note the node labels.

- Line $8 \rightarrow 9$ obtain vanilla edges and weights between blocks.

- Line 10 relabel the between-block edges (*i.e.*, the edge index) inside $G'$ (recall Sec. E).

- Line 11 merges a multi-edge into a single edge. The weight of the single edge is the sum of the edge weights of vanilla edges contained in this multi-edge.

- Line $12 \rightarrow 14$ calls the operator $\mathrm{TreeSampler}\,(\cdot)$ to separately sample a tree for each block, including the special block 0, which takes blocks as new nodes.

- Line $15 \rightarrow 16$ compute the row index of each related edge positioned in the edge index of the tree $T_0$, which is also the ID of a multi-edge. Each related edge is contained in a multi-edge and can be numbered with the ID of the multi-edge. In other words, $\mathrm{Index}\,[i] = j$ represents the $i$-th related edge (relabeled with the respective block number) corresponds to the $j$-th tree edge in block 0 (*i.e.*, $\mathrm{TreeBlock}[0]$). Each tree edge in $\mathrm{TreeBlock}[0]$ (*i.e.*, an above-mentioned single multi-edge) corresponds to many relabeled edges (called *related* edges).

- Line 17 selects all the related edges. The edge index has many repeating rows, *e.g.*, $[3,\ 5]$, $[3,\ 5]$, $[3,\ 5]$, where 3 and 5 denote the 3-th and the 5-th collapsed nodes (*i.e.*, block numbers). Note that they represent different vanilla edges.

- Line 18 selects those edge weights. Different repeating rows in the edge index may have different edge weights, since they are essentially different edges, yet with the same pair of end node block numbers.

- Line 19 calculates the edge weight of a multi-edge as the sum of all the related edges it contains. For example, there are 7 numbers of $[3,\ 5]$s with edge weights $s_0, \cdots, s_6$, and then they will be merged into a single edge (a multi-edge) with the weight $s = \sum_{i=0}^{6} s_i$. Now, there is only one copy of $[3,\ 5]$.

- Line 20 represents a normalization operation, which can calculate $n$ categorical distributions with possibly different lengths.

- Line 21 represents a Gumbel sampling step. We calculate the $\log$ values and add independently sampled Gumbel variables to them. Please refer to Jang et al. (2016) for a detailed introduction to Gumbel sampling or the Gumbel Softmax trick, which is popular for categorical sampling.

- Line 22 is the most key step. We utilize the operator $\mathrm{Scatter\_ArgMax}\,(\cdot)$ to select the maximum element for each group simultaneously, which is typically much more efficient than those less-optimized simple variants of practitioners.

- Line 23 concatenates all the trees from different blocks (*i.e.*, $T_0, \cdots T_{K_B}$) into a single tree $T$, which is our sampled tree.

- Line 24 returns the tree $T$.

## G  DETAILS OF EFFICIENT TREE SAMPLER

Restricting the tree distribution into such a parametric class (Eq. 2) has another benefit, *i.e.*, it has a deep connection with extensive prior studies on random spanning trees in Theoretical Computer Science Wilson (1996); Broder (1989); Kelner & Madry (2009); Durfee et al. (2017); Wilson & Propp (1996). Sampling such trees conditioned on a given graph can be achieved via different techniques, *e.g.*, determinant calculation/Matrix Tree Theorem, random walk Wilson (1996), and effective resistance Durfee et al. (2017). For simplicity and implementation convenience, we follow the study Wilson (1996) and adopt a random walk-based tree sampler (demonstrated in Algorithm 2). It maintains real-time tree node and edge sets $V_T$ and $E_T$, and at each iteration, find a node $v \notin V_T$ and start a new random walk $\mathcal{W}_v$ from $v$ (*i.e.*, $\mathcal{W}_v = (v \to v_1 \to v_2 \to \cdots \to v')$) until encountering a node $v' \in V_T$. The walk $\mathcal{W}_v$ is produced following the Markov Chain induced from the edge scores $\mathbf{s} = \{s(e)\}_{e \in \widehat{G}}$, *i.e.*:

$$p_{i \to j} = \frac{s(e = (i \leftrightarrow j))}{\sum_{e' = (i \leftrightarrow k) \in \widehat{G}} s(e')} \in [0, 1]. \tag{74}$$

After loop stripping, all nodes and edges contained in $\mathcal{W}'_v = \text{LoopStrip}(\mathcal{W}_v)$ are merged into $V_T$ and $E_T$, respectively. More details are provided in Algorithm 2. This sampler is simple and efficient with only $\mathcal{O}(\tau(p))$ running time, where $\tau(p)$ denotes the expected hitting time of two random nodes $u, v \in V$ sampled from stationary distribution $\pi_p$ of the Markov Chain $p$ (Eq. 74):

$$\tau(p) = \mathbb{E}_{u, v \sim \pi_p}[\text{Step}(u \to v)] = \sum_{u, v \in V} \pi_p(u) \cdot \pi_p(v) \cdot \text{Step}(u \to v), \tag{75}$$

where $\text{Step}(u \to v)$ denotes the expected number of steps for a random walk from node $u$ to (hit) node $v$ at the first time. As pointed out in Wilson (1996), $\mathcal{O}(\tau(p)) \approx \mathcal{O}(|V|)$ for most random graphs, which is much faster than their cover time (typically $\mathcal{O}(|V| \log |V|)$). Notably, it can also be parallelized and sample multiple trees simultaneously. Moreover, we propose a further optional *block acceleration* when dealing with very large graphs (detailed in Algorithm 3 and Sec. E of Appn.) with a Graph Cut technique and Gumbel-Softmax edge down-sampling, which sacrifices only some unimportant low-score edges.

# H    SET OF MOTIVATION EXPERIMENTS

In this section, we will provide some motivation experiments, whose results will justify some parts of the design of our framework.

## H.1    STUDIES ON EFFECT OF HOMOPHILY ESTIMATION ON GENERALIZABILITY

Based on Fig. 8, it can be clearly seen that as the accuracy rate of the homophilous estimator increases, the performance on all our datasets is consistently and continuously growing. Moreover, in extreme cases, when the homophilous estimator has absolute performance (with an accuracy rate of 1), the precision of all datasets also reaches 1. Thus, it can be proved that when we find a better homophilous estimator, we can obtain better generalization performance, and there is no bottleneck. Therefore, it tells us to pursue the quality of the homophilous estimator rather than randomness.

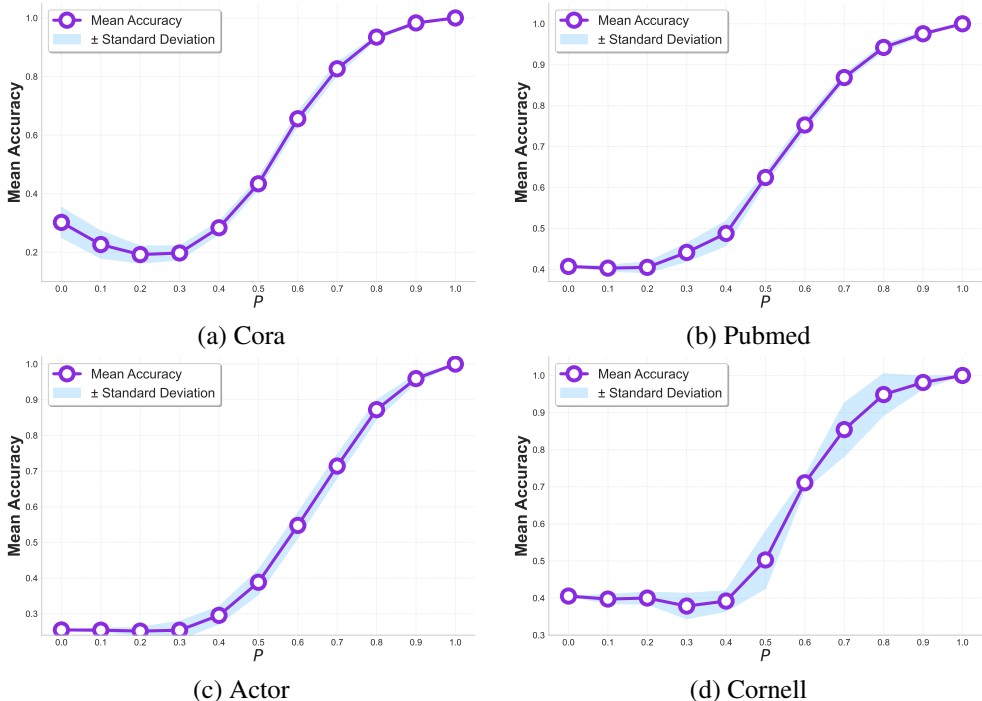

(a) Cora

(b) Pubmed

(c) Actor

(d) Cornell

Figure 8: Effect of homophily estimator accuracy on model performance

## H.2 EFFECTS OF PREDICTION-BASED GRAPH AUGMENTATION ON NHCC

We conduct a study on how the graph augmentation affects the values of NHCC. As illustrated in Fig. 9, the distillation-based top-k augmentation strategy (Sec. L) can effectively improve the NHCC values (*i.e.*, the number of homophilous connected components), which are the theoretical upper bounds of our designed ideal tree distributions.

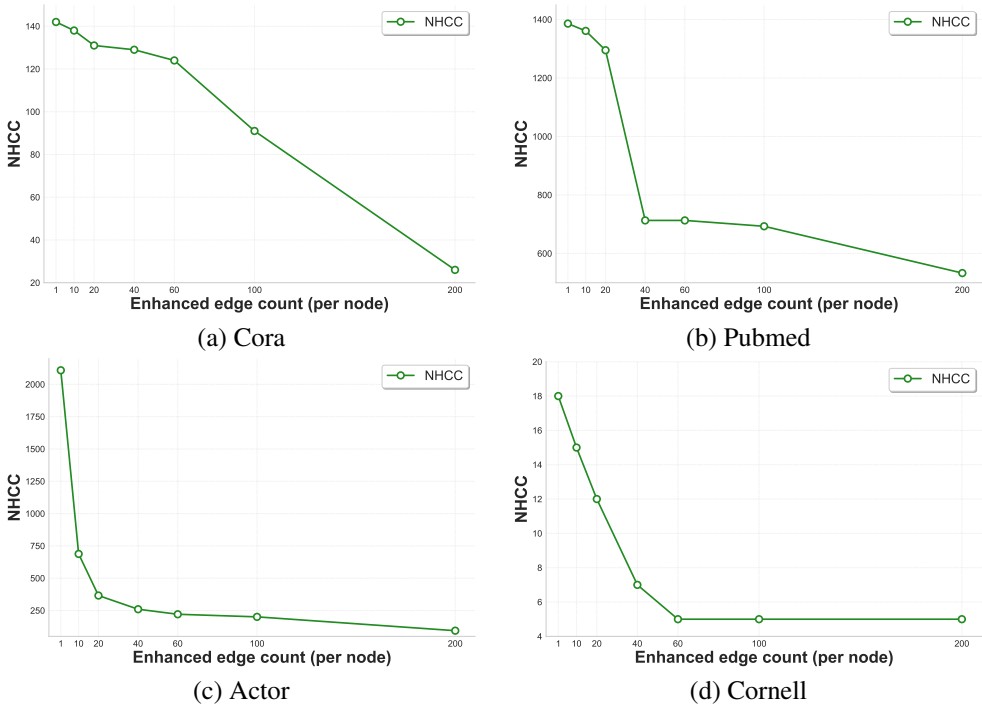

(a) Cora

(b) Pubmed

(c) Actor

(d) Cornell

Figure 9: Variation of NHCC with graph augmentation edge numbers. Discussed in Sec. H.2

## H.3 Studies on Degree Imbalance Issue of Sigmoid Edge Estimator

There is an important alternative comparison for the definition of tree distribution scores mentioned in in Sec. 4.2 of the main text. If we modify the score definition to sample based on the edge weights of the embedded inner product sigmoid, the generated tree will face a serious degree imbalance problem. This imbalance phenomenon stems from the fact that the sigmoid function tends to generate a polarized weight distribution, resulting in some nodes receiving excessive connections and attention while others are ignored or marginalized.

The proposed tree sampling distribution can effectively alleviate this problem and achieve an improvement in degree balance of nearly 40% for most moderate-sized graphs. In terms of implementation, we adopt the variance of the node degrees on the tree as the key metric to measure the degree balance, *i.e.*, the smaller the variance value, the more uniform the degree distribution between nodes and the more balanced the tree structure. Formally, the metric of node degree imbalance can be formulated as follows:

$$\mathcal{V}^{(\text{degree})} = \sqrt{\frac{1}{|V|} \cdot \sum_{v \in V} \left(d_v - \bar{d}\right)^2}, \tag{76}$$

where $d_v$ is the degree of node $v \in V$ and $\bar{d} = \frac{1}{|V|} \sum_{v \in V} d_v$.

Fig. 10 indicates that the spanning tree generated by the traditional sigmoid sampling method based on the embedded inner product has a relatively high degree variance, reflecting a serious structural imbalance. Sampling trees from this distribution significantly reduces this variance, leading to more uniform node connections in the tree structure. The improvement of this degree balance not only helps to avoid deviations in the process of information aggregation, but also ensures that each node in the network can obtain relatively fair information dissemination opportunities and the richness of information, thereby enhancing the expressive ability and generalization performance of the overall model.

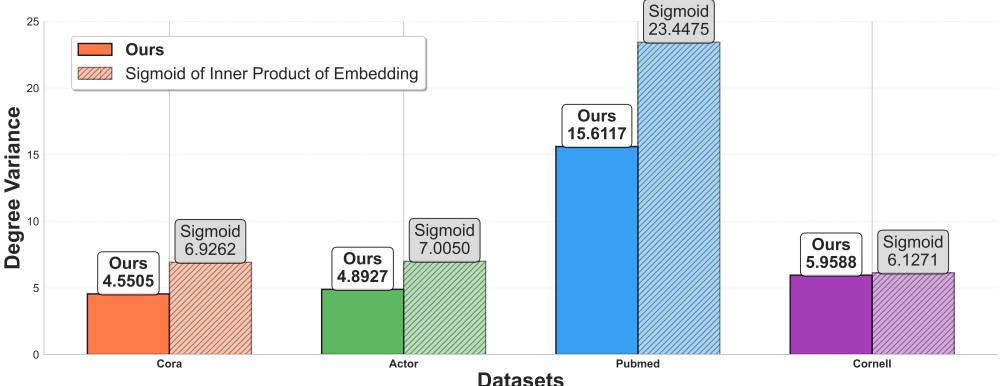

Figure 10: Degree variance comparison of two sampling methods on four datasets. The solid columns represent the method we proposed, and the diagonal columns represent the sampling method based on the sigmoid value of the inner product between node embeddings.

## I   More Discussions of Experiments of Efficiency Comparisons

In Table 2, we present a comparison of the running times of different models across multiple datasets. These datasets encompass graph-structured data of varying scales and complexities, facilitating a comprehensive evaluation of each model's efficiency. As observed in Table 2, our method achieves the fastest running times across all evaluated datasets. Specifically, several GTs (GT, SAN, Graphormer) encounter out-of-memory issues on larger graphs like Flickr and ArXiv, while our method scales efficiently to these datasets. For computationally intensive methods like ANS-GT and GOAT, which require over 1 second per epoch even on small graphs and escalate to 20-50+ seconds on larger datasets, our method maintains sub-0.02 second runtime on small graphs and only 0.246 seconds on ArXiv. Even when compared to recent efficient GTs like SGFormer and DIFFormer, or deep GNNs like GCNII, our method demonstrates 2-5× speedup across different dataset scales. This outcome aligns with the theoretical complexity analysis conducted in Sec. 4.4, affirming that our method holds a computational complexity advantage in practice.

## J   Supplementary Experiments

In this section, we provide some supplementary experimental results for a better understanding of our graph learning paradigm and framework.

### J.1   Hyper-Parameter Studies

In our experimental implementation, there are several key hyper-parameters that have a significant impact on the model's performance. First, there is the tree quantity parameter $N_T$, which determines the number of trees included in the forest model. Under a certain quantity, the prediction performance of the model will improve as the value of $N_T$ increases. However, when $N_T$ reaches a certain quantity, the performance improvement will stop or even decline, and the computational complexity will also increase accordingly. Therefore, it is necessary to find the optimal balance point between model performance and computational efficiency.

Secondly, there are the parameters $\beta_1$ and $\beta_2$. These two parameters jointly adjust the contribution weights of different sub-models in the All-In-One Local Layer. $\beta_1$ controls the influence intensity of the first GCN, while $\beta_2$ regulates the contribution degree of the attn model. By adjusting the proportional relationship of these two parameters, the optimal fusion among different model components can be achieved, thereby improving the overall prediction accuracy.

Finally, there is the residual parameter $\gamma$. It is responsible for balancing the importance weights between the Forest Layer and the Local Layer. A larger $\gamma$ value will enhance the influence of the Forest Layer, making the model focus more on long-range information. And the smaller $\gamma$ value will highlight the role of the Local Layer, allowing domain information to have a more significant proportion.

### J.1.1 THE STUDY ON THE NUMBER OF TREES, *i.e.*, $N_T$

In the experiment on the number of trees K, we tested the impact of different numbers of trees on the model performance on four different datasets. To ensure the fairness and reliability of the results, we conducted experiments on five random seeds and calculated the mean and variance, which are all reflected in Fig. 11.

The experimental results show that the number of optimal trees on different datasets is not fixed, but it often falls within the range of 6 to 10. We find that there is an obvious trade-off relationship in the number of trees: When the number of trees is too small, the model faces the problem of insufficient information richness and is unable to fully capture the complex patterns in the data; When the number of trees is too large, it is difficult to ensure that each tree maintains a high quality standard. Trees of lower quality will aggregate incorrect information to nodes, thereby leading to a decrease in the overall prediction accuracy. It can be clearly observed from the line graph of the experiment that the performance curves of all datasets show a similar trend: The model performance gradually improves with the increase in the number of trees, and begins to decline after reaching the peak. It is worth noting that while the performance declines, the variance of the model's prediction results often increases accordingly, which further confirms that the negative impact of too many low-quality trees can reduce the robustness of the model.

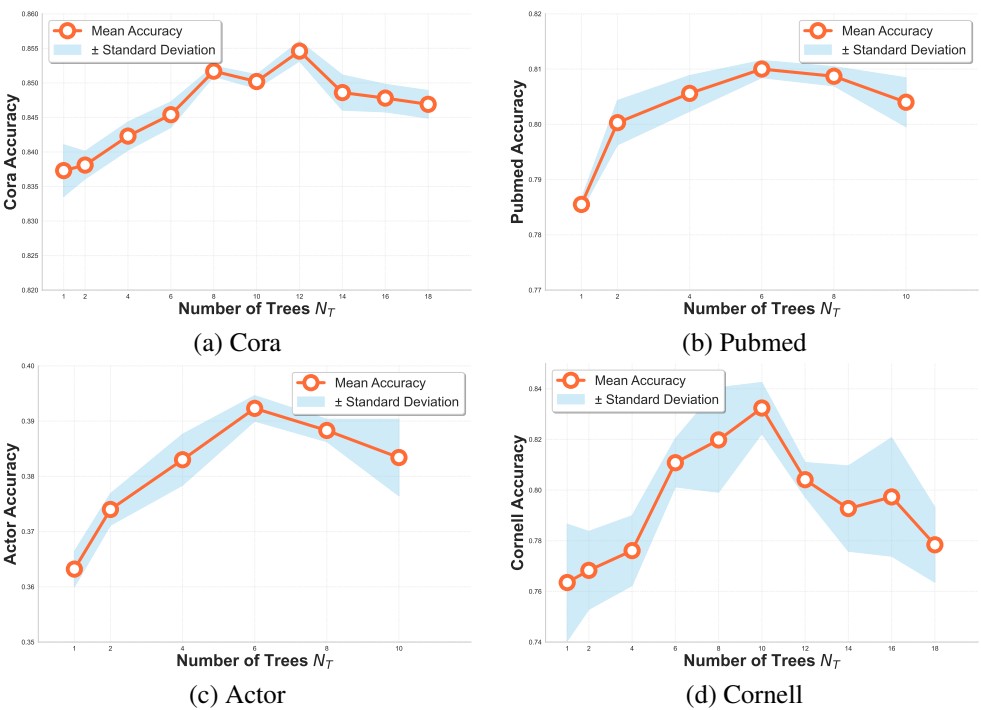

Figure 11: Model performance with varying number of trees.

### J.1.2 THE STUDY ON THE HYPER-PARAMETERS OF LOCAL SUBMODULE, *i.e.*, $\beta_1$, $\beta_2$

In the in-depth study of the parameters $\beta_1$ and $\beta_2$, we discovered an important phenomenon: Different types of datasets show obvious preference differences for the architecture design of the Local Layer, and this preference pattern directly reflects the inherent graph structure characteristics of each dataset.

Two completely different optimization modes can be clearly observed from the experimental results of the Fig. 12. For the two homophilous graph datasets, PubMed and Cora, the model performance shows a significant positive correlation with the value of $\beta_1$ - the higher the proportion of $\beta_1$, the better the model performance. It is particularly notable that when $\beta_1$ and $\beta_2$ approach 0 simultaneously, the performance of both datasets has decreased significantly. The fundamental reason for this phenomenon lies in that at this time, the Local Layer degenerates into the traditional multi-layer Perceptron (MLP) structure, and MLP is unable to effectively handle the strong correlations and neighborhood dependencies among nodes in homophilous graphs.

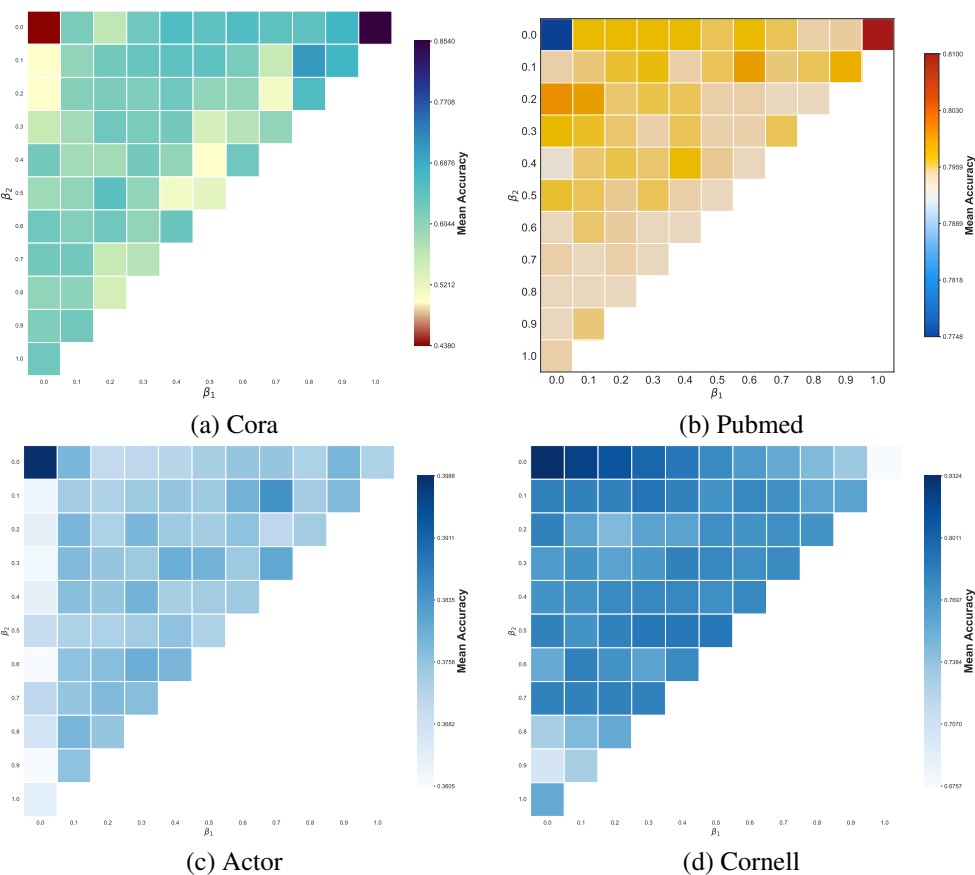

Figure 12: Heat map of the influence of parameters $\beta_1$ and $\beta_2$ on the model performance of the four datasets. (a) Cora dataset, (b) PubMed dataset, (c) Actor dataset, (d) Cornell dataset. The depth of the color represents the average accuracy rate. The horizontal axis is $\beta_1$ and the vertical axis is $\beta_2$.

In sharp contrast, the two heterophilous graph datasets, Cornell and Actor, exhibit completely opposite optimization characteristics. Both of these two datasets achieve the best performance under the condition that $\beta_1$ and $\beta_2$ are both 0, fully demonstrating that the heterophilous graph structure is more suitable for using MLP as the architectural choice for the Local Layer. This is because in heterophilous graphs, adjacent nodes often belong to different categories. Traditional graph neural networks are prone to interference from neighborhood noise, while MLP can better capture the feature information of the nodes themselves.

Further analysis also revealed the differences in sensitivity among the datasets: The Actor dataset was more sensitive to the changes in the ratios of $\beta_1$ and $\beta_2$ compared to the Cornell dataset, and the

performance degradation was more significant when the parameters deviated from the optimal values. Relatively speaking, our method shows stronger robustness on the Cornell dataset and has a larger fault-tolerant range for parameter selection.

### J.1.3 THE STUDY ON THE RESIDUAL COEFFICIENT, *i.e.*, $\gamma$

The parameter $\gamma$, as a key parameter in the model, is responsible for balancing the weight distribution between the remote information (Forest Layer) and the short-range information (Local Layer). Through systematic experiments on the dataset, we observed consistent and significant forms and patterns of performance changes.

Based on the observation of Fig. 13, whether it overly relies on short-range information or overly emphasizes long-range information, it will lead to a significant decline in the model performance. The performance curves of all datasets show a similar inverted U-shaped trend (with different peaks) : As the $\gamma$ value increases, the model performance gradually improves first, reaches a peak at a certain specific proportion, and then begins to decline. This phenomenon fully proves that a balance point needs to be found between long-range information and short-range information in order to further exert the performance of the model. However, even if the overall trend remains consistent, we can

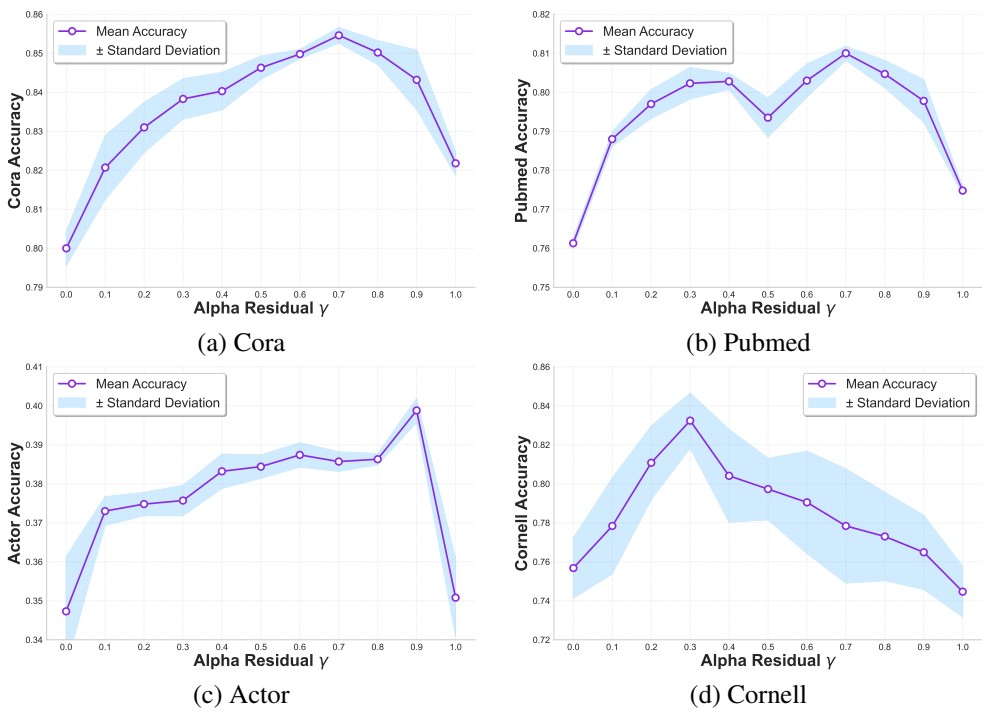

(a) Cora  (b) Pubmed

(c) Actor  (d) Cornell

Figure 13: The curves of the model performance varying with the parameter $\gamma$ on the four datasets. (a) Cora dataset, (b) PubMed dataset, (c) Actor dataset, (d) Cornell dataset. The purple solid line represents the average accuracy rate, and the blue shaded area represents the range of standard deviations.

clearly observe that different datasets have different degrees of dependence on long-range information and short-range information. This difference reflects the inherent graph structure characteristics of each dataset. For example, the Actor dataset shows a stronger dependence on remote information compared to other datasets, and its optimal $\gamma$ value is significantly biased towards remote information. This preference may stem from the fact that the long-distance dependency relationships among nodes in the Actor dataset are more important, and a tree structure is needed to capture a broader global pattern.

### J.1.4    THE STUDY ON THE HIDDEN DIMENSION HYPER-PARAMETER, *i.e.*, $d$

In this subsection, we investigate the sensitivity of our model to the hidden dimension $d$ by conducting experiments with four different settings: $d = 128$, $d = 256$, $d = 512$, and $d = 1024$. Across all three datasets (*i.e.*, Cora, PubMed, and Citeseer), the accuracy curves remain extremely stable, with fluctuations within only $0.5\% - 1.0\%$. This consistency demonstrates that our model is highly insensitive to the variations of the hidden dimension $d$ and does not rely on delicate hyperparameter tuning, making it robust and easy to deploy in practical, large-scale scenarios.

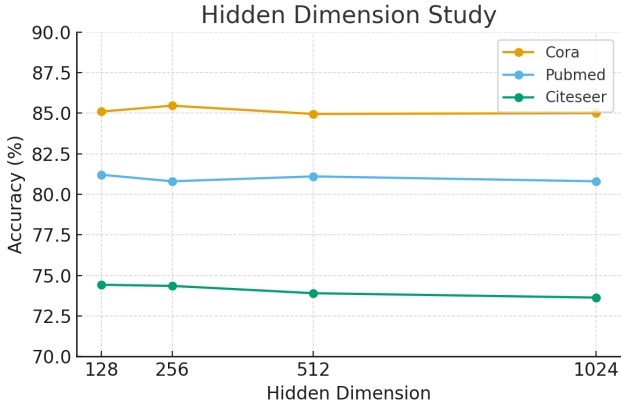

Figure 14: Hidden Dimension Study

## J.2 Interpretability Studies

Based on the theoretical proof of Theorem 2, we have verified that trees sampled from our tree distribution can obtain theoretical guarantees on the first-order homogeneity. In this section, we hope to further conduct in-depth research and explain the internal mechanism of the model performance improvement, and reveal the essential principle of performance growth by directly analyzing the quantity of long-range information aggregated by nodes and the validity of this information.

To quantify the benefit of this kind of information aggregation, we introduce a metric of **global homogeneity**. This metric can be directly achieved through our Forest Layer - the calculation can be completed simply by passing the one-hot encoding of the node as input into the network.

It can be clearly observed from the experimental results of Fig. 15 that in all scenarios, the trees sampled from our distribution exhibit significantly higher remote homophilous information aggregation capabilities. This phenomenon indicates that our method can capture and utilize the long-distance information in the graph more effectively, thereby aggregating into more feature representations with discriminative value. It is precisely because of this enhanced remote information aggregation capability that our model can achieve better performance on various datasets.

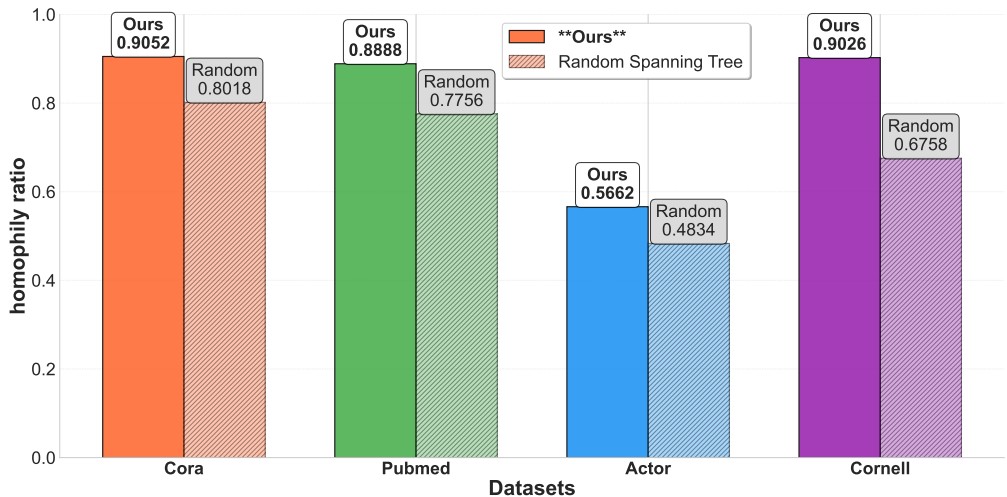

Figure 15: homophily ratio comparison based on different sample method

This analysis not only verifies and explains the effectiveness of our method from an empirical perspective, but also reveals the fundamental reason for the performance improvement from the perspective of information theory - that is, the more full utilization of valuable remote information is achieved through the improved sampling strategy.

## J.3 Node Embedding Visualizations

We plot some node embeddings produced by ours, SGC, and original features in Fig. 16. Observed from Fig. 16, our framework can obtain a much clearer gap between node embeddings from different node classes, *i.e.*, between-class embedding margins. It partly explains the performance gain of ours against some counterparts.

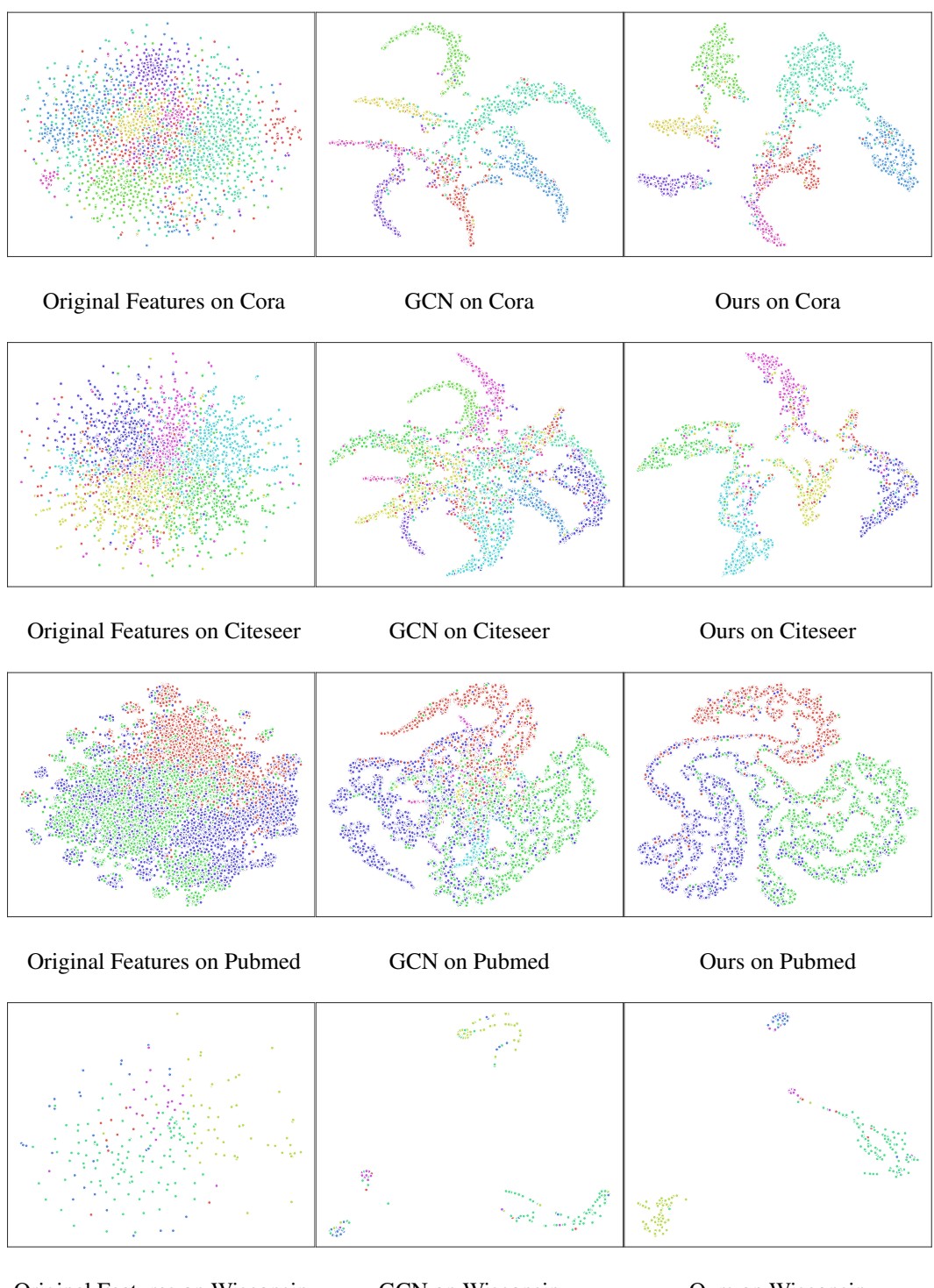

Figure 16: Scatter plots of original features and embeddings output via GCN and our method on real-world benchmarks Cora, Citeseer, Pubmed, and Wisconsin.

## J.4 Noisy-Feature Node Classification Tasks

To evaluate model robustness, we introduce a task called Noisy-Feature Node Classification, where we inject Gaussian noise (5%, 10%, 15%) into the input node features and observe how classification accuracy varies under increasing perturbation. We visualize the results in Fig. 17 and Fig. 18 to facilitate a clear comparison. By comparing accuracy trends across different models, we assess whether our method maintains performance more effectively than existing baselines under node feature noise. Across all three benchmark datasets -CORA, PUBMED, and CITESEER - our framework exhibits significantly stronger robustness to feature noise with various levels (*i.e.*, 5%, 10%, 15%) compared with existing baselines. While all comparative methods show rapidly degrading accuracies as noise level increases, our framework consistently maintains a clear and stable performance superiority. For example, on CORA and PUBMED, the advantage is clear: even at the highest noise level (15%), our model retains $0.631$ and $0.646$ accuracy respectively, far above the second-best models, whose performance typically falls into the $[0.37, 0.49]$ range. This indicates that our method is substantially less sensitive to perturbations in node features. The trend is similar in CITESEER, where accuracy remains above $0.44$, while other models are much lower, and NodeFormer collapses severely due to its instability under noise. The bar plots (see Fig. **??**) clearly illustrate that although all models naturally decline with increasing noise, ours declines much more gracefully, preserving a significantly higher accuracy. Overall, the results demonstrate that the proposed method possesses superior noise resilience and thus can maintain high discriminative power with valuable knowledge even when node features are heavily corrupted.

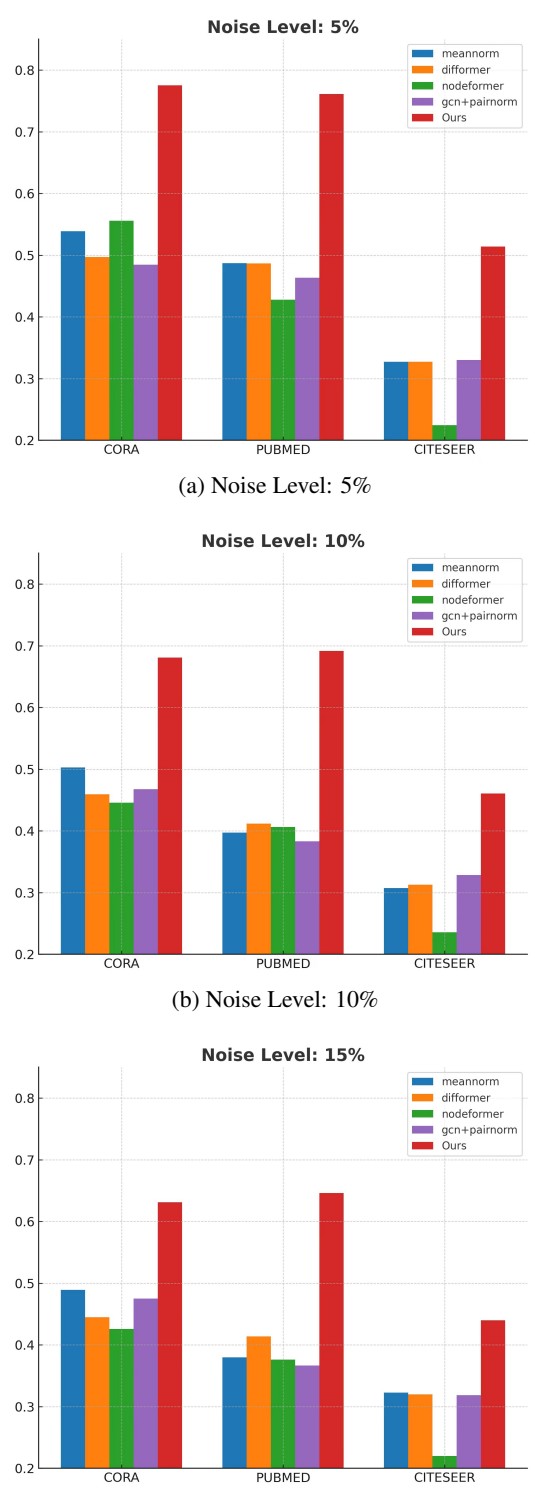

Figure 17: Robustness Comparison under Feature Noise of various levels on Cora, Pubmed, and Citeseer

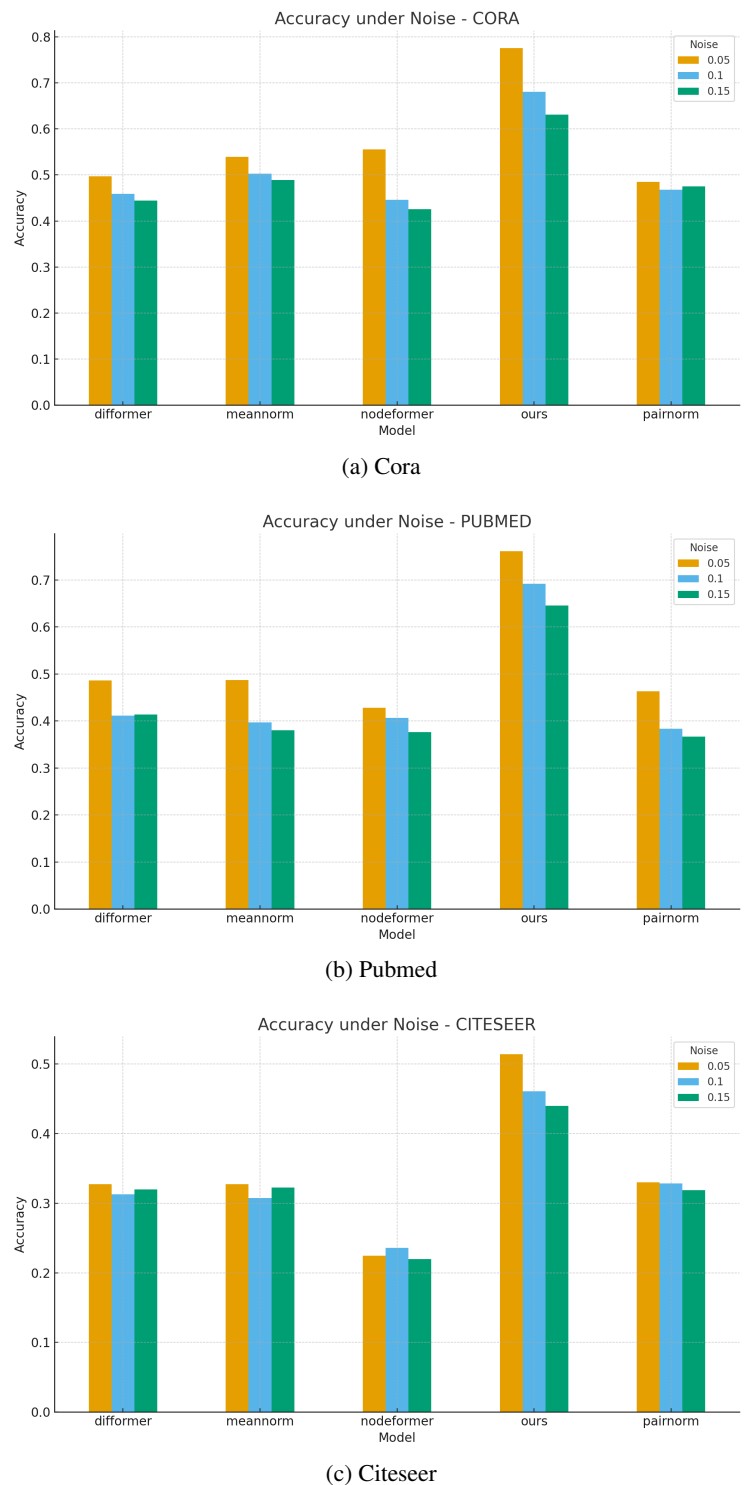

Figure 18: Robustness Comparison under Feature Noise of various levels on Cora, Pubmed, and Citeseer

## J.5 EXPERIMENTS ON LARGER-SCALE GRAPH BENCHMARK

In this subsection, we supplement experiments on a larger-scale graph dataset, *i.e.*, AMiner-CS, and report the results in Fig. 19 and Tab. 5, including final performance comparisons as well as the efficiency comparisons. Across all comparative baselines, our framework achieves the best classification accuracy, exceeding the second-best method by a significant margin. In addition to its superior predictive performance, our approach is also substantially faster, requiring only $0.152$ seconds per epoch, which is notably lighter than baseline models such as Difformer and dramatically more efficient than deep GNN methods like Meannorm and Pairnorm. These results collectively demonstrate that our method effectively breaks the challenging trade-off between accuracy and computational efficiency, making it highly suitable for large-scale graph learning scenarios.

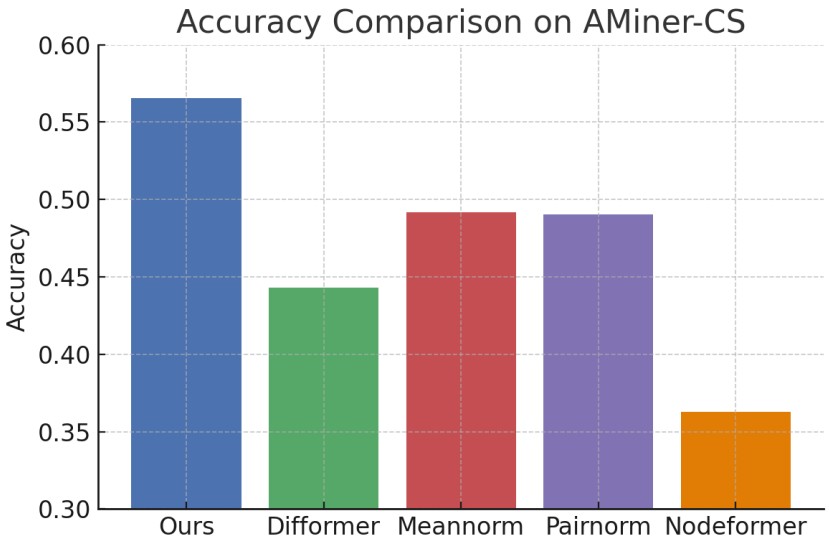

Figure 19: Accuracy comparison on the AMiner-CS dataset.

Table 5: Training speed comparison on AMiner-CS (seconds per epoch).

| Method | Avg Epoch Time (s) |
|---|---|
| Difformer | 0.2524 |
| Meannorm | 1.1620 |
| Pairnorm | 1.2253 |
| Nodeformer | 0.3771 |
| Ours | **0.152** |

## J.6 EXPERIMENTS ON HIGHLY CONNECTED GRAPH BENCHMARK

In this subsection, we supplement experiments on a dense graph, *i.e.*, Dense Cora, and report the results in Fig. 20. Dense Cora is a highly connected graph constructed by adding 100 extra edges per node while keeping comprehensive homophily rate. Observed from Fig. 20, the accuracy first increases slightly when the number of trees $N_T$ grows from 10 to 12, but then decreases slightly as the number of trees $N_T$ is further enlarged to 15 and 20. In other words, we observe that the addition of extra trees can further improve performance compared to using only a few. However, introducing too many trees cannot improve performance and may even slightly degrade it, as they introduce redundancy and would increase the risk of overfitting or over-smoothing issues. Therefore, even dealing with a very densely connected graph, a limited number of trees would be enough to encode the essential structural knowledge, without the need to introduce too many trees.

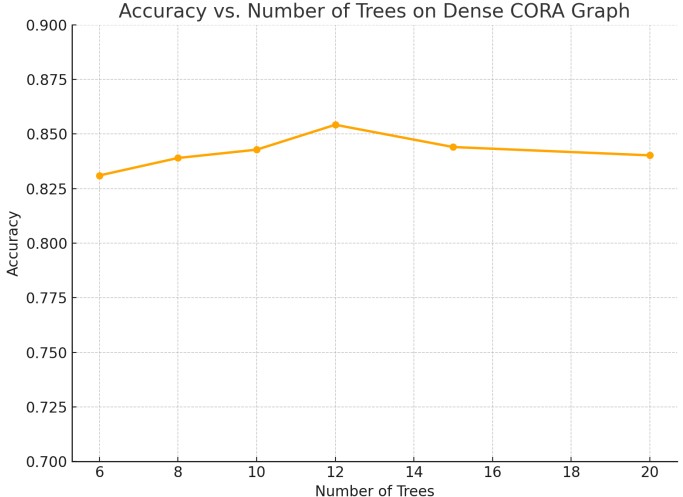

Figure 20: Accuracy vs. Number of Trees on the Dense cora Graph

## J.7 GRAPH CLASSIFICATION TASKS

In this subsection, we supplement some experiments on graph classification tasks on the ENZYMES dataset. We report the results in Fig. 21, with comparisons against seven baselines. While our method is primarily designed for node classification, we show that it still achieves the best performance when directly applied to the graph classification task, surpassing all attribute-based, structure-based, and kernel-based graph classification baselines. This demonstrates that the learned embeddings generalize effectively from node-level supervision to whole-graph-level prediction, indicating strong adaptability, generality, and robustness of our model.

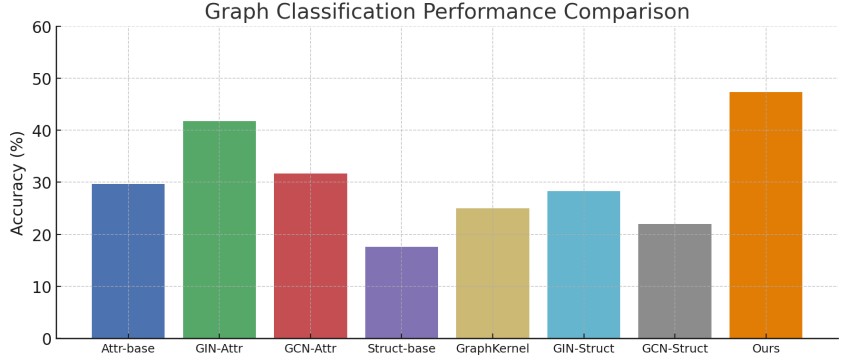

Figure 21: Graph Classification Accuracy Comparison on the ENZYMES Dataset

### J.8 COMPARISONS WITH PATH-BASED GNNs

While both our work and some path-based methods (such as PAIN Graziani et al. (2023) and PathNNs Michel et al. (2023)) can expand GNNs' receptive fields, our forest-based paradigm is fundamentally distinct. The path-based works rely on enumerating fixed-length paths (with exponential complexity like $O(nD^l)$) and stacking layers (with an additional factor $L > 2$) to approximate global coverage. These factors (path numbers, path lengths, and layer numbers) make path-based graph learning inherently suffer from a severe trade-off between global coverage comprehensiveness and computational cost. In sharp contrast, our forest-based paradigm enables native pairwise node interaction in a single layer. It achieves full global coverage in linear time, thereby avoiding complex structural encoding, layer stacking, and path-length constraints.

In this subsection, we supplement some experimental comparisons with some path-based GNN baselines on the Cora, Citeseer, and Pubmed datasets. We report the results in Fig. 22, with comparisons against path-based baseline. Compared with PAIN Graziani et al. (2023), our method consistently achieves higher classification accuracy on all three citation networks. On Cora, Pubmed, and Citeseer, our model improves the performance from 81.44% to 85.46%, from 79.14% to 81.20%, and from 71.86% to 74.42%, respectively, demonstrating a clear and stable advantage over PathGNN across different datasets.

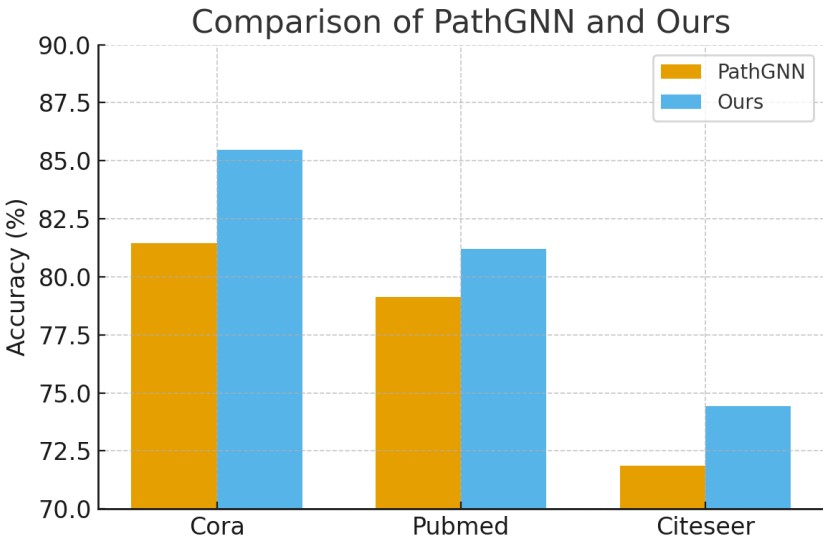

Figure 22: Comparison of PathGNN and Ours on Cora, Pubmed, and Citeseer.

### J.9 COMPARISONS WITH HETEROPHILY GNNS

In this subsection, we supplement some experimental comparisons with some Heterophily-Oriented GNN baselines on the Actor, Cornell, Texas, and Wisconsin datasets. We report the results in Tab. 6. From this figure, we can clearly see that our method achieves the highest accuracy on all four heterophily datasets, outperforming recently proposed heterophily-specific GNNs such as ADPA Sun et al. (2024), GESN Tortorella & Micheli (2022), and HiGNN Zheng et al. (2024). The improvements are particularly large on Texas and Wisconsin, where our c reaches $91.89\%$ and $86.27\%$, substantially surpassing the best existing results and demonstrating its strong capability of capturing informative structural patterns even under severe heterophily. These results confirm that our approach generalizes effectively across diverse heterophilous scenarios and consistently provides state-of-the-art performance.

Table 6: Performance comparison on heterophily datasets (Actor, Cornell, Texas, Wisconsin). The best result in each column is shown in **bold**.

| Method | Actor | Cornell | Texas | Wisconsin |
|---|---|---|---|---|
| ADPA Sun et al. (2024) | $38.8 \pm 0.3$ | $82.9 \pm 3.0$ | $83.8 \pm 2.7$ | $81.6 \pm 3.5$ |
| GESN Tortorella & Micheli (2022) | $34.56 \pm 0.76$ | $81.14 \pm 6.00$ | $84.31 \pm 4.44$ | $83.33 \pm 3.81$ |
| HiGN Zheng et al. (2024) | $37.21 \pm 1.35$ | $80.00 \pm 4.62$ | $86.22 \pm 4.67$ | $85.88 \pm 3.18$ |
| **Ours** | $\mathbf{39.88 \pm 0.43}$ | $\mathbf{83.24 \pm 2.02}$ | $\mathbf{91.89 \pm 0.0}$ | $\mathbf{86.27 \pm 0.0}$ |

## J.10    COMPARISONS WITH OTHER RANDOM TREE-BASED GNNS

We find a recent work, *i.e.*, Bonchi et al. (2025), has some similarities with ours, but we are fundamentally different. Bonchi et al. (2025) also introduces the tree structures to the graph learning domain. However, it is different from our graph learning paradigm in the following perspectives:

(1) Core ideas and main motivations: It introduces random trees mainly to accelerate the local GNN aggregations, by leveraging the tree sparsity to reduce the average number of neighbors. In contrast, we aim to break the trade-off between complexities and comprehensive global aggregations, i.e., providing insights on how to conduct all pairwise node interactions (achieving global coverage) while significantly reducing running cost compared to traditional graph learning paradigms (with only linear complexities), which is more challenging and needs novel fundamental revolutions.

(2) Techniques to sample and utilize trees are different: It samples trees from a uniform distribution, while our technical framework samples trees from a distribution that theoretically biases towards homophily, facilitating beneficial node knowledge propagations in a tree. It deals with such trees via linearization based on path splits or a straightforward depth-first search (DFS) visit order, which reduces informative neighbors for nodes in a tree and risks introducing noisy neighbors. Moreover, they focus more on local aggregation and still require layer stacking to cover global receptive fields. Yet, our work proposes a powerful tree aggregator to explicitly address knowledge propagation along tree paths, achieving global coverage in a single layer without information loss, while keeping efficiency.

Furthermore, we supplement some comparative experiments and empirically find that our technical framework consistently achieves better results compared to GERN-GCNBonchi et al. (2025). The results are reported in Fig. 23.

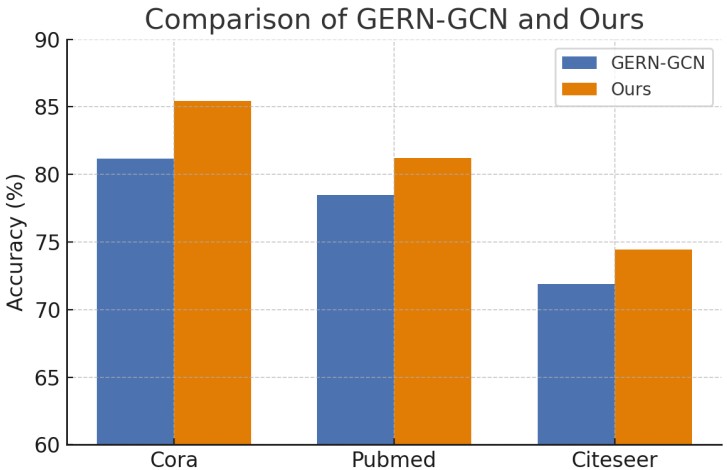

Figure 23: Comparison of GERN-GCN and Ours

We can observe that our method significantly outperforms GERN-GCN across all three datasets. On Cora, our model achieves an accuracy of $85.46\%$, while GERN-GCN achieves only $81.17\%$. Similarly, on Pubmed and Citeseer, our method shows clear superiority, with accuracy improvements of approximately $3.0\%$ and $2.5\%$ respectively. These results highlight that our approach not only surpasses GERN-GCN in accuracy but also exhibits strong generalization across diverse datasets.

## K   MORE EXPERIMENTAL DETAILS

In this section, more details of the experiments are provided to complement the experiments in the main text, including the experimental environments or platforms, dataset descriptions, and the specific hyper-parameter configurations, searching strategies, and searching space.

### K.1   EXPERIMENTAL ENVIRONMENTS, PLATFORMS, AND TOOLS

We implement it via Pytorch Paszke et al. (2019) and optimize it with Adam Optimizer Kingma & Ba (2015). All experiments are performed on an Ubuntu system with a single NVIDIA RTX $A6000$ GPU (48GB Memory) and 32 AMD EPYC 7543 CPUs.

### K.2   DATASET DESCRIPTIONS AND EXPERIMENTAL SETUP

We summarize some important details of all the graph benchmarks in Tab. 7 utilized for evaluation, including the specific splits. We adopt the standard splits similar to GCN Kipf & Welling (2017). For large-scale graph datasets Flickr and Arxiv, we randomly generated 20 class-balanced data splits to ensure statistical robustness and report the average results across all splits. This multi-split evaluation approach mitigates potential bias from single data partitions and provides more reliable performance estimates. To guarantee fair comparison, all baseline methods were evaluated under identical experimental conditions, including the exact same 20 data splits, consistent hyperparameter search spaces, and uniform optimization procedures. This standardized evaluation protocol ensures that performance differences reflect genuine algorithmic capabilities rather than experimental variations. Our comprehensive comparison framework eliminates potential confounding factors, enabling objective assessment of each method's true performance on these challenging large-scale benchmarks. Notably, in semi-supervised settings, we have only limited labeled nodes since the labels are quite expensive to obtain in the real world, which is quite significant in our world, since collecting so many data samples is already challenging enough, not alone collecting enough labels. In our cross-dataset rank analysis, we employed a fair ranking methodology to handle Out-of-Memory (OOM) scenarios. For algorithms without OOM issues, we calculated their average ranking across all nine datasets. When algorithms encountered OOM errors due to their inherently high space complexity, we excluded those specific datasets from their ranking calculation to ensure fair performance comparison. For example, if an algorithm experienced OOM on 2 out of 9 datasets, its final ranking was based on the average across the remaining 7 datasets. While OOM represents a fundamental scalability limitation of certain architectures, this approach prevents hardware constraints from skewing our comparative analysis, focusing instead on actual algorithmic performance.

Table 7: Dataset statistics of nine real-world benchmarks with their splits

| Dataset | Nodes | Edges | Ave.Degree | Features | Classes | train/val/test |
|---------|-------|-------|-----------|----------|---------|----------------|
| Cora | 2,708 | 5,429 | 4.0 | 1,433 | 7 | 140/500/1000 |
| Citeseer | 3,327 | 4,732 | 2.84 | 3,703 | 6 | 120/500/1000 |
| PubMed | 19,717 | 44,338 | 4.5 | 500 | 3 | 60/500/1000 |
| OGBN-ArXiv | 169,343 | 1,166,243 | 13.77 | 128 | 40 | 800/800/167743 |
| Texas | 183 | 309 | 3.38 | 1,703 | 5 | 87/59/37 |
| Wisconsin | 251 | 499 | 5.45 | 1,703 | 5 | 120/80/51 |
| Cornell | 183 | 295 | 3.22 | 1,703 | 5 | 87/59/37 |
| Actor | 7,600 | 33,544 | 8.83 | 931 | 5 | 3648/2432/1520 |
| Flickr | 89,250 | 899,756 | 10.08 | 500 | 7 | 140/140/88970 |

### K.3 HYPER-PARAMETERS

We tune hyper-parameters via a two-stage strategy. For the first stage, we treat a tree as a hyper-parameter and we tune all of them together. In every stage, we choose the hyper-parameters according to their best validation performance. After the first stage, we enter the second tuning stage. We select the best trees based on the best validation results, and then fix the trees and tune the other hyper-parameters. The search spaces of all meaningful hyper-parameters are listed in Tab. 9, and we omit some unimportant ones because they are actually robust to model performance. For every kind of experiment, we search in the space randomly for 200 times in total. Additionally, for better reproducibility, we report all hyper-parameters configurations used by *Ours* for comparative experiments in Tab. 8.

Table 8: The Hyper-parameter Configurations of *Ours* for semi-supervised node classification tasks on nine public graph benchmarks.

| Dataset | Hyper-parameter Configurations |
|---|---|
| Cora | lr $= 0.01$, epochs $= 50$, $d = 256$, $N_T = 12$, dropout $= 0.9$, weight_decay $= 0.85$, $\gamma = 0.7$, $\beta_1 = 1.0$, $\beta_2 = 0.0$, $K_L = 2$ |
| Citeseer | lr $= 0.005$, epochs $= 100$, $d = 128$, $N_T = 12$, dropout $= 0.9$, weight_decay $= 0.75$, $\gamma = 0.7$, $\beta_1 = 1.0$, $\beta_2 = 0.0$, $K_L = 2$ |
| Pubmed | lr $= 0.001$, epochs $= 100$, $d = 128$, $N_T = 6$, dropout $= 0.6$, weight_decay $= 0.0001$, $\gamma = 0.7$, $\beta_1 = 1.0$, $\beta_2 = 0.0$, $K_L = 2$ |
| Actor | lr $= 0.01$, epochs $= 90$, $d = 128$, $N_T = 5$, dropout $= 0.9$, weight_decay $= 0.0$, $\gamma = 0.9$, $\beta_1 = 0.0$, $\beta_2 = 0.0$, $K_L = 1$ |
| Cornell | lr $= 0.01$, epochs $= 60$, $d = 256$, $N_T = 15$, dropout $= 0.7$, weight_decay $= 0.001$, $\gamma = 0.3$, $\beta_1 = 0.0$, $\beta_2 = 0.0$, $K_L = 2$ |
| Texas | lr $= 0.005$, epochs $= 100$, $d = 256$, $N_T = 5$, dropout $= 0.5$, weight_decay $= 0.00001$, $\gamma = 0.6$, $\beta_1 = 0.4$, $\beta_2 = 0.0$, $K_L = 2$ |
| Wisconsin | lr $= 0.01$, epochs $= 100$, $d = 128$, $N_T = 5$, dropout $= 0.6$, weight_decay $= 0.0$, $\gamma = 0.1$, $\beta_1 = 0.2$, $\beta_2 = 0.8$, $K_L = 1$ |
| Ogbn-Arxiv | lr $= 0.0005$, epochs $= 10$, $d = 256$, $N_T = 4$, dropout $= 0.8$, weight_decay $= 0.0$, $\gamma = 0.5$, $\beta_1 = 0.4$, $\beta_2 = 0.6$, $K_L = 2$ |
| Flickr | lr $= 0.01$, epochs $= 30$, $d = 128$, $N_T = 5$, dropout $= 0.3$, weight_decay $= 0.00001$, $\gamma = 0.6$, $\beta_1 = 1.0$, $\beta_2 = 0.0$, $K_L = 2$ |

Table 9: The Hyper-parameter Search Spaces.

| Hyper-parameters | Hyper-parameter Search Spaces |
|---|---|
| lr | 0.01, 0.001, 0.0005 |
| epochs | linspace(10, 110, 10), 200 |
| $d$ | 64, 128, 256 |
| $N_T$ | 4, 5, 6, 8, 10, 12, 15 |
| dropout | linspace(0.1, 1, 0.1) |
| weight_decay | linspace(0.6, 0.95, 0.05), 0.0, 0.001, 0.0001, 0.00001 |
| $\gamma$ | linspace(0.1, 1, 0.1) |
| $\beta_2$ | linspace(0.1, 1, 0.1) |
| $\beta_1$ | linspace(0.1, 1, 0.1) |
| $K_L$ | 1, 2 |

## L    Details of Prediction-based Graph Augmentation

As mentioned in Sec. 4.2 and Sec. F.1 of Appn., we introduce a simple graph augmentation to support our framework. The motivation is basic: we expect to make the vanilla graph $G$ become connected, and thus we can effectively and conveniently sample trees on the augmented variant. Besides, we also find that it can also improve the NHCC value of the graph (Sec. H.2).

In our implementation, we adopt a Maximum Inner Product Search (MIPS) between $\mathbb{K}_0$, *i.e.*, node label predictions, which is efficient and easy to implement via the Faiss Johnson et al. (2019) library (supporting even billion-scale similarity search with GPUs). We conduct top-k selection for each node (via the metric inner product) on the vanilla graph for simplicity. Further improvement can be developed to, *e.g.*, consider node specialty (*e.g.*, densities or degrees) and shrink the number of added graphs. The added edges are merged into the vanilla edge set and drop those duplicated edges. For reproducibility, we add 12, 10, 15, 15, 10, 5, 8, 8, and 6 edges for each node on datasets ora Citeseer, Pubmed, Actor, Cornell, Texas, Wisconsin, OGBN-Arxiv, and Flick, respectively. We find that the hyper-parameter is not a sensitive hyper-parameter and its influence on the generalizability of final performance is limited. Yet, note that the heterophilous graphs may require a slightly larger value, since, as highlighted above, it can improve the value of the NHCC for the vanilla graph, which has a theoretical connection with the upper bound of the tree quality (Sec. 4.6).

## M    Illustration of Comparisons Between Different Graph Learning paradigms

We provide a figure (*i.e.*, Fig. 24) to illustrate and compare different graph learning paradigms, *e.g.*, neighborhood-based, walk-based, and our forest-based paradigms.

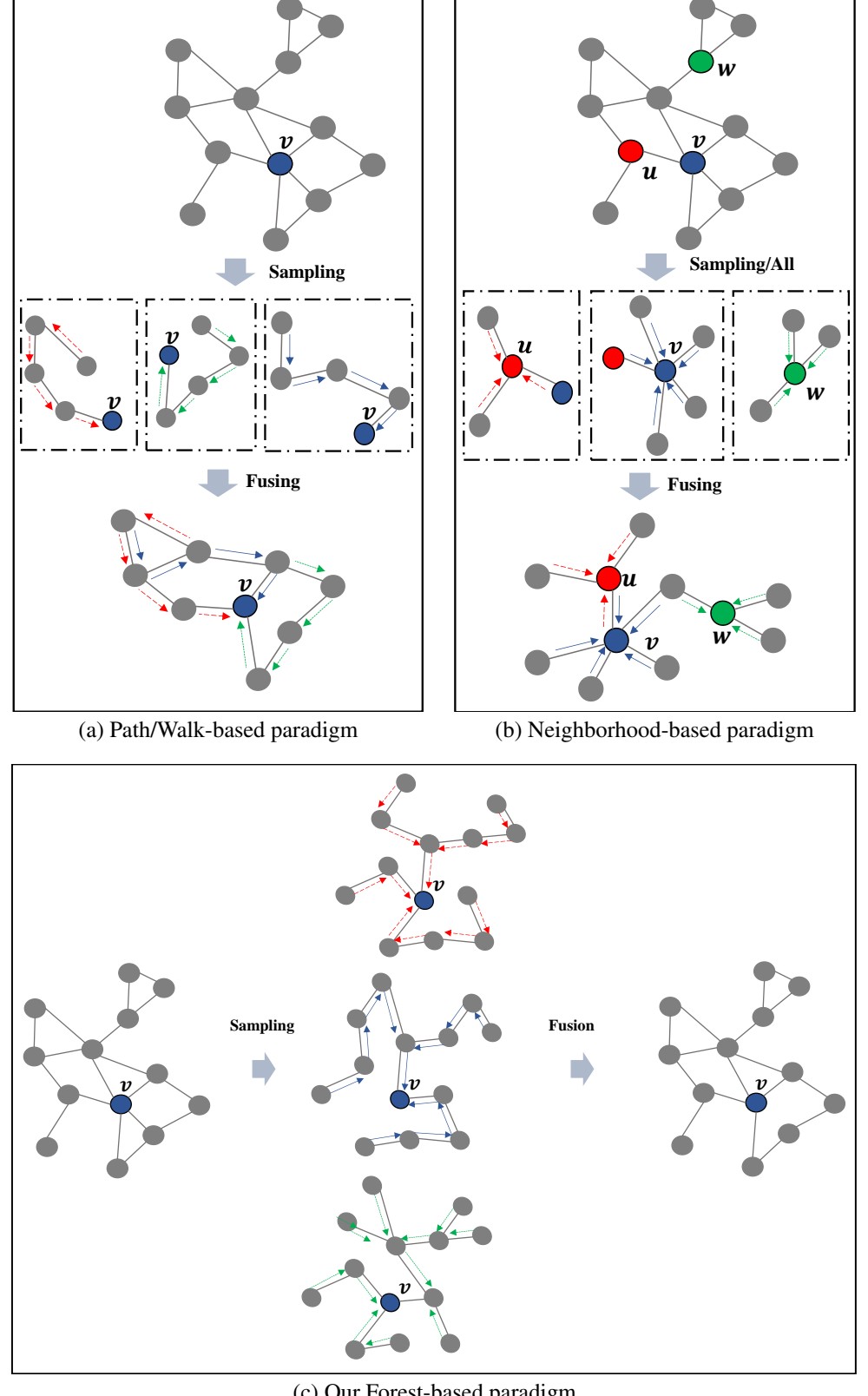

(a) Path/Walk-based paradigm

(b) Neighborhood-based paradigm

(c) Our Forest-based paradigm

Figure 24: Graph Learning paradigm Comparison: (a) Path/Walk-based paradigm; (b) Neighborhood-based paradigm; (c) Our Forest-based paradigm.

Table 10: The results of performance comparison (with the best bolded and the runner-ups underlined).

| Method | Category | Cora | Citeseer | Pubmed | Actor | Cornell | Texas | Wisconsin | Arxiv | Flickr | Avg.R |
|---|---|---|---|---|---|---|---|---|---|---|---|
| MLP | Classic | 58.30 ± 0.11 | 58.68 ± 0.04 | 72.94 ± 0.05 | 35.62 ± 0.18 | 72.70 ± 1.99 | 77.84 ± 1.14 | 79.61 ± 1.01 | 32.84 ± 0.29 | 42.01 ± 0.47 | 14.11 |
| GCN | GNN | 82.06 ± 0.69 | 71.60 ± 0.32 | 79.58 ± 0.26 | 27.88 ± 0.07 | 53.51 ± 1.21 | 69.19 ± 1.48 | 57.25 ± 1.25 | 53.77 ± 0.34 | 38.40 ± 2.23 | 14.89 |
| GAT | GNN | 82.84 ± 0.67 | 72.28 ± 0.72 | 78.52 ± 0.32 | 28.71 ± 0.25 | 55.14 ± 0.48 | 68.65 ± 1.48 | 58.82 ± 2.40 | 55.73 ± 0.46 | 40.32 ± 1.75 | 12.78 |
| GraphSAGE | GNN | 81.40 ± 0.52 | 71.68 ± 0.13 | 78.50 ± 0.44 | 36.24 ± 0.39 | 63.78 ± 2.42 | 75.14 ± 1.21 | 76.08 ± 0.88 | 51.42 ± 0.21 | 41.42 ± 0.89 | 11.00 |
| SuperGAT$_{SD}$ | GNN | 82.70 ± 0.60 | 72.50 ± 0.80 | 81.30 ± 0.50 | 30.18 ± 0.25 | 54.59 ± 1.21 | 69.73 ± 1.21 | 58.04 ± 1.07 | 51.52 ± 0.37 | 36.24 ± 4.15 | 13.22 |
| APPNP | GNN | 84.10 ± 0.43 | 72.14 ± 0.22 | 80.02 ± 0.19 | 33.47 ± 0.38 | 61.08 ± 1.48 | 71.35 ± 1.48 | 65.10 ± 0.88 | 55.60 ± 0.12 | 43.07 ± 0.85 | 9.22 |
| ClusterGCN | GNN | 82.04 ± 0.22 | 70.08 ± 0.26 | 77.26 ± 0.11 | 29.66 ± 0.19 | 49.73 ± 5.27 | 63.24 ± 2.42 | 62.35 ± 3.23 | 53.35 ± 0.12 | 39.58 ± 4.62 | 16.89 |
| GraphSAINT | GNN | 82.00 ± 0.20 | 70.30 ± 0.21 | 77.36 ± 0.24 | 29.55 ± 0.19 | 48.65 ± 1.91 | 63.78 ± 2.42 | 61.96 ± 1.08 | 53.55 ± 0.24 | 35.26 ± 2.73 | 17.67 |
| Pairnorm | DeepGNN | 66.24 ± 1.58 | 44.20 ± 1.23 | 72.12 ± 3.01 | 24.33 ± 1.60 | 40.68 ± 12.89 | 41.08 ± 18.04 | 52.94 ± 11.35 | 54.58 ± 0.04 | 31.41 ± 4.66 | 22.56 |
| Nodenorm | DeepGNN | 80.14 ± 0.51 | 65.74 ± 1.88 | 78.64 ± 0.52 | 29.74 ± 1.01 | 40.00 ± 2.02 | 66.49 ± 2.76 | 48.24 ± 3.64 | 54.22 ± 0.87 | 44.11 ± 2.76 | 16.33 |
| Meannorm | DeepGNN | 79.54 ± 0.56 | 72.16 ± 0.54 | 73.06 ± 3.06 | 25.46 ± 0.00 | 25.41 ± 13.52 | 61.62 ± 2.02 | 52.94 ± 1.24 | 20.37 ± 16.24 | 42.40 ± 2.12 | 19.67 |
| DropEdge | DeepGNN | 81.69 ± 0.91 | 71.43 ± 0.72 | 79.06 ± 0.81 | 26.38 ± 2.41 | 52.97 ± 1.32 | 64.86 ± 0.00 | 60.78 ± 0.00 | 39.23 ± 0.80 | 32.11 ± 8.09 | 18.33 |
| GCNII | DeepGNN | 85.34 ± 0.32 | 73.24 ± 0.78 | 79.88 ± 0.17 | 34.64 ± 0.71 | 74.61 ± 6.48 | 69.19 ± 6.56 | 70.31 ± 4.75 | 51.91 ± 1.50 | 41.79 ± 0.30 | 8.78 |
| ShadowGNN | Deep GNN | 82.32 ± 0.24 | 70.06 ± 0.15 | 77.30 ± 0.20 | 29.45 ± 0.08 | 51.35 ± 3.31 | 64.32 ± 2.96 | 62.35 ± 1.64 | 53.35 ± 0.32 | 37.59 ± 1.88 | 17.00 |
| GT | GT | 77.58 ± 0.22 | 66.96 ± 0.36 | 76.48 ± 0.13 | 37.15 ± 0.40 | 61.62 ± 2.26 | 74.60 ± 1.48 | 71.76 ± 1.75 | OOM | OOM | 15.57 |
| SAN | GT | 77.60 ± 0.23 | 68.64 ± 0.92 | 76.62 ± 0.22 | 37.79 ± 0.20 | 63.24 ± 1.48 | 75.14 ± 2.26 | 77.25 ± 1.75 | OOM | OOM | 13.00 |
| Graphormer | GT | 63.08 ± 0.27 | 61.08 ± 0.04 | OOM | OOM | 62.70 ± 2.26 | 76.76 ± 1.48 | 72.16 ± 0.88 | OOM | OOM | 15.40 |
| ANS-GT | GT | 77.68 ± 0.81 | 64.16 ± 1.16 | 77.98 ± 1.38 | 38.29 ± 0.61 | 74.92 ± 1.86 | 76.22 ± 2.26 | 76.47 ± 1.96 | 41.83 ± 0.62 | 21.86 ± 1.93 | 13.22 |
| Nodeformer | GT | 79.02 ± 0.57 | 69.66 ± 0.13 | 76.06 ± 0.98 | 34.80 ± 0.48 | 68.11 ± 1.21 | 77.84 ± 1.71 | 76.47 ± 4.80 | 39.47 ± 1.08 | 40.31 ± 4.42 | 13.11 |
| NAGphormer | GT | 79.51 ± 0.90 | 67.34 ± 0.80 | 78.32 ± 0.40 | 37.33 ± 0.20 | 63.78 ± 1.48 | 71.89 ± 1.48 | 66.27 ± 0.88 | 52.00 ± 0.29 | 38.59 ± 1.16 | 13.44 |
| GOAT | GT | 83.18 ± 1.27 | 71.99 ± 1.26 | 79.13 ± 0.38 | 37.66 ± 0.51 | 64.32 ± 2.02 | 76.76 ± 2.76 | 73.33 ± 1.57 | 52.46 ± 1.12 | 35.53 ± 3.13 | 9.11 |
| Exphormer | GT | 82.77 ± 1.38 | 71.63 ± 1.19 | 79.46 ± 0.35 | 35.53 ± 0.62 | 62.16 ± 2.42 | 75.68 ± 1.71 | 70.98 ± 2.29 | 41.12 ± 0.69 | 22.79 ± 3.09 | 12.67 |
| SGFormer | GT | 82.38 ± 0.70 | 71.82 ± 0.18 | 80.64 ± 0.52 | 37.80 ± 0.47 | 68.65 ± 2.42 | 78.92 ± 1.21 | 80.00 ± 0.88 | 45.73 ± 1.61 | 40.13 ± 2.49 | 7.22 |
| DIFFormer | GT | 83.32 ± 0.52 | 74.46 ± 0.42 | 78.16 ± 0.32 | 34.51 ± 0.76 | 60.00 ± 1.21 | 68.11 ± 2.26 | 63.92 ± 1.07 | 53.60 ± 0.64 | 44.25 ± 0.86 | 10.56 |
| TDGNN | GT | 85.35 ± 0.49 | 73.78 ± 0.60 | 80.20 ± 0.33 | 32.84 ± 0.76 | 35.68 ± 4.15 | 61.35 ± 2.72 | 46.86 ± 4.76 | OOM | 38.25 ± 1.88 | 15.00 |
| GraphMamba | Mamba | 54.36 ± 2.34 | 58.98 ± 2.39 | 70.90 ± 1.16 | 36.05 ± 0.40 | 74.05 ± 3.24 | 77.29 ± 2.16 | 80.39 ± 1.24 | 33.59 ± 3.50 | 42.30 ± 0.00 | 13.89 |
| Ours | Forest | 85.46 ± 0.29 | 74.42 ± 0.29 | 81.00 ± 0.26 | 39.88 ± 0.43 | 83.24 ± 2.02 | 91.89 ± 0.00 | 86.27 ± 0.00 | 56.47 ± 0.60 | 47.22 ± 1.98 | 1.22 |

---

**Algorithm 3** Algorithm of Block Acceleration of Tree Sampler

---

**Description**: Given a positively weighted directed graph $G$, define a tree distribution conditioned on graph $G$ as $P_G(T)$, with the unnormalized score equaling to the product of all edge weights in a tree (Recall Eq. 2 in the main text). Return a spanning tree of graph $G$ approximately sampled from $P_G(T)$ yet with higher parallelizability via a trick called *Block Acceleration* (Sec. E and Sec. F.3). The key idea is to identify a set of unimportant edges with relatively low scores and distinguish them from other edges by first ignoring and then reconsidering them, which provides a way that first divides the input graph into several blocks, parallelizably processes intra-block edges, and finally adds inter-block edges. Check Sec. F.3 for the line-by-line explanations of this algorithm.

**Input**: a graph $G = (V, E)$ with its edge index $\text{EdgeIndex} \in \mathbb{R}^{m \times 2}$ and its edge weights $\text{EdgeWeights} \in \mathbb{R}^{m \times 1}$ where $m = |E|$ and we also denote $n = |V|$

**Output**: the tree $T \sim P_G(T)$ with $T = (V_T, E_T)$, where $V_T$ and $E_T$ denote its node set and its (undirected) edge set, respectively

**Hyper-Parameters**:    $K_B$, the number of blocks

**Note**: (1) We will call Algorithm 2 with the operator TreeSampler (EdgeIndex, EdgeWeights);

(2) We will call a graph cut technique GraphCut (EdgeIndex, EdgeWeights) (*e.g.*, efficient METIS Library for implementations), which will return a partition solution of the node set $V$ of an input graph $G$ with its edge index and its undirected edge weights;

(3) We denote the operator $\text{LookUp}(S, X) \in \mathbb{N}^{q \times 1}$ with $S \in \mathbb{R}^{p \times k}, X \in \mathbb{R}^{q \times k}$ to find the row index of a row $X_i$ in $S$ ($-1$ for rows not contained), assuming no two rows in $S$ are exactly the same;

(4) We will call two operators $\text{Scatter\_Add}(a, \text{Index})$ and $\text{Scatter\_ArgMax}(a, \text{Index})$ from library torch_scatter, where $\text{Index}[i]$ denotes the class number of the $i$-th row of the matrix $a$, with their definions as follows:

$$\text{Scatter\_Add}(a, \text{Index})[i] = \sum_{j=0}^{|a|-1} \mathbb{I}(\text{Index}[j] = i) \cdot a[j], \tag{77}$$

$$\text{Scatter\_ArgMax}(a, \text{Index})[i] = \text{argmax}_{j: \ 0 \leq j < |a| \ \& \ \text{Index}[j]=i} \ a[j].$$

1: $\text{BlockNo} \leftarrow \text{GraphCut}(\text{EdgeIndex}, \text{EdgeWeights}) \in \mathbb{N}^{n \times 1}$
2: $\text{BnLeft}, \text{BnRight} \leftarrow \text{BlockNo}(\text{EdgeIndex}[:, 0]), \ \text{BlockNo}(\text{EdgeIndex}[:, 1])$
3: **for** each $i \in [1, K_B]$ **do**
4:     $\text{Mask} \leftarrow (\text{BnLeft} = \text{BnRight} \quad \& \quad \text{BnLeft} = i)$
5:     $\text{EdgeIndexBlock}[i], \text{EdgeWeightsBlock}[i] \leftarrow \text{EdgeIndex}[\text{Mask}], \text{EdgeWeights}[\text{Mask}]$
6: **end for**
7: $\text{ZeroMask} \leftarrow (\text{BnLeft} \neq \text{BnRight})$
8: $\text{EdgeIndexBlockZeroVanilla} \leftarrow \text{EdgeIndex}[\text{ZeroMask}]$
9: $\text{EdgeWeightsBlockZero} \leftarrow \text{EdgeWeights}[\text{ZeroMask}]$
10: $\text{EdgeIndexBlockZeroBlockNo} \leftarrow \text{BlockNo}[\text{EdgeIndexBlockZeroVanilla}]$
11: $\text{EdgeIndexBlock}[0], \text{EdgeWeightsBlock}[0] \leftarrow \text{Merge}(\text{EdgeIndexBlockZeroBlockNo}, \text{EdgeWeightsBlockZero})$
12: **for** $i \in [0, K_B]$ **do**
13:     $\text{TreeBlock}[i] \leftarrow \text{TreeSampler}(\text{EdgeIndexBlock}[i], \text{EdgeWeightsBlock}[i])$
14: **end for**
15: $\text{Index} \leftarrow \text{LookUp}(\text{TreeBlock}[0], \text{EdgeIndexBlockZeroBlockNo})$
16: $\text{Index} \leftarrow \text{Index}[\text{Index} \geq 0]$
17: $\text{EdgeIndexBlockZeroBlockNoInTree} \leftarrow \text{EdgeIndexBlockZeroBlockNo}[\text{Index}]$
18: $\text{EdgeWeightsBlockZeroInTree} \leftarrow \text{EdgeWeightsBlockZero}[\text{Index}]$
19: $\text{EdgeSum} \leftarrow \text{Scatter\_Add}(\text{EdgeWeightsBlockZeroInTree}, \text{Index})$
20: $\text{EdgeProbabilities} \leftarrow \text{EdgeWeightsBlockZeroInTree} / \text{EdgeSum}[\text{Index}]$
21: $\text{tmp} \leftarrow \log(\text{EdgeProbabilities}) + Gumbels$, where each element in $Gumbels$ is $x = -\log(-\log(t)), \ t \sim \text{Uniform}(0, 1)$
22: $\text{TreeBlock}[0] \leftarrow \text{EdgeWeightsBlockZeroInTree}[\text{Scatter\_ArgMax}(\text{tmp}, \text{Index})]$
23: $V_T \leftarrow V, \ E_T \leftarrow \text{Concat}\left(\{\text{TreeBlock}[k]\}_{k \in [0, K_B]}\right)$
24: **return** $T \leftarrow (V_T, E_T)$

---

## N    MORE RELATED WORK

In this section, we will provide more discussions on the literature related to our work.

### N.1    GNNs ON HETEROPHILIC GRAPH

While traditional graph neural networks (GNNs) excel at semi-supervised node classification under the homophily assumption, they face challenges in heterophilic graphs—where dissimilar nodes (with different labels) are often connected—due to misleading message aggregation and over-smoothing.Existing heterophilic GNNs can be categorized into three main types: aggregation calibration, graph modification, and other approaches. (1) Aggregation calibration methods optimize message aggregation to mitigate heterophily's negative effects while preserving local topology. H2GCN Zhu et al. (2020) distinguishes between ego-node and neighbor representations, combining node embeddings to balance local and global information. ACMGCN Luan et al. (2021) adaptively mixes different frequency signals via low-pass, high-pass, and identity channels, which successfully separates meaningful information from noise in heterophilic scenarios. (2) Graph modification methods adjust the original graph structure to enhance semantic similarity between connected nodes.Geom-GNN Pei et al. (2020b) constructs structural connections via geometric measurements, preserving topological properties while linking semantically relevant nodes. WRGAT Suresh et al. (2021) learns a new computation graph based on node proximity and local structural similarity, thereby breaking the constraints imposed by the original edges. GloGNN Li et al. (2022) captures node correlations via feature and topology similarity by learning a coefficient matrix, strengthening connections between semantically similar nodes. DIGL Gasteiger et al. (2019b) utilizes generalized graph diffusion (*e.g.*, personalized PageRank) to adjust edge weights, thereby promoting connectivity between nodes with short diffusion distances, aligning with semantic similarity. Other methods adopt alternative techniques to overcome the limitations of traditional message-passing GNNs on heterophilic graphs. GESN Tortorella & Micheli (2022) employs a reservoir computing framework, where node embeddings are generated by an unlearnable recursive message-passing function, thereby avoiding over-smoothing by controlling the Lipschitz constant to effectively encode structural knowledge. ADPA Sun et al. (2024) proposes the AMUD framework to assess how node features interact with directed topology—it helps determine whether the graph should be modeled as undirected or directed—and utilizes hierarchical attention to integrate message information across different scales. HiGNN Zheng et al. (2024) defines *heterophilous information* as the label distribution of each node's neighbors, constructs a new adjacency matrix to connect nodes with similar *heterophilous information*, and fuses this matrix with the vanilla graph structure to improve performance.

## O    LLM USAGE STATEMENT

Large language models were employed in this study exclusively for the purpose of linguistic refinement and stylistic enhancement.

