# OpenReview forum: "Forest-Based Graph Learning for Semi-Supervised Node Classification"
_ICLR.cc/2026/Conference — ICLR 2026 Poster_

### Official Review · Reviewer_vja3 · 2025-10-27

**Soundness:** 3
**Presentation:** 2
**Contribution:** 2
**Rating:** 4
**Confidence:** 5

**Summary:**

The paper proposes Forest-Based Graph Learning (FGL) to reconcile global receptive fields with efficiency in GNNs. Instead of deep stacking or quadratic global attention, FGL models message passing as transport over a small forest of spanning trees: (i) build an augmented graph via pseudo-label kNN; (ii) sample trees using a homophily-guided, weighted Wilson method; (iii) run a two-pass, linear-time tree aggregator (bottom-up then top-down) on each tree; and (iv) fuse normalized per-tree embeddings with a lightweight local module via a residual. Theoretically, better homophily scoring increases expected tree homophily and approaches a structural upper bound.  FGL reports competitive or superior accuracy and favorable runtime on node classification benchmarks.

**Strengths:**

1. The paper proposes a novel graph learning framework—Forest-Based Graph Learning (FGL) that combines global structural modeling with local feature propagation by viewing graphs through a forest of spanning trees to capture hierarchical node dependencies.

2. The technical quality is strong: the objectives and their links to smoothing and spectral properties are clearly derived, and extensive experiments across benchmarks.

3. The writing is clear and well-structured.

**Weaknesses:**

1. The approach largely reads as a straightforward assembly of existing components: pseudo-label kNN augmentation (standard self-training/graph augmentation), weighted Wilson sampling driven by attention scores (classic LERW with learned edge weights), two-pass tree DP (bottom-up/top-down rerooting), and RowNorm + mean + residual fusion. There seems to be little apparent redesign of the modules or new learning objectives beyond composition.

2. The paper primarily evaluates on semi-supervised node classification, leaving unclear whether the forest paradigm generalizes to broader settings (graph classification/regression, link prediction, inductive and dynamic graphs, heterophily extremes, robustness to noise, and scalability in billion-edge graphs). Broader tasks and protocols would better establish practical generality.

3. The method introduces multiple hyperparameters ($\beta_1, \beta_2, \gamma$ etc.) that require careful tuning for different scenarios. The hyper-parameter part implies that performance may be deeply dependent on these parameters, which could make the method difficult to deploy in practice, potentially limiting its practical applicability.

4. The paper does not thoroughly examine when the forest paradigm might underperform. For instance, what happens when node features are noisy or weakly correlated with the graph topology? Does the method perform well on highly dense graphs where local tree-structured neighborhoods may not be able to effectively capture the graph’s global dependency patterns?

**Questions:**

Please address the concerns raised in the weaknesses section.

---

> ### Author Response · Authors · 2025-11-23
>
> *We are deeply grateful to you for the careful and insightful comments on our manuscript.
> Our detailed responses to your questions are outlined below.*
>
> **Weakness 1: assembly of existing components**
>
>
> We would like to clarify that the learning paradigm and particular technical instantiations are fundamentally distinct.
>
> - Our primary contribution is not the specific combination of components, but a new graph learning paradigm that achieves comprehensive pairwise node interactions with linear computational cost—resolving a long-standing trade-off between complexity and global coverage. The current implementation is only one realization of this paradigm; each module can be replaced or improved without undermining the core contribution.
>
> - Besides, we also have some conceptual advances. For example, our treatment of tree sampling includes the design principles, construction of samplers with the required properties, and theoretical guarantees. These cannot be reduced to ``two-pass tree DP.'' We hope this clarifies that the novelty lies in the paradigm and its theoretical underpinnings, rather than in a mere assembly of existing techniques.
>
> **Weakness 2: broader tasks or settings for practical generality**
>
> We incorporate additional experiments on various tasks to show our practical generality, including:
>
> **(1) The noisy-feature tasks:** We add noise of various levels to node features and conduct node classifications.
> The performance is reported in Fig. 17 and Fig. 18 of Sec. J.4, where we obtain notable improvements of up to $77.7$\% over pairnorm on Pubmed with $5$\% noise, $43.9$\% over Meannorm and $56.0$\% over Difformer on Cora with $5$\% noise.
> These results show our consistently better robustness compared with existing baselines.
>
>
> **(2) Larger-scale graph:** We incorporate a new larger-scale graph dataset and report the results in Fig. 19 and Tab. 5 of Sec. J.5, showing a similar superiority of final performance to other datasets. For example, we obtain an improvement of approximately $27.5$\% over Difformer and $14.9$\% over Meannorm. Additionally, our framework shows a significant speed advantage, running approximately $38.6$\% faster than Meannorm and $38.7$\% faster than Pairnorm, further highlighting its efficiency in larger-scale graph tasks.
>
>
> **(3) More heterophilous baselines:** We additionally incorporate some strong baselines that are specifically designed for heterophilous graphs.
> The results are reported in Tab. 6 of Sec. J.9, in which we achieve an improvement of nearly $5.7$\% over ADPA, $2.9$\% over GESN, and $5.6$\% over HiGN on average.
> From these results, we observe clearly that our proposed framework can still obtain competitive performance against strong baselines specifically designed for heterophilous graphs, highlighting the effectiveness of our paradigm in handling heterophily.
>
>
> **(4) Graph classification:**
> We also incorporate a graph classification task on ENZYMES benchmark, with the results shown in Fig. 21 of Sec. J.7.
> For example, we find that our framework outperforms GIN with Attributes and GCN with Attributes by approximately $13.1$\% and $4.9$\%, respectively.
> These findings show that our framework can still achieve superior accuracy against baselines, further demonstrating the generality and the effectiveness.
>
>
> We have incorporated all of these experimental results and comparisons into our manuscript and hope to strengthen and broaden our potential.
>
> **Weakness 3: hyperparameter sensitivity**
>
> We introduce several hyper-parameters, but most of them are easy to tune with a simple random search.
> Notably, in Sec. J.1 of Appendix in our manuscript, we have already studied and analysed the effects of nearly all of them, e.g., $N_T$, $\beta_1$, $\beta_2$, and $\gamma$.
> We empirically find that $\beta_1$ and $\beta_2$ have a minimal impact on the final performance, demonstrating the robustness of our framework and training to these hyper-parameters.
> Despite the different peaks of $N_T$ and $\gamma$, they exhibit some shared general trends across different benchmarks, namely an initial increase followed by a decrease, indicating that we should choose intermediate values of these parameters.
> In fact, $N_T\in [4,10]$ and $\gamma\in [0.3,0.6]$ are generally competitive in most cases.
>
> In addition to these already included studies, we also supplement a new study on the effects of the hidden dimension $d$, with results reported in Fig. 14 of Sec. J.1.4.
> As observed from the results, the final performance is relatively robust to this hyper-parameter.

---

> ### Author Response · Authors · 2025-11-23
>
> **Weakness 4: when our paradigm might underperform?**
>
> Thank you for raising such an interesting and valuable question, particularly for those works on new paradigms like ours.
> Following your suggestions, we conduct additional experiments (in Sec. J.4 and J.6) to study whether our paradigm might underperform in the suggested situations.
>
> First, we empirically find that with gradually increased feature noise, our final performance will gradually decrease.
> But similar phenomena is also observed from other baselines.
> Fortunately, we also find that **our framework is more robust to feature noise than baselines** (performing better at the same noise level, in Fig. 17 and Fig. 18 of Sec. J.4).
> Indeed, we further mitigate the negative impacts of noise by improving the homophily estimator, which would be explored in future.
>
>
> Second, we acknowledge that high density of graphs might cause insufficient extraction of global dependency patterns.
> However, this issue can be addressed by appropriately **incorporating a few more trees with a manageable complexity**, which has been verified via empirical evident (Fig. 20 of Sec. J.6).
> Besides, we also notice that too many trees seem to be unnecessary due to possible structural redundancy and over-smoothing risks.
>
> Overall, the above analysis shows that our paradigm has strong generality and flexibility.
> With only minor modifications or extensions, it can be easily adapted for different situations and settings.
>
>
> *Thank you once again for your insightful feedback.
> We will incorporate all the experiments and the discussions into our manuscript to improve the quality.*

---

> ### Comment · Reviewer_vja3 · 2025-11-25
>
> Thank you for your efforts during the rebuttal stage. Your responses have addressed my concerns to a certain extent, and I have accordingly raised my score.

---

> > ### Author Response · Authors · 2025-11-25
> >
> > We also sincerely appreciate you for the thoughtful feedback and for raising the score. We will refine the final version of the paper with all the discussions and experiments to improve its quality.

---

### Official Review · Reviewer_oK4D · 2025-10-28

**Soundness:** 3
**Presentation:** 3
**Contribution:** 3
**Rating:** 6
**Confidence:** 4

**Summary:**

The paper propose the forest-based graph learning paradigm, which achieve the efficient long-range information propagation. The core idea is to reinterpret graph message passing as transport over spanning trees.

**Strengths:**

Novel conceptual paradigm. The proposed Tree Aggregator is a meaningful and technically novel formulation that enables composable and reversible message passing on trees, achieving efficient long-range propagation with linear complexity.

Well writing.

The experiment section is solid and broad. The evaluation includes both homophilous and heterophilous benchmarks with extensive baselines, ablations, and hyperparameter analysis. The empirical performance (especially on Wisconsin, Texas) is notably strong.

**Weaknesses:**

While the paper emphasizes “forest-based” modeling, the connection to existing tree decomposition / hierarchical GNN approaches(For example, [1]) is not clearly distinguished.

Overly complex pipeline: The four-stage pipeline (preprocessing → tree sampling → tree aggregator → tree fuser) introduces multiple training stages and parameters (β₁, β₂, γ, Kᴸ, N_T), yet lacks analysis on training stability and sensitivity.


[1] Fast and Effective GNN Training with Linearized Random Spanning Trees

**Questions:**

See weaknesses.

---

> ### Author Response · Authors · 2025-11-23
>
> *We sincerely thank you for your insightful and constructive comments. We have carefully considered your feedback and responded to each of your points below.*
>
> **Weakness 1: connection and differences compared with existing works**
>
> Thank you for providing such a valuable question.
> The referenced methods have some similarities with ours (also introducing trees), but we are fundamentally different.
>
> **(1) Core ideas**:
> They introduce random trees mainly to accelerate the local GNN aggregations, by leveraging the tree sparsity to reduce the average number of neighbors.
> In contrast, we aim to break the severe trade-off between complexities and comprehensive global aggregations, where we leverage the global connectivity and compactness of the trees to enable efficient long-distance knowledge propagation.
> Notably, their methods still suffer from the above trade-off, since they focus more on local aggregation and still require layer stacking to cover global receptive fields.
>
> **(2) Techniques to sample trees and utilize trees are different**:
> - They sample trees from a uniform distribution, while our technical framework samples trees from a distribution that theoretically biases towards homophily, facilitating beneficial node knowledge propagations in a tree.
> - They deal with such trees via linearization based on path splits or a straightforward depth-first search (DFS) visit order.
> Yet, our work preserve the holistic tree topologies and proposes a powerful tree aggregator to explicitly address knowledge propagation along trees, achieving global coverage in a single layer while keeping efficiency.
>
> Furthermore, we supplement some comparative experiments and report the results in Fig. 23 of Sec. J.10.
> These findings empirically demonstrates that our technical framework consistently achieves better results compared to the referenced methods (average $4.1$\% relative performance gain).
>
> **Weakness 2.1: overly complex pipeline**
>
> While we introduce a four-stage technical pipeline, most of the stages are very simple and easy to implement.
> For example, the pre-processing stage is just to pre-train a single-layer linear or GCN.
> The fusion stage is simply a Mean fusion.
> The key technical contributions in our framework are the tree sampler and the tree aggregation.
> We detail all these stages in our paper primarily to facilitate a clearer understanding for readers and better reproducibility, and we do not highlight other simple stages as our main contribution.
>
> **Weakness 2.2: hyper-parameter sensitivity**
>
> We introduce several hyper-parameters, but most of them are easy to tune with a simple random search.
> Notably, in Sec. J.1 of Appendix in our manuscript, we have already studied and analysed the effects of nearly all of them, e.g., $N_T$, $\beta_1$, $\beta_2$, and $\gamma$.
> We empirically find that $\beta_1$ and $\beta_2$ have a minimal impact on the final performance, demonstrating the robustness of our framework and training to these hyper-parameters.
> Despite the different peaks of $N_T$ and $\gamma$, they exhibit some shared general trends across different benchmarks, namely an initial increase followed by a decrease, indicating that we should choose intermediate values of these parameters.
> In fact, $N_T\in [4,10]$ and $\gamma\in [0.3,0.6]$ are generally competitive in most cases.
> The hyper-parameter $K_L$ is generally not critical, and is set to either $1$ or $2$.
>
> In addition to these already included studies, we also supplement a new study on the effects of the hidden dimension $d$, with results reported in Fig. 14 of Sec. J.1.4.
> As observed from the results, the final performance is relatively robust to this hyper-parameter.
>
> *We are profoundly grateful for your invaluable time and dedication in reviewing our work. We will incorporate these discussions and experimental results, which significantly refine our manuscript.*

---

> > ### Comment · Reviewer_oK4D · 2025-11-26
> >
> > Thank you for the detailed clarification. My concerns are essentially resolved, and I will maintain my positive rating.

---

### Official Review · Reviewer_Tajx · 2025-10-30

**Soundness:** 3
**Presentation:** 2
**Contribution:** 2
**Rating:** 4
**Confidence:** 5

**Summary:**

This paper proposes a Forest-Based Graph Learning paradigm for semi-supervised node classification, addressing the trade-off between cost-effectiveness and global receptive field in existing GNNs and Graph Transformers. FGL models information propagation as transport over a forest of spanning trees, generated via a homophily-guided sampler. It incorporates a linear-time tree aggregator for efficient long-range interaction and a tree fuser to merge multi-tree knowledge. Experiments on show FGL outperforms state-of-the-art methods (e.g., 11.9% gain over GCNII, 16.1% over DIFFormer) with linear complexity and faster runtime.

**Strengths:**

1.Breaks the local-global trade-off by leveraging spanning trees, enabling efficient global coverage with low structural cost.

2. Linear time/space complexity per epoch, with 2–5× speedup over GTs (e.g., DIFFormer) and deep GNNs (e.g., GCNII).

3. Excels on both homophilous (Cora, Pubmed) and heterophilous (Actor, Cornell) graphs, with robust generalization under label scarcity.

**Weaknesses:**

1.Using a forest to address homophily–heterophily problems is not a particularly novel idea, as many path-based methods [1] also aim to expand the receptive field. The idea of employing trees follows a similar intuition, broadening the range to enhance the model’s ability to capture global structural knowledge.

2.In line 252, there seems to be a definition error — what does Fa(u) and g(u) represent, and how do they satisfy Properties (I) and (II)? If both letters H and S denote embeddings, it would be better to use a consistent notation, otherwise it becomes confusing. I suggest the authors clarify this theorem more explicitly.

3.My main concern with this method lies in the semi-supervised setting: how to accurately characterize homophily becomes a major issue. The paper uses “the cross-entropy loss with targets... We train the local graph attention by minimizing...” to obtain pseudo labels. However, if the model performs poorly, the predicted homophily may deviate significantly from the true one, leading to a tendency toward over-smoothing.

4.In Section 4.1, what does c represent? If Y is a one-hot label, then finding k nearest neighbors in the representation space seems meaningless.

5.The paper lacks a discussion of related work on heterophilous graphs, and there is also a lack of baselines specifically designed for heterophilous GNNs

**Questions:**

See weaknesses

---

> ### Author Response · Authors · 2025-11-23
>
> *We are sincerely thankful for your thorough and insightful feedback on our work. We address your specific queries below.*
>
> **Weakness 1: discussion on differences between path-based GNNs and FGL**
>
> Thank you for raising this point regarding path-based GNNs. We clarify below why our forest-based paradigm is both novel and substantially different from prior path-based approaches, from both theoretical and empirical perspectives.
>
> Briefly, path-based GNNs become prohibitively expensive when pursuing full global coverage, while our paradigm maintains linear complexities.
> Specifically, their total complexities reach as high as $O(n\times l)=O(n^2)$, since they need to sample paths for each node and in this case each path must be extended to $l=O(n)$ length to cover all nodes (if using only a single layer).
> In contrast, based on the insight that a tree is the smallest structure that connects *all* nodes, we propose a tree-based paradigm wherein all nodes share a single tree, thereby eliminating redundant sampling for each node.
> Besides, simple linear recursions can be included whereby aggregations of all nodes can be completed in linear time, since substantial reusable knowledge exists in global aggregation behaviors of adjacent nodes on a shared tree.
> In summary, as our key differences, **tree sharing** and **linear recursions** help us to break the coverage–complexity trade-off that fundamentally limits path-based approaches, and thus those counterparts cannot derive our paradigm via simple extension.
>
> We also compare with a recent path-based method, PAIN [1]. As we could not locate the specific reference you mentioned, we use PAIN as a representative alternative. According to Fig. 22 in Sec. J.8, our approach consistently outperforms it (average $3.7$% relative gain), demonstrating the empirical superiority of our method.
>
> [1] No PAIN no Gain: More Expressive GNNs with Paths
>
> **Weakness 2: symbol and notational interpretations**
>
> In Line 252: "Denote the father node and the children nodes of $u$ on tree $T$ as $\operatorname{Fa}\left(v\right)$ and $\operatorname{Child}\left(v\right)$."
> Here, "of $u$" should be "of $v$", which is a typo.
> In a tree $T$ rooted at $r\in V$, we denote as  $\operatorname{Fa}\left(v\right)$ the unique father node of node $v$.
> $g\left(\cdot\right)$ denotes a general feature transformation function, which is applied to node features or embeddings, such as $g\left(H_u\right)$ in Eq. (6).
> Properties (I) and (II) are mainly related to the basic message aggregator $f_{\operatorname{Agg}}\left(\cdot\right)$, which is not related to the symbols $\operatorname{Fa}\left(\cdot\right)$ and $g\left(\cdot\right)$.
>
> Here, $H \in \mathbb{R}^{n \times d}$ and $H' \in \mathbb{R}^{n \times d}$ denote node embeddings before and after a tree aggregating operation.
> Different from them, $S \in \mathbb{R}^{n \times d}$ is an internal matrix that stores the aggregated messages from different subtrees, which mainly facilitates the subsequent efficient global message aggregation in Recursion (II), i.e., Eq. (6).
>
> Briefly, Theorem $1$ says that for any basic message aggregator $f_{\operatorname{AGG}}$ that satisfies Properties (I) and (II), we can always construct an algorithm to efficiently aggregate all global messages along tree paths simultaneously for each node in a tree with only linear time cost, which can be formulated exactly as Recursion (I) and (II).
>
> We are sorry about these unavoidably introduced symbols and hope the clarification for notations can address your misunderstandings.
> We consider introducing a symbol table in our manuscript.

---

> ### Author Response · Authors · 2025-11-23
>
> **Weakness 3: justification of pre-trained local attention**
>
> Thank you for providing such a valuable question.
> Our main contribution in this work is the introduction of a novel forest-based graph learning paradigm (not just an implementation), where we sample trees based on edge-homophily estimates.
> We implement the edge-homophily estimator in our technical framework via simple attention-based pre-training **just for simplicity**.
>
> First, it should be noted that any sophisticated edge-homophily estimator can be employed to further improve the final performance, such as state-of-the-art graph contrastive learning-based methods and heterophily-oriented techniques.
> In other words, this attention-based implementation is not the only choice in our technical framework, and it is our default choice only due to its simplicity, light-weightness, and ease of implementation.
> Substituting this with other candidates does not undermine the value of our paradigm.
>
> Second, we aim to demonstrate that even utilizing a simple attention-based implementation has already achieved competitive performance against various baseline categories, which clearly highlights the potential and value of our paradigm.
> In fact, in these real-world graph benchmarks, the pre-trained attentions have already successfully identified a large portion of real edge homophily, since attentions can intrinsically filter out some node noise and facilitate beneficial knowledge propagation for node classifications.
> To justify its effectiveness, we empirically verify the homophily rates it can achieve: $87.1$% on Cora, $82.0$% on Citeseer, and $86.5$% on PubMed, which we believe is high enough.
> Even so, practitioners can always replace this with more sophisticated candidates to further improve the homophily rates, which may, however, risk higher complexities.
>
> **Weakness 4: interpretation of details of KNN**
>
> In Sec. 4.1, we introduce the symbol $c$ to denote the number of label classes for a node, $Y \in \{0,1\}^{n \times c}$ denotes the ground-truth node labels in one-hot forms, and use $Y' \in [0,1]^{n \times c}$ to denote pseudo-labels.
> Given that the rows of $Y'$ are continuous probabilistic vectors, it is reasonable and meaningful for us to find K-nearest neighbors based on $Y'$.
>
> **Weakness 5: adding related work and baselines on heterophily graphs**
>
> We are glad to strictly follow your suggestions - incorporating a new section of detailed related works on heterophilous graphs (Sec. N.1) and introducing some baselines specifically designed for heterophilous graphs.
> We provide the additional comparative results in Tab. 6 of Sec. J.9, from which we can clearly observe that our framework can also achieve competitive performance against those strong state-of-the-art counterparts tailored for heterophilous graphs, further highlighting our potential and merits.
>
> *Thank you once again for your thoughtful and detailed feedback, which offers valuable guidance for the improvement of our paper.
> We have already incorporated the above discussions and experimental results into our manuscript.*

---

> > ### Comment · Reviewer_Tajx · 2025-11-25
> >
> > Thank you for the detailed response. I am willing to increase my score, and I hope the revised manuscript provides clearer  notation.

---

> > > ### Author Response · Authors · 2025-11-25
> > >
> > > Thank you once again for the efforts you have put into reviewing our manuscript and for raising your score. We will include these valuable discussions and experiments in the final version of the paper to enhance its quality.

---

### Official Review · Reviewer_GwAw · 2025-11-02

**Soundness:** 3
**Presentation:** 3
**Contribution:** 4
**Rating:** 8
**Confidence:** 4

**Summary:**

This paper proposes a Forest-based Graph Learning (FGL) framework to address the trade-off between global receptive fields and computational efficiency in graph neural networks. The authors reinterpret graph information propagation as a transmission process across multiple spanning trees (a forest) to efficiently achieve long-range information aggregation. Methodologically, the paper designs a tree sampler based on homogeneity estimation and a linear-time tree aggregator, achieving Transformer-like global interaction capabilities while maintaining linear complexity. Experimental results demonstrate that FGL outperforms existing GNN and Graph Transformer models on multiple semi-supervised node classification tasks while maintaining significant training efficiency.

**Strengths:**

1. This is an interesting paper. The methodology of this paper represents a significant innovation compared to GNN and GT.
2. I really like the design of Tree Aggregator and it's quite interesting. The paper also provides ample theoretical analysis.
3. The paper is well-written and easy to follow. The experimental results demonstrate the superiority of this method.

**Weaknesses:**

1. Given that the paper emphasizes the efficiency of this method, experiments on large-scale graph datasets may be warranted. In my view, the experimental data presented in the paper is insufficiently large.
2. Compared to GNN and GT, the expressive power advantage of FGN remains unknown. I look forward to the authors providing further insights on this aspect in their paper.

**Questions:**

Does FGL struggle with relatively dense graphs, resulting in more edges and information being discarded in the generated tree compared to the original graph?

---

> ### Author Response · Authors · 2025-11-23
>
> *We are extremely grateful for your detailed constructive feedback and positive recognition on our manuscript. Below, we address your questions point-by-point.*
>
> **Weakness 1: experiments on a larger-scale graph**
>
> We strongly agree with you that it is necessary to supplement a larger graph benchmark to further demonstrate our superiority in efficiency.
> We incorporate a new large-scale graph dataset (AMiner-CS) with $593,486$ nodes and $6,217,004$ edges, which is widely adopted by many GNN works (such as InterpGNN [1] and DUALFormer [2]).
> We report the experimental results in Fig. 19 and Tab. 5 (Sec. J.5).
> Compared with Graph Transformer methods, we obtain a $27.6$% relative performance gain against Difformer with a reduced $39.8$% running time.
> Besides, compared with Deep GNN methods, we obtain a $15.0$% relative performance gain against Meannorm with a reduced $87.0$% running time.
> Observed from these results, we can clearly see that even on a larger graph, our superiority still remains while keeping efficiency.
>
> [1] InterpGNN: Understand and Improve Generalization Ability of Transdutive GNNs through the Lens of Interplay between Train and Test Nodes, ICLR 2024.
>
> [2] DUALFormer: Dual Graph Transformer, ICLR 2025.
>
>
> **Weakness 2: insights of expressive power**
>
>
> It is well-known that typical GNNs and Graph Transformers have expressivities restricted in 1-WL.
> In fact, surpassing 1-WL theoretically is challenging, since it may introduce higher complexities (e.g, higher-order GNNs) or sophisticated structure encodings with extra conditions presented in [1].
>
> Currently, although we might not strictly prove that the expressivity of our FGL paradigm can surpass 1-WL, we can provide some insights into the ability to identify some non-isomorphic graphs.
> Specifically, in Sec. A.7, we present two example graphs, $G_1$ and $G_2$, that are not isomorphic and can be successfully identified by our paradigm, yet fail to be identified by typical GNNs.
> These insights show that our paradigm has potential to efficiently surpass 1-WL.
> The rigorous proof would be an interesting and valuable question deserving further exploration in future.
>
> Despite so, we have included an initial theoretical result in Sec. A.7 that the proposed paradigm can successfully identify two given isomorphic graphs, which is a critical preliminary to further prove the expressive power.
>
> [1] On Structural Expressive Power of Graph Transformers
>
>
> **Question 1: Does FGL struggle with relatively dense graphs?**
>
> Thank you for providing such an valuable question regarding the generality of our proposed paradigm.
>
> We acknowledge that high density of graphs might cause insufficient extraction of global knowledge.
> However, this issue can be addressed by appropriately incorporating a few more trees with a manageable complexity, which has been verified via empirical evident (Fig. 20 of Sec. J.6).
> Besides, we also notice that too many trees seem to be unnecessary due to possible structural redundancy and over-smoothing risks.
>
> *Thank you once again for your valuable time and effort in reviewing our work.
> We will supplement these discussions, which can significantly improve the quality of our manuscript.*

---

### Author Response · Authors · 2025-12-01
**To Area Chair**

We sincerely appreciate the Area Chair’s time and effort in reviewing our work and navigating the rebuttal history.

Our paper proposes a **novel Forest-Based Graph Learning (FGL) paradigm** that fundamentally resolves graph learning’s long-standing dilemma: breaking the challenging trade-off between *global receptive field* and *computational efficiency*. We sample spanning trees with an elaborate distribution to facilitate long-range knowledge propagations (with theoretical guarantees) and design a **linear-time tree aggregator** for efficient pair-wise global node interactions.

Reviewers have explicitly recognized key strengths of our paradigm, such as its “*significant innovation compared to GNNs and Graph Transformers*” and the “*meaningful and interesting Tree Aggregator design*” supported by solid theoretical analysis.

In rebuttal, we fully addressed all reviewers’ concerns: supplemented experiments (larger-scale graphs, noisy-feature tasks, graph classification, heterophilous baselines) and clarified differences from prior works, added theoretical insights on expressivity, clarified notations, added more related work, and validated hyperparameter robustness.

As all concerns were fully resolved, ***reviewers raised their scores before the known OpenReview leakage, leading to consistent positive ratings ($\ge$6, average 6.5)***. We again thank the AC for their careful attention.

---

### Meta-Review · Area_Chair_FCPr · 2026-01-01

**Summary:**

In this paper, the authors propose a Forest-Based Graph Learning (FGL) paradigm for semi-supervised node classification, aiming to resolve the long-standing trade-off between global receptive field and computational efficiency in graph learning. The reviewers found the core idea of modeling message passing as transport over a small forest of spanning trees to be novel and technically interesting. There were initial concerns on scalability, novelty compared to prior tree- or path-based methods, pipeline complexity, hyperparameter sensitivity, and clarity of notation. Through a thorough rebuttal, the authors added large-scale experiments, clarified theoretical distinctions from prior work, expanded empirical evaluation to heterophilous and noisy settings, and addressed notation and stability concerns. Overall, most reviewers increased their scores after rebuttal.

**Reviewer Concerns:**

Concerns addressed by the rebuttal:

1. The reviewers requested validation on larger datasets given the paper’s emphasis on efficiency. The authors added experiments on a large-scale graph (e.g., AMiner-CS), demonstrating that FGL maintains both accuracy gains and runtime advantages over deep GNNs and Graph Transformers.

2. Several reviewers questioned whether the approach was a straightforward extension of existing path-based or random-tree methods. The authors clarified both theoretically and empirically that FGL differs in its homophily-guided tree sampling, shared-tree global aggregation, and linear-time tree recursion, and provided direct comparisons with representative baselines.

3. The reviewers raised concerns about missing baselines/experiments for heterophilous graphs and noisy features added. The added experiments show consistent performance improvements over heterophily-oriented baselines and stronger robustness under feature noise.

Outstanding concerns:

1. Although robustness was demonstrated, some reviewers still noted that the multi-stage design and number of components may increase implementation complexity for practitioners, even if individual modules are simple.

2. While the authors added graph classification and noisy-feature experiments, most results still focus on semi-supervised node classification. The applicability of the forest paradigm to broader tasks remains less explored.

**Reviewer Scores:**

Reviewer GwAw: This reviewer initially gave 8 and would likely remain this score.

Reviewer Tajx: This reviewer initially gave 4. They explicitly raised the score after rebuttal, indicating that most of their concerns were resolved. So, likely they would have given 6 after rebuttal.

Reviewer oK4D: This reviewer originally gave 6. Their concerns about novelty and pipeline complexity were addressed during the rebuttal. The reviewer confirmed maintaining a positive rating after rebuttal.

Reviewer vja3: This reviewer initially gave 4. They raised their score after rebuttal, indicting that their concerns were addressed to a significant extent. So, likely the score would have been raised to 6 after rebuttal.

---

### Decision · Program_Chairs · 2026-01-26

Accept (Poster)